# Amino acid–dependent TSC2 dephosphorylation by lysosome–PP2A regulates mTORC1 signaling transduction

Takanori Nakamura[1,2,3,4,*] , Shigeyuki Nada[5,*] , Masaki Matsumoto[6], Nuha Loling Othman[7], Hidetaka Kosako[8] , Kazuki Ikeda[9] , Naohiko Koshikawa[9,10], Junya Masumoto[3], Tatsuya Sawasaki[2], Mutsuhiro Takekawa[1,11] , Takashi Suzuki[7], Masato Okada[5,12]

The mammalian target of rapamycin complex 1 (mTORC1) signaling pathway, composed of amino acid (AA)–sensing (Ragulator/LAMTOR-Rag) and growth factor (GF)–sensing (AKT-TSC1/2-Rheb) axes, pivotally regulates intracellular anabolism and catabolism. mTORC1 deregulation is associated with various metabolic diseases, including cancer and diabetes. As a key regulator of nutrient signaling, mTORC1 integrates a variety of nutrient signals. However, signal integration and crosstalk in the mTORC1 pathway remain incompletely understood. Therefore, in this study, we aimed to understand the complex mTORC1 signaling cascade by constructing an integrated mathematical model of temporal mTORC1 regulation using two AA-sensing and GF-sensing axes. Mathematical simulations and experimental data revealed robust AKT phosphorylation (P-T308/P-S473) after insulin stimulation, regardless of the intracellular AA levels. Conversely, AKT-mediated inhibitory TSC2 phosphorylation (P-T1462) substantially diminished during AA deprivation compared with AA treatment. Furthermore, we highlighted PP2A-mediated TSC2 dephosphorylation during AA removal, ensuring complete mTORC1 activation only upon concurrent AA and GF sensing. Thus, we elucidated mTORC1 signaling dynamics, revealing the complex interplay between AAs and GFs and offering insights into metabolic regulation.

## Introduction

The evolutionarily conserved serine–threonine kinase mammalian target of rapamycin (mTOR) serves as the catalytic core of the structurally and functionally distinct multiprotein complexes, mTOR complex 1 (mTORC1) and complex 2 (mTORC2). mTORC1 is a central signaling hub that promotes protein and lipid biogenesis and concurrently suppresses catabolic processes such as autophagy and lysosomal biogenesis (Mossmann et al, 2018; Kim & Guan, 2019; Hoxhaj & Manning, 2020; Liu & Sabatini, 2020). In response to stimuli such as extracellular growth factors (GFs) (e.g., insulin [Ins] and insulin-like GF [IGF]), mTORC2 localizes to the plasma membrane and phosphorylates AKT, also known as PKB, at S473 (Yang et al, 2015; Saxton & Sabatini, 2017), PKC (Saxton & Sabatini, 2017), and serum- and glucocorticoid-induced protein kinase (SGK) (Saxton & Sabatini, 2017), regulating cytoskeletal dynamics and cellular survival. In contrast, mTORC1 localizes to the lysosomal membrane in response to nutrient amino acids (AAs) or glucose and integrates multiple signals to promote protein synthesis and cell growth by phosphorylating its substrates, including eukaryotic initiation factor 4E–binding proteins (4E-BP) at T37/T46/S65/T70 and p70 S6 kinase (S6K) at T389.

mTORC1, which consists of mTOR, RAPTOR, DEPTOR, PRAS40, and mLST8, is activated by extracellular GFs and nutrient AAs via the AKT-TSC1/2-Rheb and Ragulator/LAMTOR-Rag pathways, respectively (Fig 1A). Upon Ins exposure, the insulin receptor (IR) associates with Ins on the surface of the plasma membrane, subsequently inducing IR activation via transautophosphorylation. Active IRs phosphorylate the cytoplasmic IR substrate (IRS), which consequently recruits phosphatidylinositol 3 kinase (PI3K) to the plasma membrane, leading to an increase in the levels of phosphatidylinositol 3,4,5-triphosphate [PI(3,4,5)P3] on the cytoplasmic side of the plasma membrane. PI(3,4,5)P3 preferentially binds to the pleckstrin homology (PH) domain of PDK, leading to its activation on the cell membrane. Activated PDK phosphorylates AKT at T308, which in turn phosphorylates tuberous sclerosis complex 2

[1]Division of Cell Signaling and Molecular Medicine, Institute of Medical Science, The University of Tokyo, Tokyo, Japan   [2]Division of Cell-Free Sciences, Proteo-Science Center, Ehime University, Matsuyama, Japan   [3]Division of Pathology, Proteo-Science Center and Graduate School of Medicine, Ehime University, Toon, Japan   [4]Division of Informatics and Computation, Premier Institute for Advanced Studies, Ehime University, Matsuyama, Japan   [5]Department of Oncogene Research, Research Institute for Microbial Diseases, Osaka University, Osaka, Japan   [6]Department of Omics and Systems Biology, Niigata University, Niigata, Japan   [7]Center for Mathematical Modeling and Data Science, Osaka University, Osaka, Japan   [8]Division of Cell Signaling, Fujii Memorial Institute of Medical Sciences, Institute of Advanced Medical Sciences, Tokushima University, Tokushima, Japan   [9]Department of Life Science and Technology, Institute of Science Tokyo, Yokohama, Japan   [10]Clinical Cancer Proteomics Laboratory, Kanagawa Cancer Center Research Institute, Yokohama, Japan   [11]Department of Biological Sciences, Graduate School of Science, The University of Tokyo, Tokyo, Japan   [12]Center for Advanced Modalities and Drug Delivery System, Osaka University, Osaka, Japan

Correspondence: nakamura.takanori.cb@ehime-u.ac.jp; suzuki@sigmath.es.osaka-u.ac.jp; okadam@biken.osaka-u.ac.jp
*Takanori Nakamura and Shigeyuki Nada contributed equally to this work

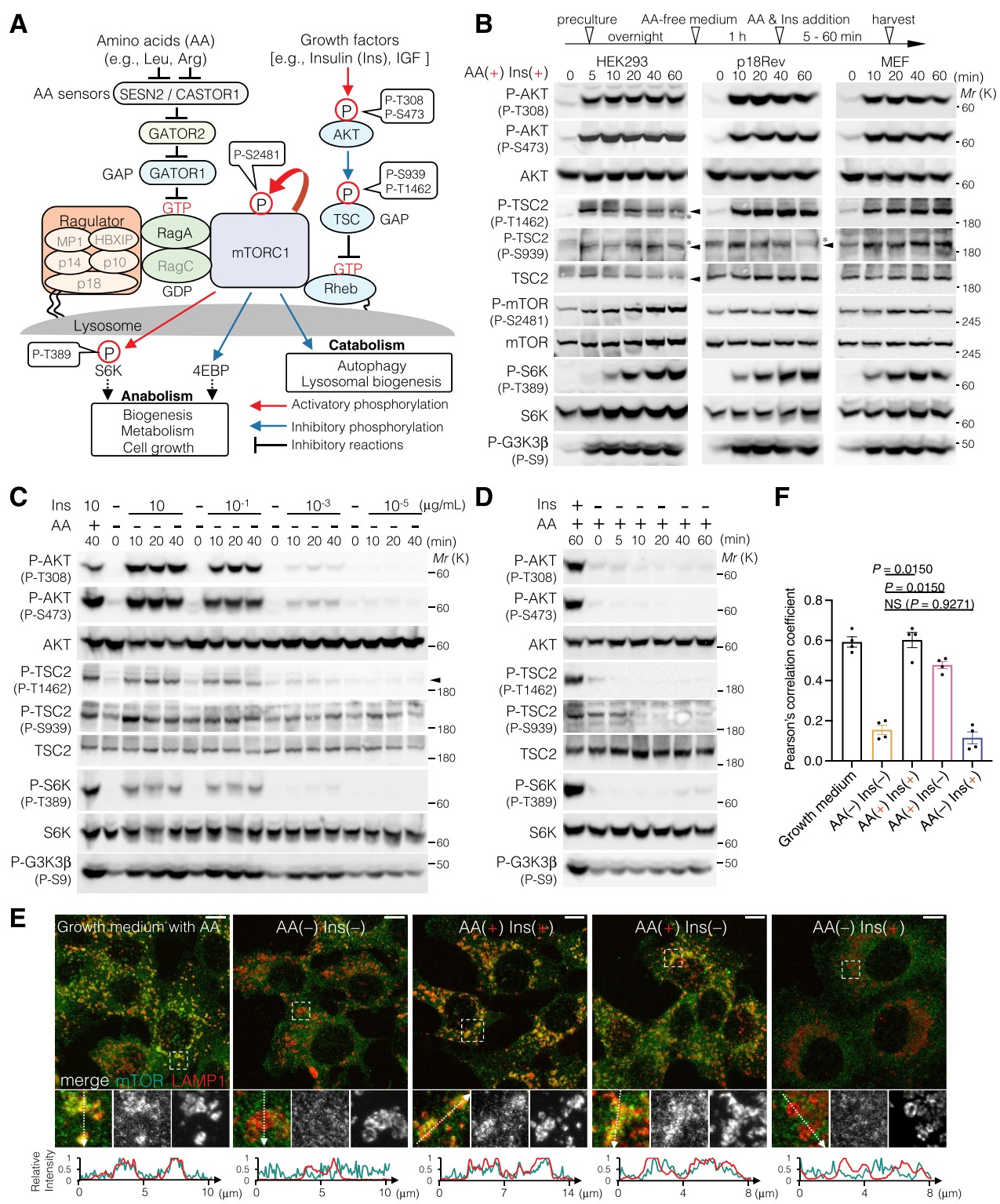

**Figure 1. mTORC1 is cooperatively activated by Ins- and AA-sensing pathways.**

**(A)** Schematic overview of the mTORC1 pathway comprising an insulin (Ins)-sensing module (AKT-TSC-Rheb) and an amino acid (AA)–sensing module (Ragulator/LAMTOR-Rag). Red arrows indicate activatory phosphorylation, blue arrows indicate inhibitory phosphorylation, and black lines indicate inhibitory reactions. GAP: GTPase-activating protein. **(B)** Phosphorylation levels of AKT at T308 and S473, TSC2 at S939 and T1462, mTOR at S2481, S6K at T389, and GSK3β at S9 monitored when HEK293 (left), and p18 re-expressing p18-KO (p18Rev) (middle) and MEF cells (right) were exposed to AA deprivation and subsequently treated with AA and Ins. Cells were cultured in minimal medium without AA and Ins for 1 h, then incubated with medium containing AA and Ins (10 μg/ml) for the indicated times. Cell lysates were analyzed by Western blotting using the indicated antibodies. *: nonspecific band. **(C, D)** Phosphorylation levels of AKT at T308 and S473, TSC2 at S939 and T1462, S6K at

(TSC2), a component of the TSC protein complex that acts as a GTPase-activating protein (GAP) for Rheb at multiple sites, including S939 and T1462 (Manning et al, 2002; Menon et al, 2014; Manning & Toker, 2017). TSC2 phosphorylation destabilizes its association with TSC1 and prevents TSC (TSC1-TSC2-TBC1D7)–induced Rheb inhibition (Inoki et al, 2002), resulting in active Rheb (Rheb$^{GTP}$) accumulation on the lysosomal surface. In addition to extracellular GF treatment, nutrient AA treatment after AA removal activates mTORC1. The sensing of nutrient AAs via Sestrin2 (SESN2) for leucine (Saxton et al, 2016b; Wolfson et al, 2016) and CASTOR1 for arginine (Saxton et al, 2016a; Chantranupong et al, 2016) leads to the dissociation of AA-sensor proteins from GATOR2, a heteropentameric complex composed of MIOS, WDR24, WDR59, SEH1L, and SEC13. The association of GATOR2 with GATOR1 (DEPDC5, NPRL2, and NPRL3), a GAP for Rag GTPases (Bar-Peled et al, 2013), inhibits GATOR1, leading to an increase in active Rag$^{GTP}$. As Rag GTPase is tethered to the lysosomal surface via the Ragulator/LAMTOR complex, a heteropentameric protein complex composed of LAMTOR1-5 (p18, p14, MP1, p10, and HBXIP, respectively) (Yonehara et al, 2017), the active form of Rag GTPase recruits mTORC1 to the lysosomal surface. Here, mTORC1 is activated following its association with Rheb$^{GTP}$ and subsequent autophosphorylation at S2481, the critical phosphorylation site for mTORC1 activation.

The mTORC1 signaling pathway is regulated via numerous GF pathways that converge on the TSC (TSC1-TSC2-TBC1D7) complex to catalyze the conversion of active Rheb$^{GTP}$ to inactive Rheb$^{GDP}$; for example, IGF, Ras, and Wnt signaling pathways induce inhibitory TSC2 phosphorylation (Inoki et al, 2006; Manning & Toker, 2017; Saxton & Sabatini, 2017), whereas the TNF-$\alpha$ pathway induces inhibitory TSC1 phosphorylation (Lee et al, 2007). As the TSC is a key negative regulator of the mTORC1 signaling pathway, loss-of-function mutations in TSC1 or TSC2 lead to the development of various cancers, as well as the tuberous sclerosis complex, characterized by hamartoma formation in multiple tissues (Henske et al, 2016). In addition, oncogenic mutations that activate Ras and PI3K-AKT pathways disrupt TSC regulation, resulting in constitutive mTORC1 activation, even in the absence of GF signals (Saxton & Sabatini, 2017; Liu & Sabatini, 2020). Thus, the TSC serves as a pivotal molecular brake in the mTORC1 signaling pathway; however, the precise integration of numerous signals and their relative impact on mTORC1 activity are not fully understood.

Mathematical modeling is a useful tool that facilitates a comprehensive understanding of complex signaling pathways with crosstalk and feedback regulation, such as the mTORC1 pathway (Sulaimanov et al, 2017). The GF-sensing (AKT-TSC1/2-Rheb) pathway has been extensively studied using various mathematical models (Sedaghat et al, 2002; Jain & Bhalla, 2009; Kubota et al, 2012; Brännmark et al, 2013; Sulaimanov et al, 2017). Although several studies in recent years have reported that interplay exists between the AA and GF signaling branches (Demetriades et al, 2014; Carroll

et al, 2016; Yang et al, 2019), to the best of our knowledge, an integrated mathematical model of mTORC1 regulation through two distinct AA- and GF-sensing axes has not yet been proposed. This gap arises primarily from the lack of quantitative data, such as the intracellular concentrations of AA-sensing module proteins and their reaction rates. Therefore, these issues require clarification to enhance our understanding of the physiological metabolic regulation mediated by mTORC1, as well as cancer pathophysiology.

Therefore, in this study, we aimed to quantify the intracellular concentrations of proteins responsible for mTORC1 signaling. In addition, based on the protein quantification data, we aimed to develop a new integrated mathematical model for mTORC1 regulation through two distinct AA- and GF-sensing axes. Using mathematical simulations and experimental data, we found that AA-dependent TSC2 dephosphorylation was a general regulatory mechanism underlying mTORC1 signal transduction. Mechanistically, cytosolic TSC2 is phosphorylated by AKT upon Ins stimulation during AA supplementation, whereas upon AA removal, cytosolic TSC2 localizes to the lysosomal surface and is preferentially dephosphorylated by lysosome-accumulated phosphatases such as protein phosphatase 2A (PP2A), a ubiquitously expressed heterotrimeric phosphatase composed of a highly conserved catalytic C subunit (C$\alpha$ and C$\beta$), a scaffold A subunit (A$\alpha$ and A$\beta$), and one of 23 regulatory B subunits (e.g., B55$\alpha/\beta/\gamma/\delta$ and B56$\alpha/\beta/\gamma/\delta/\varepsilon$) (Hoffman et al, 2017). Our results delineate a molecular mechanism by which cells precisely sense intracellular levels of nutrient AAs and GFs and induce mTORC1 activation only upon concurrent sensing of AAs and GFs.

# Results

## mTORC1 is coordinately activated by AA- and GF-sensing pathways

To comprehensively understand the intricate mTORC1 signaling pathway, the time courses of AKT phosphorylation at T308 and S473, TSC2 at S939 and T1462, S6K at T389, mTOR at S2481, and GSK3$\beta$ at S9 were monitored in normal (noncancerous) HEK293 cells treated with Ins and AA after AA deprivation. Western blot analysis using phospho-TSC2– and TSC2-specific antibodies (Fig S1A–D) showed that AKT at T308 and S473 and its targets TSC2 at T1462, and GSK3$\beta$ at S9 were completely phosphorylated within 5 min of simultaneous AA and Ins treatment, whereas mTOR at S2481 and S6K at T389 were gradually phosphorylated after AA and Ins treatment, and their phosphorylation levels peaked at 20–40 min (Fig 1B, left). Similar results were obtained in normal (noncancerous) p18Rev cells generated by reintroducing p18 into p18-knockout (KO) MEFs (Nada et al, 2009) (Fig 1B, middle, and Fig S1E and F) and control MEFs (Fig 1B, right). This indicates that the patterns of rapid AKT (~5 min) and relatively slow S6K

---

T389, and GSK3$\beta$ at S9 monitored in p18Rev cells treated with Ins and AA alone or in combination as indicated. p18Rev cells were treated with AA and/or Ins, as shown in (B). Cell lysates were analyzed by Western blotting using the indicated antibodies. **(E)** Immunofluorescence analysis of p18Rev cells treated with AA and/or Ins for 20 min, as indicated. The areas in the small squares in the top panels are enlarged and shown in the middle panels. mTOR, green; LAMP1, red. Scale bars, 10 $\mu m$. Line scans of mTOR (green) and LAMP1 (red) are shown at the bottom of the panel. **(E, F)** mTOR-LAMP1 colocalization in (E) was measured. The mean values of Pearson's correlation coefficient from four cells and the SEM are shown. *P*-values were assessed using one-way ANOVA with Tukey's multiple comparison test. NS, not significant.

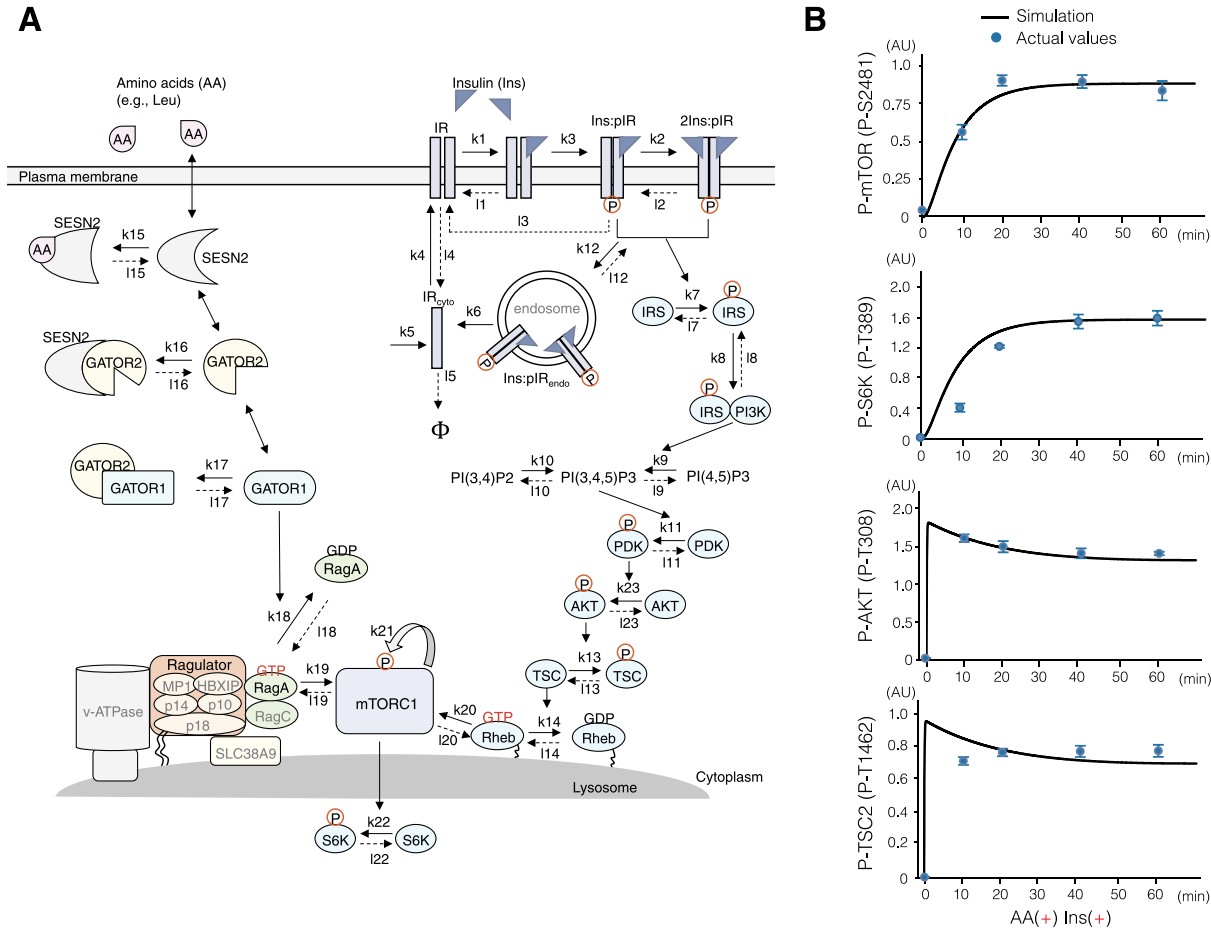

**Figure 2. Development of the mTORC1 pathway model combined with the AA- and Ins-sensing pathways.**
**(A)** Schematic representation of reactions involved in the spatiotemporal regulation of the mTORC1 pathway. A mathematical model of the AKT-TSC1/2-Rheb axis consisting of 25 equations was defined based on available mathematical models. A mathematical model of the AA-sensing (Ragulator/LAMTOR-Rag) axis was defined based on a standard reaction network system with nine equations. Reactions depicting mTORC1 activation on the lysosomal surface, mediated by the active forms of Rag$^{GTP}$ and Rheb$^{GTP}$, were defined using six equations. k (solid-line arrow) and l (dotted line arrow) represent the rate constants, whereas Φ represents the degradation of the insulin receptor (IR). **(B)** Comparison of the phosphorylation levels of mTOR at S2481, S6K at T389, AKT at T308, and TSC2 at T1462 between simulation and experimental values under AA(+) Ins(+) stimulation. The solid black lines indicate the values obtained from the simulation, whereas the blue dots indicate the values measured during the experiment. Error bars indicate the SEM (n = 3).

phosphorylation (≥20 min) upon addition of Ins plus AA were evolutionarily conserved, at least in humans (HEK293 cells) and mice (p18Rev MEFs and control MEFs). To examine the effects of AA and Ins on AKT, TSC2, and S6K phosphorylation, p18Rev cells were treated with either AA or Ins after AA removal. Immunoblot analysis revealed that Ins, and not AA, stimulation primarily induced phosphorylation of AKT at T308 and S473, as well as that of TSC2 at T1462 and S939 in a dose-dependent manner (Fig 1C and D). Stimulation with AA or Ins alone only led to a marginal increase in S6K phosphorylation at T389, compared to that with the simultaneous administration of AA and Ins (Fig 1C and D). In addition, immunofluorescence showed that after AA stimulation, mTOR rapidly localized to the lysosomal surface labeled with lysosome-associated membrane protein 1 (LAMP1), whereas under AA deprivation, mTOR exhibited a diffuse cytoplasmic distribution (Fig 1E and F). These findings suggest that Ins-triggered robust AKT phosphorylation is primarily regulated by the activation of the

kinase cascade (IR-PI3K-PDK) upstream of AKT, whereas complete mTORC1 activation (monitored by S6K phosphorylation) requires the synergistic effects of AA and Ins on the spatiotemporal regulation of mTOR, including mTORC1 autophosphorylation and AA-induced lysosomal mTORC1 localization.

## Integrated mathematical model of temporal mTORC1 regulation

To elucidate the regulatory mechanism of mTORC1 activation upon stimulation with AA and Ins alone or in combination, an integrated mathematical model of temporal mTORC1 regulation was constructed using two distinct pathways: the AA-sensing (Ragulator/LAMTOR-Rag) and GF-sensing (AKT-TSC1/2-Rheb) pathways (Fig 2A). As various theoretical studies proposed mathematical models of the AKT-TSC1/2-Rheb axis (Sedaghat et al, 2002; Jain & Bhalla, 2009), they were used to construct a mathematical model of

**Table 1. Reactions, symbols, rate constants (parameters), and units for the mathematical model.**

| Reaction | Symbol | Parameters | Unit | Comment |
|---|---|---|---|---|
| Insulin(Ins) + IR => [Ins:IR] | k1 | $6 \times 10^7$ | $M^{-1}$ $min^{-1}$ | Sedaghat et al (2002) |
| [Ins:IR] => Ins + IR | l1 | 0.2 | $min^{-1}$ | Sedaghat et al (2002) |
| Ins + [Ins:pIR ] => [2Ins:pIR] | k2 | $6 \times 10^7$ | $M^{-1}$ $min^{-1}$ | Sedaghat et al (2002) |
| 2Ins:pIR] => Ins + [Ins:pIR] | l2 | 20 | $min^{-1}$ | Sedaghat et al (2002) |
| [Ins:IR] => [Ins:pIR] | k3 | 2,500 | $min^{-1}$ | Sedaghat et al (2002) |
| [Ins:pIR] => Ins + IR | l3 | 0.2 | $min^{-1}$ | Sedaghat et al (2002) |
| IRmembrane => IRcytoplasm | k4 | $3.3 \times 10^{-4}$ | $min^{-1}$ | Sedaghat et al (2002) |
| IRcytoplasm => IRmembrane | l4 | $5.0 \times 10^{-4}$ | $min^{-1}$ | Arbitrary |
| Biosynthesis of IRcytoplasm | k5 | $1.67 \times 10^{-17}$ | $M^{-1}$ $min^{-1}$ | Sedaghat et al (2002) |
| Degradation of IRcytoplasm | l5 | $1.67 \times 10^{-18}$ | $min^{-1}$ | Sedaghat et al (2002) |
| [2Ins:pIR] + [Ins:pIR] => IRcytoplasm | k6 | 0.461 | $min^{-1}$ | Sedaghat et al (2002) |
| IRS + [Ins:pIR] or [2Ins:pIR] => pIRS + [Ins:pIR] or [2Ins:pIR] | k7 | 4.61 | $min^{-1}$ | Sedaghat et al (2002) |
| pIRS => IRS | l7 | 0.519 | $min^{-1}$ | Arbitrary |
| pIRS + PI3K => [pIRS:PI3K] | k8 | $7.1 \times 10^{11}$ | $M^{-1}$ $min^{-1}$ | Sedaghat et al (2002) |
| [pIRS:PI3K] => pIRS + PI3K | l8 | 10 | $min^{-1}$ | Sedaghat et al (2002) |
| PI(4,5)P2 => PI(3,4,5)P3 | k9 | $1.259 \times [pIRS1:PI3K]/(1 \times 10^{-13}) + 0.131$ | $min^{-1}$ | Sedaghat et al (2002) |
| | k9 (basal) | 0.131 | $min^{-1}$ | Sedaghat et al (2002) |
| | k9 (stimulated) | 1.39 | $min^{-1}$ | Sedaghat et al (2002) |
| PI(3,4,5)P3 => PI(4,5)P2 | l9 | 42 | $min^{-1}$ | Sedaghat et al (2002) |
| PI(3,4)P2 => PI(3,4,5)P3 | k10 | 2.96 | $min^{-1}$ | Sedaghat et al (2002) |
| PI(3,4,5)P3 => PI(3,4)P2 | l10 | 2.77 | $min^{-1}$ | Sedaghat et al (2002) |
| PDK => pPDK | k11 | $0.6931 \times ([PI(3,4,5)P3] - 0.31)/2.79$ | $min^{-1}$ | Sedaghat et al (2002) |
| pPDK => PDK | l11 | 6.931 | $min^{-1}$ | Sedaghat et al (2002) |
| [Ins:pIR]membrane => [Ins:pIR]endosome | k12 | $2.1 \times 10^{-3}$ | $min^{-1}$ | Sedaghat et al (2002) |
| [Ins:pIR]endosome => [Ins:pIR]membrane | l12 | $2.1 \times 10^{-4}$ | $min^{-1}$ | Sedaghat et al (2002) |
| TSC + pAKT => pTSC + pAKT | k13 | $3.6 \times 10^2$ (wo/w-crosstalk model), 0.6 (compartment model) | $min^{-1}$ | Jain & Bhalla (2009) |
| | Km | $10.3 \times 10^{-6}$ | M | Jain & Bhalla (2009) |

**Table 1.  Continued**

| Reaction | Symbol | Parameters | Unit | Comment |
|---|---|---|---|---|
| pTSC + PP2A => TSC + PP2A | l13 | 0.6 (wo-crosstalk model), 4.0 (w-crosstalk model), 0.5 (compartment model) | min$^{-1}$ | Arbitrary |
|  | Km | 8.8 × 10$^{-6}$ | min$^{-1}$ | Jain & Bhalla (2009) |
| RhebGTP + TSC => RhebGDP + TSC | k14 | 1.2 × 10$^{3}$ | min$^{-1}$ | Jain & Bhalla (2009) |
|  | Km | 0.3 × 10$^{-6}$ | M | Jain & Bhalla (2009) |
| RhebGDP => RhebGTP | l14 | 12 (wo- and w-crosstalk model), 14 (compartment model) | min$^{-1}$ | Jain & Bhalla (2009) |
| The concentration of phosphorylated surface receptors achieved after maximal insulin stimulation | IRp | 8.97 × 10$^{-13}$ | M | Sedaghat et al (2002) |
| Amino acid (AA) + SESN2 => [AA:SESN2] | k15 | 50 | M$^{-1}$ min$^{-1}$ | Arbitrary |
| [AA:SESN2] => AA + SESN2 | l15 | 1.0 × 10$^{-2}$ | min$^{-1}$ | Arbitrary |
| SESN2 + GATOR2 => [SESN2:GATOR2] | k16 | 2.5 × 10$^{-2}$ | M$^{-1}$ min$^{-1}$ | Arbitrary |
| [SESN2:GATOR2] => SESN2 + GATOR2 | l16 | 1.0 × 10$^{-2}$ | min$^{-1}$ | Arbitrary |
| GATOR2 + GATOR1 => [GATOR2:GATOR1] | k17 | 2.5 × 10$^{-2}$ | M$^{-1}$ min$^{-1}$ | Arbitrary |
| [GATOR2:GATOR1] => GATOR2 + GATOR1 | l17 | 1.0 × 10$^{-2}$ | min$^{-1}$ | Arbitrary |
| [Ragulator:RagAGTP] + GATOR1 => [Ragulator:RagAGDP] + GATOR1 | k18 | 2.5 × 10$^{-2}$ | min$^{-1}$ | Arbitrary |
|  | Km | 1 × 10$^{-7}$ | M | Arbitrary |
| [Ragulator:RagAGDP] => [Ragulator:RagAGTP] | l18 | 1.0 × 10$^{-2}$ | min$^{-1}$ | Arbitrary |
| [Ragulator:RagAGTP] + mTORC1 => [Ragulator:RagAGTP:mTORC1] | k19 | 2.5 × 10$^{-2}$ | M$^{-1}$ min$^{-1}$ | Arbitrary |
| [Ragulator:RagAGTP:mTORC1] => [Ragulator:RagAGTP] + mTORC1 | l19 | 1.0 × 10$^{-2}$ | min$^{-1}$ | Arbitrary |
| [Ragulator:RagAGTP:mTORC1] + RhebGTP => [Ragulator:RagAGTP:mTORC1:RhebGTP] | k20 | 2.5 × 10$^{-2}$ | M$^{-1}$ min$^{-1}$ | Arbitrary |
| [Ragulator:RagAGTP:mTORC1:RhebGTP] => [Ragulator:RagAGTP:mTORC1] + RhebGTP | l20 | 1.0 × 10$^{-2}$ | min$^{-1}$ | Arbitrary |
| [Ragulator:RagAGTP:mTORC1:RhebGTP] => [Ragulator:RagAGTP:pmTORC1:RhebGTP] | k21 | 2.5 × 10$^{-2}$ | min$^{-1}$ | Arbitrary |
| [Ragulator:RagAGTP:pmTORC1:RhebGTP] => [Ragulator:RagAGTP:mTORC1:RhebGTP] | l21 | 1.0 × 10$^{-2}$ | min$^{-1}$ | Arbitrary |
| S6K + [Ragulator:RagAGTP:pmTORC1:RhebGTP] => pS6K + [Ragulator:RagAGTP:pmTORC1:RhebGTP] | k22 | 3.6 (wo- and w-crosstalk model), 60 (compartment model) | min$^{-1}$ | Jain & Bhalla (2009) |
|  | Km | 0.8 × 10$^{-6}$ | M | Jain & Bhalla (2009) |
| pS6K => S6K | l22 | 60 (wo- and w-crosstalk model), 3.7 (compartment model) | min$^{-1}$ | Jain & Bhalla (2009) |
|  | Km | 8.8 × 10$^{-6}$ | M | Jain & Bhalla (2009) |
| AKT + pPDK => pAKT + pPDK | k23 | 600 | min$^{-1}$ | Jain & Bhalla (2009) |
|  | Km | 0.4 × 10$^{-6}$ | M | Jain & Bhalla (2009) |
| pAKT => AKT | l23 | 10$^{8}$ | min$^{-1}$ | Jain & Bhalla (2009) |
| AA + PP2A => [AA:PP2A] | k24 | 1,000 | M$^{-1}$ min$^{-1}$ | Arbitrary |
| [AA:PP2A] => AA + PP2A | l24 | 1 | min$^{-1}$ | Arbitrary |
| TSC (cyto) => TSC (lyso) | k25 | 0.15 (AA(+)Ins(+)) 0.25 (AA(+)Ins(−)) 0.4 (AA(−)Ins(+)) | min$^{-1}$ | Arbitrary |

**Table 1.** Continued

| Reaction | Symbol | Parameters | Unit | Comment |
|---|---|---|---|---|
| TSC (lyso) => TSC (cyto) | l25 | 0.85 (AA(+)Ins(+)) 0.75 (AA(+)Ins(−)) 0.6 (AA(−)Ins(+)) | min$^{-1}$ | Arbitrary |
| TSC (lyso) => pTSC (lyso) | k26 | 0 | min$^{-1}$ | Arbitrary |
| pTSC (lyso) + PP2A (lyso) => TSC (lyso) + PP2A (lyso) | l26 | 0.2 | min$^{-1}$ | Arbitrary |
| GATOR1 (cyto) => GATOR1 (lyso) | k27 | 1 | min$^{-1}$ | Arbitrary |
| GATOR1 (lyso) => GATOR1 (cyto) | l27 | 1 | min$^{-1}$ | Arbitrary |
| mTORC1 (cyto) => mTORC1 (lyso) | k28 | 0.6 (AA(+)Ins(+)) 0.45 (AA(+)Ins(−)) 0.12 (AA(+)Ins(−)) | min$^{-1}$ | Arbitrary |
| mTORC1 (lyso) => mTORC1 (cyto) | l28 | 0.4 (AA(+)Ins(+)) 0.55 (AA(+)Ins(−)) 0.88 (AA(−)Ins(+)) | min$^{-1}$ | Arbitrary |

the AKT-TSC1/2-Rheb axis consisting of 25 Equations (1), (2), (3), (4), (5), (6), (7), (8), (9), (10), (11), (12), (13), (14), (15), (16), (17), (18), (19), (20), (21), (22), (23), (24), and (25). However, few studies have reported a model for the AA-sensing (Ragulator/LAMTOR-Rag) axis. Therefore, we constructed a mathematical model based on the standard manner of the reaction network system with nine Equations (26), (27), (28), (29), (30), (31), (32), (33), and (34). Given that lysosomal mTOR localization was primarily induced upon AA treatment (Fig 1E and F) and that complete mTORC1 activation required signaling induction of both the Ragulator/LAMTOR-Rag and AKT-TSC1/2-Rheb axes (Fig 1B–D), we concluded that the mTORC1 complex initially bound to Ragulator-Rag$^{GTP}$ on the lysosomal surface. Consequently, Ragulator-Rag$^{GTP}$–mediated lysosomal localization of mTORC1 enhanced the association of mTORC1 with Rheb$^{GTP}$, leading to mTORC1 activation on the lysosomal surface, with six Equations (35), (36), (37), (38), (39), and (40). A schematic illustration of the reactions involved in the spatiotemporal regulation of the mTORC1 pathway is shown in Fig 2A. The reactions after AA and Ins stimulation are represented by ordinary differential equations, with the parameters listed in Table 1. Cytoplasmic and endosomal IRs are indicated by the subscripts cyto and endo, respectively.

$$\frac{d[Ins]}{dt} = -k1[Ins][IR] + l1[Ins:IR] - k2[Ins][Ins:pIR] + l2[2Ins:pIR] + l3[Ins:pIR] \tag{1}$$

$$\frac{d[IR]}{dt} = l1[Ins:IR] + l3[Ins:pIR] - k1[Ins][IR] + l4[IR_{cyto}] - k4[IR] \tag{2}$$

$$\frac{d[Ins:IR]}{dt} = k1[Ins][IR] - l1[Ins:IR] - k3[Ins:IR] \tag{3}$$

$$\frac{d[2Ins:pIR]}{dt} = k2[Ins][Ins:pIR] - l2[2Ins:pIR] + l12[2Ins:pIR_{endo}] - k12[2Ins:pIR] \tag{4}$$

$$\frac{d[Ins:pIR]}{dt} = k3[Ins:IR] - k2[Ins][Ins:pIR] + l2[2Ins:pIR] - l3[Ins:pIR] - k12[Ins:pIR] + l12[Ins:pIR_{endo}] \tag{5}$$

$$\frac{d[IR_{cyto}]}{dt} = k5 - l5[IR_{cyto}] + k6([2Ins:pIR_{endo}] + [Ins:pIR_{endo}]) + k4[IR] - l4[IR_{cyto}] \tag{6}$$

$$\frac{d[2Ins:pIR_{endo}]}{dt} = k12[2Ins:pIR] - l12[2Ins:pIR_{endo}] - k6[2Ins:pIR_{endo}] \tag{7}$$

$$\frac{d[Ins:pIR_{endo}]}{dt} = k12[Ins:pIR] - l12[Ins:pIR_{endo}] - k6[Ins:pIR_{endo}] \tag{8}$$

$$\frac{d[IRS]}{dt} = l7[pIRS] - \frac{k7[IRS]([2Ins:pIR] + [Ins:pIR])}{8.97 \times 10^{-13}} \tag{9}$$

$$\frac{d[pIRS]}{dt} = \frac{k7[IRS]([2Ins:pIR] + [Ins:pIR])}{8.97 \times 10^{-13}} + l8[pIRS:PI3K] - l7[pIRS] - k8[pIRS][PI3K] \tag{10}$$

$$\frac{d[PI3K]}{dt} = l8[pIRS:PI3K] - k8[pIRS][PI3K] \tag{11}$$

$$\frac{d[pIRS:PI3K]}{dt} = k8[pIRS][PI3K] - l8[pIRS:PI3K] \tag{12}$$

$$\frac{d[PI(3,4,5)P3]}{dt} = k9[PI(4,5)P2] + k10[PI(3,4)P2] - l9[PI(3,4,5)P3] - l10[PI(3,4,5)P3] \tag{13}$$

$$k9 = \frac{1.259[pIRS1:PI3K]}{1 \times 10^{-13}} + 0.131 \tag{14}$$

$$\frac{d[PI(4,5)P2]}{dt} = l9[PI(3,4,5)P3] - k9[PI(4,5)P2] \tag{15}$$

$$\frac{d[PI(3,4)P2]}{dt} = l10[PI(3,4,5)P3] - k10[PI(3,4)P2] \tag{16}$$

$$\frac{d[PDK]}{dt} = l11[PI(3,4,5)P3:PDK] - k11[PDK] \tag{17}$$

$$\frac{d[PI(3,4,5)P3:PDK]}{dt} = k11[PDK] - l11[PI(3,4,5)P3:PDK] \tag{18}$$

$$k11 = \frac{0.6931([PI(3,4,5)P3] - 0.31)}{2.79} \tag{19}$$

$$\frac{d[AKT]}{dt} = l23[pAKT] - \frac{k23[PI(3,4,5)P3:PDK][AKT]}{0.4 \times 10^{-6} + [AKT]} \tag{20}$$

$$\frac{d[pAKT]}{dt} = \frac{k23[PI(3,4,5)P3:PDK][AKT]}{0.4 \times 10^{-6} + [AKT]} - l23[pAKT] \tag{21}$$

$$\frac{d[TSC]}{dt} = l13[pTSC] - \frac{k13[pAKT][TSC]}{10.3 \times 10^{-6} + [TSC]} \tag{22}$$

$$\frac{d[pTSC]}{dt} = \frac{k13[pAKT][TSC]}{10.3 \times 10^{-6} + [TSC]} - l13[pTSC] \tag{23}$$

$$\frac{d[Rheb^{GTP}]}{dt} = l14[Rheb^{GDP}] - \frac{k14[TSC][Rheb^{GTP}]}{0.3 \times 10^{-6} + [Rheb^{GTP}]} - k20[Ragulator:RagA^{GTP}:mTORC1][Rheb^{GTP}] + l20[Ragulator:RagA^{GTP}:mTORC1:Rheb^{GTP}] \tag{24}$$

$$\frac{d[Rheb^{GDP}]}{dt} = \frac{k14[TSC][Rheb^{GTP}]}{0.3 \times 10^{-6} + [Rheb^{GTP}]} - l14[Rheb^{GDP}] \tag{25}$$

$$\frac{d[AA]}{dt} = -k15[AA][SESN2] + l15[AA:SESN2] \tag{26}$$

**Table 2.  Two synthetic artificial protein sequences, mTOR_concat01 and mTOR_concat02.**

| Artificial protein | Sequences |
|---|---|
| mTOR_concat01 | VFEAEQFGCPQRE<br>LSEITTAEADP<br>VVPRFASGGCDNLIKDVAW<br>APSIGLPTSTIASCSQD<br>GRFLLLPACVTATKEMVYCL<br>EQGLIYRLLSVLGASEEDNV<br>KEAYDSLCTSGVVRAAAVA<br>LFNLDIREEEEASQEEMAS<br>RNGGQILIADLRDSVSTFSGQS<br>ESVRLPPDFFSVLVRELDQW<br>VEQLNECKSLQEACLSYVT<br>KEQVSISSFYYKTIDSVTSAQ<br>ELRELDQWIEQLNECKLSQA<br>PLLPSLLKDVNLVHSAFEVLR<br>SSLGMVESSRDEVAHTVTES<br>RFGSLTYSDPRFYGAEIVSA<br>LEYLHSRSQASGQLLAAES<br>RILDSVEDNYIVLNKVTMND<br>FDYLKHMGLMLATCSADGIV<br>RVSTALSGSALTGRVTMNEF<br>EYLKTLVTFSDERGYTISDS<br>APSRSNVLLSFDDTPEKVQIF<br>EYNENTRAIQILNEGASSEKLG<br>DIVEIAHPNDEYSPLLLQVKTA<br>AVWEEIVGESNDKSI<br>DEFPEINRFYGAEIVSALDYLH<br>SEKLQNLGELNKFFANIVWQD<br>VYEKFSLPVSLSEFRAAILNVP<br>AAQNPEDLRSDPDLE<br>TPWAR |
| mTOR_concat02 | SLQLEPSFVGNKTDEQALLS<br>SILAKGAMAATYSALNRETVG<br>FGMLKTVSNLILTGPRLDDQIF<br>LNRSIICYYNTYQVVQF<br>NRETIGVGSYSVCKALAESW<br>NAAFLESSAKE<br>LAPLFEELIKHINWEEL<br>LARNLVFSDGYVVKPLAQLA<br>DPWQKFFIPEGC<br>IQRIGPVAQDLLQRLYSLLSD<br>PIPEVRLSDGFVAAEAKLNLPPY<br>LTQEARGGELLEHIRWM<br>YTGGEDCTARLPPLPSLTS<br>QPHQVLASEPIPFSDLQQVSRL<br>WCVETGEIKLLEAPDVLCLRSLI<br>VAGLGDGSIRLSVVF<br>GEHTLLVTVSGQRHDGITVA<br>VHKLAASFQSMEVRSIIFANYIAR<br>VQIPIMLVGNKLVLPPYLTPDA<br>RIGGGLGDAADVQR<br>FYTVTEPQRFEISETSVNRG<br>FSFVATGLMEDDGKPRALN<br>GAEPNYHSLP<br>SARFILMDCM<br>EGRNQSPVLEPVGRNV<br>IDCQLSRMAEVL<br>VTGEQLRLFSESE<br>ASQILRVANDSAPEHALRP<br>GFLSTFALATDQGGSKAIHEVF<br>EVGVTSHRLIPQLPTLENLL<br>NIFISNSGIEKLFDAPEVPL<br>PSRANDDLADAGLEKVLGL<br>LGALDPYKNIASVGFHEDGRSS<br>QGDALASGPVETG<br>PMKVTAAIASNIWAA<br>YDRPSLQSEED<br>VSQFDARMDLVQEDQRWSEVEDSCDSDAVAR |

Two synthetic artificial protein sequences (mTOR_concat01; mTOR_concat02) derived from the proteotypic peptides of mTOR signaling components were used as internal standards for the precise and absolute quantification of proteins.

$$\frac{d[SESN2]}{dt} = -k15[AA][SESN2] + l15[AA:SESN2] - k16[SESN2][GATOR2]$$
$$+ l16[SESN2:GATOR2] \qquad (27)$$

$$\frac{d[AA:SESN2]}{dt} = k15[AA][SESN2] - l15[AA:SESN2] \qquad (28)$$

$$\frac{d[GATOR2]}{dt} = -k16[SESN2][GATOR2] + l16[SESN2:GATOR2]$$
$$- k17[GATOR2][GATOR1] + l17[GATOR2:GATOR1] \qquad (29)$$

$$\frac{d[SESN2:GATOR2]}{dt} = k16[SESN2][GATOR2] - l16[SESN2:GATOR2] \qquad (30)$$

$$\frac{d[GATOR1]}{dt} = -k17[GATOR2][GATOR1] + l17[GATOR2:GATOR1] \qquad (31)$$

$$\frac{d[GATOR2:GATOR1]}{dt} = k17[GATOR2][GATOR1] - l17[GATOR2:GATOR1] \qquad (32)$$

$$\frac{d[Ragulator:Rag^{GTP}]}{dt} = -\frac{k18[GATOR1][Ragulator:Rag^{GTP}]}{1 \times 10^{-7} + [Ragulator:Rag^{GTP}]}$$
$$+ l18[Ragulator:Rag^{GDP}] - k19[Ragulator:Rag^{GTP}][mTORC1]$$
$$+ l19[Ragulator:Rag^{GTP}:mTORC1] \qquad (33)$$

$$\frac{d[Ragulator:Rag^{GDP}]}{dt} = \frac{k18[GATOR1][Ragulator:Rag^{GTP}]}{1 \times 10^{-7} + [Ragulator:Rag^{GTP}]}$$
$$- l18[Ragulator:Rag^{GDP}] \qquad (34)$$

$$\frac{d[mTORC1]}{dt} = -k19[Ragulator:Rag^{GTP}]$$
$$[mTORC1] + l19[Ragulator:Rag^{GTP}:mTORC1] \qquad (35)$$

$$\frac{d[Ragulator:Rag^{GTP}:mTORC1]}{dt} = k19[Ragulator:Rag^{GTP}]$$
$$[mTORC1] - l19[Ragulator:Rag^{GTP}:mTORC1]$$
$$- k20[Ragulator:Rag^{GTP}:mTORC1][Rheb^{GTP}]$$
$$+ l20[Ragulator:Rag^{GTP}:mTORC1:Rheb^{GTP}] \qquad (36)$$

$$\frac{d[Ragulator:Rag^{GTP}:mTORC1:Rheb^{GTP}]}{dt} = k20[Ragulator$$
$$:Rag^{GTP}:mTORC1][Rheb^{GTP}]$$
$$- l20[Ragulator:Rag^{GTP}:mTORC1:Rheb^{GTP}]$$
$$- k21[Ragulator:Rag^{GTP}:mTORC1:Rheb^{GTP}]$$
$$+ l21[Ragulator:Rag^{GTP}:pmTORC1:Rheb^{GTP}] \qquad (37)$$

$$\frac{d[Ragulator:Rag^{GTP}:pmTORC1:Rheb^{GTP}]}{dt} = k21[Ragulator$$
$$:Rag^{GTP}:mTORC1:Rheb^{GTP}] - l21[Ragulator:Rag^{GTP}$$
$$:pmTORC1:Rheb^{GTP}] \qquad (38)$$

$$\frac{d[S6K]}{dt} = -\frac{k22[Ragulator:Rag^{GTP}:pmTORC1:Rheb^{GTP}][S6K]}{0.8 \times 10^{-6} + [S6K]}$$
$$+ \frac{l22 \times 0.15 \times 10^{-6}[pS6K]}{8.8 \times 10^{-6} + [pS6K]} \qquad (39)$$

**Table 3.  Initial concentrations of molecules in the mathematical model.**

| Species | Initial concentration | Unit | Comment |
|---|---|---|---|
| Insulin (Ins) | $6 \times 10^{-11}$ | M | Kolb et al (2020) |
| Insulin receptor (IR) in plasma membrane | $9 \times 10^{-13}$ | M | Sedaghat et al (2002) |
| IR in cytoplasm | $1 \times 10^{-13}$ | M | Sedaghat et al (2002) |
| IRS1 | $1 \times 10^{-12}$ | M | Sedaghat et al (2002) |
| PI3K | $1 \times 10^{-13}$ | M | Sedaghat et al (2002) |
| PI(4,5)P2 | $7 \times 10^{-6}$ | M | Jain & Bhalla (2009) |
| PDK | $1 \times 10^{-6}$ | M | Jain & Bhalla (2009) |
| TSC1/2 | $1 \times 10^{-6}$ in cytoplasm (wo/w-crosstalk model), $0.25 \times 10^{-6}$ in cytoplasm and $0.75 \times 10^{-6}$ in lysosome (compartment model) | M | Jain & Bhalla (2009) |
| Rheb | $1 \times 10^{-6}$ in cytoplasm (wo/w-crosstalk model), $1 \times 10^{-4}$ in lysosome (compartment model) | M | Jain & Bhalla (2009) |
| Amino acid (AA) | $6 \times 10^{-4}$ | M | Schmidt et al (2016) |
| SESN2 | $2.33 \times 10^{-7}$ (SESN1) | M | This study |
| GATOR2 | $1.63 \times 10^{-5}$ (SEC13) | M | This study |
|  | $6.25 \times 10^{-6}$ (SEH1) |  |  |
|  | $1.40 \times 10^{-7}$ (WDR24) |  |  |
|  | $1.53 \times 10^{-7}$ (WDR59) |  |  |
|  | $7.56 \times 10^{-7}$ (MIO) |  |  |
| GATOR1 | $2.50 \times 10^{-7}$ in cytoplasm (DEPDC5) | M | This study |
|  | $1.57 \times 10^{-6}$ in cytoplasm (NPRL2) |  |  |
|  | $1.93 \times 10^{-7}$ in cytoplasm (NPRL3) (wo/w-crosstalk model), $0.93 \times 10^{-7}$ in cytoplasm and $1.0 \times 10^{-7}$ in lysosome (NPRL3) (compartment model) |  |  |
| Ragulator:RagA-GTP | $1.25 \times 10^{-5}$ in cytoplasm (p18) | M | This study |
|  | $1.95 \times 10^{-5}$ in cytoplasm (p14) |  |  |
|  | $1.02 \times 10^{-5}$ in cytoplasm (MP1) (wo/w-crosstalk model), $1.02 \times 10^{-3}$ in cytoplasm (MP1) (compartment model) |  |  |
|  | $1.62 \times 10^{-5}$ in cytoplasm (p10) |  |  |
|  | $1.19 \times 10^{-5}$ in cytoplasm (HBXIP), $6.23 \times 10^{-6}$ in cytoplasm (RagA), $4.67 \times 10^{-6}$ in cytoplasm (RagC) |  |  |
| mTORC1 | $1.91 \times 10^{-6}$ in cytoplasm (mTOR) (wo/w-crosstalk model) (compartment model) | M | This study |
|  | $2.17 \times 10^{-6}$ in cytoplasm (RAPTOR) |  |  |
|  | $3.52 \times 10^{-6}$ in cytoplasm (mLST8), $1.90 \times 10^{-7}$ in cytoplasm (DEPTOR) |  |  |
| S6K | $5.39 \times 10^{-6}$ in cytoplasm (RPS6KA1) | M | This study |
|  | $4.53 \times 10^{-6}$ in cytoplasm (RPS6KA3) |  |  |
|  | $2.29 \times 10^{-6}$ in cytoplasm (RPS6KA4) |  |  |
|  | $2.88 \times 10^{-6}$ in cytoplasm (RPS6KB1) |  |  |
|  | $1.27 \times 10^{-6}$ in cytoplasm (RPS6KB2) |  |  |
|  | $8.24 \times 10^{-7}$ in cytoplasm (RPS6KC1) (wo/w-crosstalk model), $8.24 \times 10^{-5}$ in lysosome (RPS6KC1) (compartment model) |  |  |
| AKT | $2 \times 10^{-7}$ | M | Jain & Bhalla (2009) |
| PP2A | $0.15 \times 10^{-6}$ in cytoplasm (w-crosstalk model), $0.05 \times 10^{-6}$ in cytoplasm and $1.0 \times 10^{-6}$ in lysosome (compartment model) | M | Jain & Bhalla (2009) |

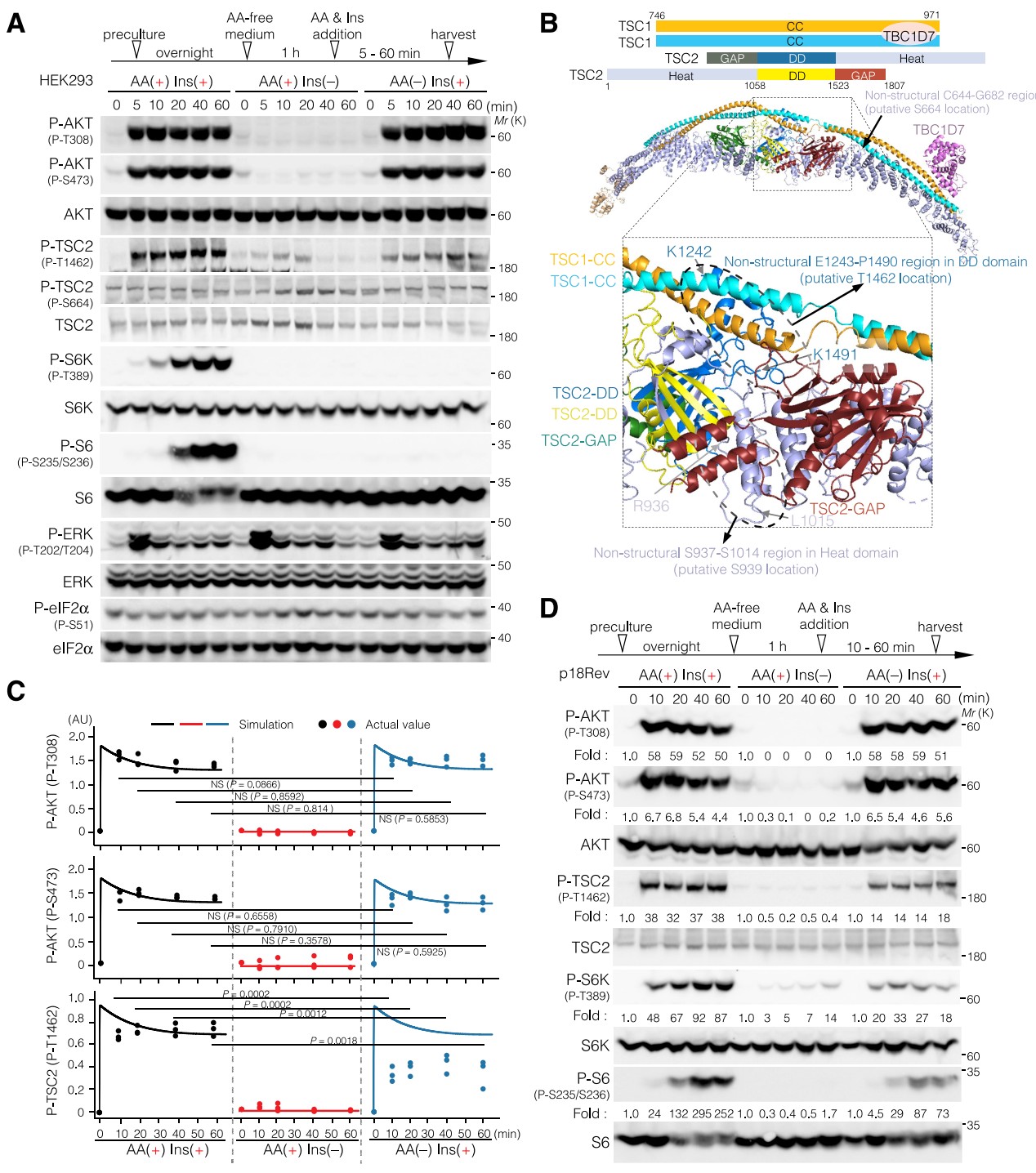

**Figure 3. Ins-induced phosphorylation of TSC2 at T1462 is substantially inhibited under the AA-free conditions.**
**(A)** Phosphorylation levels of AKT at T308 and S473, TSC2 at T1462 and S664, S6K at T389, S6 at S235 and S236, ERK at T202 and Y204, and eIF2α at S51 monitored in HEK293 cells exposed to AA deprivation and subsequent AA and Ins treatment. Cells were cultured in minimal medium without AA and Ins for 1 h, then incubated with medium containing AA and Ins (10 μg/ml) alone or in combination for the indicated times. Cell lysates were analyzed by Western blotting using the indicated antibodies. **(B)** Cryo-EM structure of the human TSC1-TSC2-TBC1D7 complex (PDB ID: 7DL2) with a 2:2:1 stoichiometry of TSC1, TSC2, and TBC1D7. The two pairs of TSC1 coiled coils (746–971 aa) are shown in cyan and orange. TSC2 (1–1,807 aa) comprised a heat repeat domain (Heat) (1–1,058 aa), dimerization domain (DD) (1,059–1,523 aa), and GTPase-activating domains (GAP) (1,524–1,807 aa). Heat, light purple; DD, yellow and blue; GAP, dark red and green; TBC1D7 (21–287 aa), purple. The area in the small square in the upper panel is enlarged in the lower panel. Here, the dotted lines indicate the predicted location of the T1462 and S939 residues in TSC2. The T1462 residue is putatively positioned at or near the TSC1-CC– and TSC2-DD–associated interfaces, whereas the S939 residue is putatively positioned on the opposite side of the TSC1-CC– and TSC2-associated interfaces. **(C)** Comparison of the phosphorylation levels of AKT at T308 and S473 and TSC2 at T1462 between the simulation and experimental values. Red, blue, and black lines indicate the simulation values upon treatment with AA [AA(+) Ins (−)] and Ins [AA(−) Ins (+)] alone or in combination [AA(+) Ins (+)], respectively. Black [AA(+) Ins (+)], red [AA(+) Ins(−)], and blue [AA(−) Ins(+)] dots indicate experimental values. P-values were assessed using one-way ANOVA with

$$\frac{d[pS6K]}{dt} = \frac{k22\left[Ragulator:Rag^{GTP}:pmTORC1:Rheb^{GTP}\right][S6K]}{0.8 \times 10^{-6} + [S6K]}$$
$$- \frac{l22 \times 0.15 \times 10^{-6}[pS6K]}{8.8 \times 10^{-6} + [pS6K]} \qquad (40)$$

Quantitative data on intracellular protein concentrations are required to accurately simulate the dynamics of mTORC1 signal transduction. However, little is known on intracellular SESN, GATOR2 (SEC13, SEH1, WDR24, WDR59, and MIO), GATOR1 (DEPDC5, NPRL2, and NPRL3), Ragulator/LAMTOR-RagA/C (p18, p14, MP1, p10, HBXIP, RagA, and RagC), and mTORC1 (mTOR, RAPTOR, mLST8, and DEPTOR) concentrations. Therefore, we conducted absolute quantification of the intracellular mTORC1 signaling proteins using in vitro proteome-assisted multiple reaction monitoring for protein absolute quantification (iMPAQT) (Matsumoto et al, 2017). Two synthetic artificial protein sequences (mTOR_concat01 and mTOR_concat02), derived from the proteotypic peptides of mTOR signaling components, were used (Table 2). Assuming that the total proteins and volume per cell were 169 pg/cell (338 $\mu$g/2 × 10$^6$ cells) and 3.4 × 10$^{-12}$ L/cell (Aoki et al, 2011), respectively, the intracellular concentrations of mTORC1 signaling components were calculated and are summarized in Table 3 and Supplemental Data 1 (Sedaghat et al, 2002; Jain & Bhalla, 2009; Schmidt et al, 2016; Kolb et al, 2020). The obtained concentrations were used as the initial values for mathematical model simulation in this study. The mathematical simulation showed that after simultaneous AA and Ins treatment, the simulated phosphorylation levels of mTOR at S2481, S6K at T389, AKT at T308, and TSC2 at T1462 closely matched the experimental values obtained from reverse-phase protein array (RPPA) and Western blotting (Figs 2B and S2A–D). These results indicate that our mathematical model successfully describes the temporal regulation of mTORC1 signal transduction.

## Mathematical model of the mTORC1 pathway without crosstalk

To examine the effects of AA and Ins on mTORC1 signal transduction, the time courses of phosphorylated AKT (P-T308 and P-S473), TSC2 (P-T1462 and P-S664), S6K (P-T389), S6 (P-S235 and P-S236), ERK (P-T202 and P-Y204), and eIF2$\alpha$ (P-S51) were monitored in HEK293 cells subjected to AA starvation and subsequent AA, Ins, or AA plus Ins stimulation (Figs 3A and S3A). Among cells treated with AA and Ins, either alone or in combination, the levels of P-TSC2 (P-S664; an ERK phosphorylation site), P-ERK (P-T202 and P-Y204; active phosphorylation sites of ERK), and P-eIF2$\alpha$ (P-S51; a phosphorylation site mediated by GCN2 upon AA deprivation) were comparable. In contrast, the T308 and S473 residues of AKT were rapidly phosphorylated upon Ins treatment, and their phosphorylation levels peaked ~5 min after stimulation (Figs 3A and S3A). Similarly, the T1462 residue in TSC2, a major AKT phosphorylation site putatively located at or near the interface between TSC1-CC and TSC2-DD (PDB ID: 7DL2) (Yang et al,

2021) (Fig 3B), was also rapidly phosphorylated after Ins stimulation. TSC2 phosphorylation also peaked ~5 min after Ins stimulation (Figs 3A and S3A). Note that the T1462 residue is putatively positioned at or around the TSC1-CC– and TSC2-DD–associated interfaces, whereas the S939 residue is putatively positioned on the opposite side from the TSC1-CC– and TSC2-associated interfaces (Fig 3B).

Furthermore, relatively slow phosphorylation levels of S6K (downstream of the AKT-TSC1/2 axis) at T389 and S6 (an S6K target) at S235 and S236 were observed 20 min or later after AA plus Ins stimulation (Figs 3A and S1F, S2A–D, and S3A), whereas S6K and S6 phosphorylation occurred to a lesser extent upon Ins or AA stimulation alone in HEK293 cells (Figs 3A and S3A). This indicates that complete mTORC1 activation requires signal induction of both the AA- and GF-sensing axes. Next, we mathematically simulated temporal changes in the phosphorylation levels of AKT (P-T308 and P-S473) and TSC2 (P-T1462) through stimulation with AA and Ins alone or in combination and compared the changes with the experimental values. As expected, the simulated values of AKT phosphorylation at T308 and S473 upon Ins stimulation were similar to the obtained experimental values (Fig 3A and C). In addition, both AKT residues were robustly phosphorylated upon Ins stimulation, regardless of AA addition (Fig 3A and C). Immunoblotting revealed similar results in p18Rev, parental (control) MEFs, and HeLa cells (Figs 3D and S3B). These data indicate that AKT phosphorylation depended only on Ins stimulation and was independent of intracellular AA levels. Theoretically, after Ins stimulation, similar to AKT phosphorylation, AKT-mediated TSC2 phosphorylation at T1462 was expected to be robust despite intracellular AA levels (Fig 3C). However, upon Ins stimulation TSC2 phosphorylation at T1462 during AA removal was substantially lower than that observed in the presence of AA (Figs 3A, C, and D and S3A and B). These data strongly suggest that Ins-induced TSC2 phosphorylation at T1462 is suppressed upon AA starvation through a mechanism different from that underlying AKT phosphorylation.

## Model with crosstalk more accurately reflects mTORC1 changes

TSC2 phosphorylation was assumed to be selectively dephosphorylated by phosphatase(s) that were blunted by AA addition (Fig 4A), given that Ins-induced TSC2 phosphorylation at T1462 was attenuated only upon AA removal, despite similar activatory phosphorylation levels of T308 and S473 in AKT upon Ins stimulation, irrespective of the cellular AA levels. To test this hypothesis using our mathematical simulations, we defined the reactions involving AA-sensitive phosphatase (PPase) that dephosphorylates TSC2 at the T1462 residue (from [pTSC] to [TSC]) and added the following five Equations (41), (42), (43), (44), and (45) to the mTORC1 mathematical model (hereafter, mTORC1 model with crosstalk).

Tukey's multiple comparison test. NS, not significant; AU, arbitrary units. **(D)** Phosphorylation levels of AKT at T308 and S473, TSC2 at T1462, S6K at T389, and S6 at S235 and S236 monitored in p18Rev cells exposed to AA deprivation and subsequently treated with AA or Ins alone and in combination, as in (A). Cell lysates were analyzed by Western blotting using the indicated antibodies.

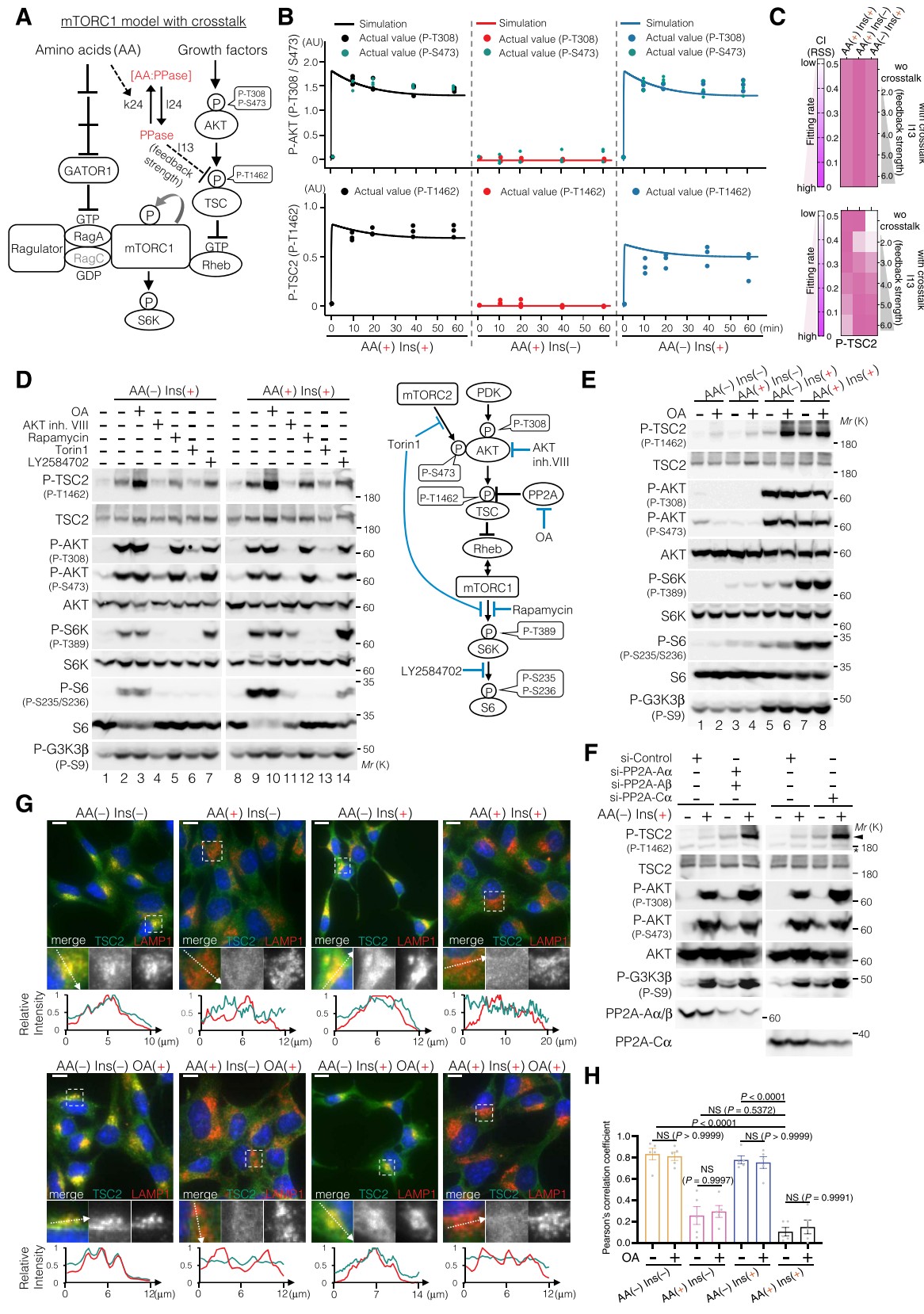

$$\frac{d[TSC]}{dt} = \frac{l13[PPase][pTSC]}{8.8 \times 10^{-6} + [pTSC]} - \frac{k13[pAKT][TSC]}{10.3 \times 10^{-6} + [TSC]} \quad (41)$$

$$\frac{d[pTSC]}{dt} = \frac{k13[pAKT][TSC]}{10.3 \times 10^{-6} + [TSC]} - \frac{l13[PPase][pTSC]}{8.8 \times 10^{-6} + [pTSC]} \quad (42)$$

$$\frac{d[AA]}{dt} = -k15[AA][SESN2] + l15[AA:SESN2] - k24[AA][PPase] + l24[AA:PPase] \quad (43)$$

$$\frac{d[PPase]}{dt} = -k24[AA][PPase] + l24[AA:PPase] \quad (44)$$

$$\frac{d[AA:PPase]}{dt} = k24[AA][PPase] - l24[AA:PPase] \quad (45)$$

Mathematical simulations revealed that compared to the model without crosstalk (Fig 3C), the simulation values of P-AKT (P-T308 and P-S473) and P-TSC2 (P-T1462) obtained from the model with crosstalk fit relatively well (better) with the experimental values (Fig 4B and C). Consistently, Western blot analysis using various chemical inhibitors showed that TSC2 phosphorylation at T1462 post-Ins stimulation was markedly increased upon stimulation with both Ins and okadaic acid (OA), an inhibitor of phosphatases such as PP2A and PP1 (Medina et al, 2013) in both p18Rev (Fig 4D and E) and HEK293 cells (Fig S4A). TSC2 phosphorylation was also restored upon knockdown of PP2A scaffold A subunit Aα and Aβ (PP2A-Aα/β) or catalytic C subunit Cα (PP2A-Cα) (Fig 4F). Consistently, TSC2 interactions with the catalytic C subunits (Cα and Cβ) and regulatory B subunits (e.g., B55δ, B56β, and B56δ) of PP2A were detected using an AlphaScreen assay (Fig S4B) and CoIP assay (Fig S4C–E). Furthermore, TSC2 phosphorylation levels at T1462 correlated with AKT activation––its inhibition by AKT inhibitor VIII (also known as Akti-1/2) or Torin1 drastically inhibited TSC2 phosphorylation after Ins stimulation for 20 min (Fig 4D). AKT inhibitor VIII prevents AKT1/2 association with the cell membrane by binding to the PH domain of AKT1/2 isoenzymes (Wu et al, 2010), whereas Torin1 inhibits both mTORC1 and mTORC2 (Wang et al, 2022), the latter of which is responsible for AKT phosphorylation at S473. In contrast, the blockade of mTORC1 or S6K alone by rapamycin (Gaubitz et al, 2015) or LY2584702 (Ebner et al, 2023) did not interrupt Ins-induced TSC2 phosphorylation (Fig 4D). Although PP2A has been reported to dephosphorylate AKT at T308 and S473 (Andjelković et al, 1996; Ugi et al, 2004; Kuo et al, 2008), Ins treatment for up to 60 min induced comparable levels of AKT phosphorylation at T308 and S473 irrespective of OA addition (Figs 4D and S4A). Moreover, despite PP2A-mediated S6K

dephosphorylation at T389 (Hahn et al, 2010), OA treatment led to only a marginal increase in S6K phosphorylation at T389 upon Ins stimulation for 20 min (Figs 4D and S4A), indicating that active S6K slightly increased upon simultaneous OA and Ins stimulation but did not enhance TSC2 phosphorylation at T1462 via feedback regulation of the Ins-sensing (AKT-TSC-Rheb) pathway (Rozengurt et al, 2014) under experimental conditions. Furthermore, immunofluorescence using a TSC2-specific antibody (Fig S1A–D) showed that TSC2 diffused into the cytoplasm during AA treatment but preferentially localized to the lysosomes after AA starvation (Figs 4G and H and S4F), which is consistent with previous findings (Demetriades et al, 2014; Carroll et al, 2016). In contrast, AKT was constantly localized in the cytoplasm and the nucleus, irrespective of intracellular AA or Ins levels (Fig S4G–I), whereas mTOR diffused to the cytoplasm upon AA removal and displayed lysosomal localization after the addition of AAs (Figs 1E and F and S4J). These data strongly suggest that upon Ins treatment, TSC2 phosphorylation at T1462 is cooperatively regulated by the balance between kinase (AKT) and phosphatase (PP2A) along with lysosomal localization of TSC2 in response to AA removal.

## Comprehensive identification of proteins on lysosomal surface using a proximity-labeling method Lyso-BioID

To comprehensively monitor proteins localized to lysosomes upon AA supplementation and AA removal, we employed a proximity-dependent biotin identification (BioID) method (Fujikawa et al, 2023). LAMP1, a lysosomal membrane protein, was fused to the N terminus of AGIA-tagged AirID, a proximity biotinylation enzyme (Kido et al, 2020) (Fig 5A). LAMP1-AGIA-AirID stably expressing HEK293 cells were treated with biotin under AA-supplemented (growth medium [GM] containing AA) and AA-removal (AA(–)) conditions. LAMP1-AGIA-AirID was constantly localized on the lysosomal surface, regardless of the AA-supplemented or AA-removal condition (Fig 5B). After biotin treatment, LAMP1-AGIA-AirID biotinylated proteins on the lysosomal surface, as monitored by fluorescent staining of cells with streptavidin–Alexa Fluor 647 (Fig 5B) and by Western blot analyses of cell extracts with streptavidin–HRP (Fig 5C). Based on liquid chromatography–tandem mass spectrometry (LC-MS/MS) analysis

**Figure 4. TSC2 phosphorylation at T1462 is susceptible to PP2A in the AA-free condition.**
**(A)** Schematic representation of the reactions involved in the mTORC1 pathway with crosstalk. Reactions depicting the AA-sensitive phosphatase (PPase) that dephosphorylates TSC2 at the T1462 residue (from pTSC to TSC) were defined and added to the mTORC1 mathematical model without crosstalk. l13 is defined as a rate constant for feedback strengths. **(B)** Simulation of phospho-AKT at T308 and S473 and phospho-TSC2 at T1462 in the mTORC1 model with crosstalk. The dots indicate the experimental values. **(C)** The correlations between the simulation and actual values of P-AKT(T308) (upper) and P-TSC2 (T1462) (lower) were evaluated by the correlation indexes (CI) based on the residual sum of squares (RSS). **(C)** shows fits for different values of the parameter l13 (feedback strength). A smaller CI (RSS) indicates a higher fitting rate of the model to the actual data. **(D)** Phosphorylation levels of TSC2 at T1462, AKT at T308 and S473, S6K at T389, S6 at S235 and S236, and GSK3β at S9 monitored in p18Rev cells subjected to AA starvation followed by Ins (left) or Ins plus AA treatment (right) for 20 min, or together with okadaic acids (OA), AKT inhibitor VIII, rapamycin, Torin1, or LY2584702 (an S6K inhibitor). Note that the anti-P-S6 antibodies added during the first immunoblotting cover the antigen S6 protein, and therefore, the newly added anti-S6 antibodies are repelled when the same membrane is reblotted. **(E)** Phosphorylation levels of TSC2 at T1462, AKT at T308 and S473, S6K at T389, S6 at S235 and S236, and GSK3β at S9 monitored in p18Rev cells subjected to AA starvation followed by treatment with AA, Ins, or Ins plus AA for 15 min, or together with OA. **(F)** PP2A-Aα/β or PP2A-Cα knockdown enhanced Ins-induced TSC2 (T1462) phosphorylation. HEK293 cells were transfected with a control siRNA or siRNAs specific for PP2A-Aα, PP2A-Aβ, and PP2A-Cα, and subjected to AA starvation, followed by Ins treatment alone for 20 min. Cell lysates were analyzed via immunoblotting. **(G)** Representative immunofluorescence data for TSC2 in p18Rev cells subjected to AA starvation and subsequently treated, as indicated, with AA, Ins, and OA for 15 min. The area in the small squares is enlarged. TSC2, green; LAMP1, red; DAPI, blue. Scale bars, 10 $\mu$m. Line scans of TSC2 (green) and LAMP1 (red) are shown at the bottom of the figure. **(G, H)** TSC2-LAMP1 colocalization in (G) was measured based on Pearson's correlation coefficient. The mean values of Pearson's correlation coefficient from five cells and the SEM are shown. P-values were assessed using one-way ANOVA with Tukey's multiple comparison test. NS, not significant. Source data are available for this figure.

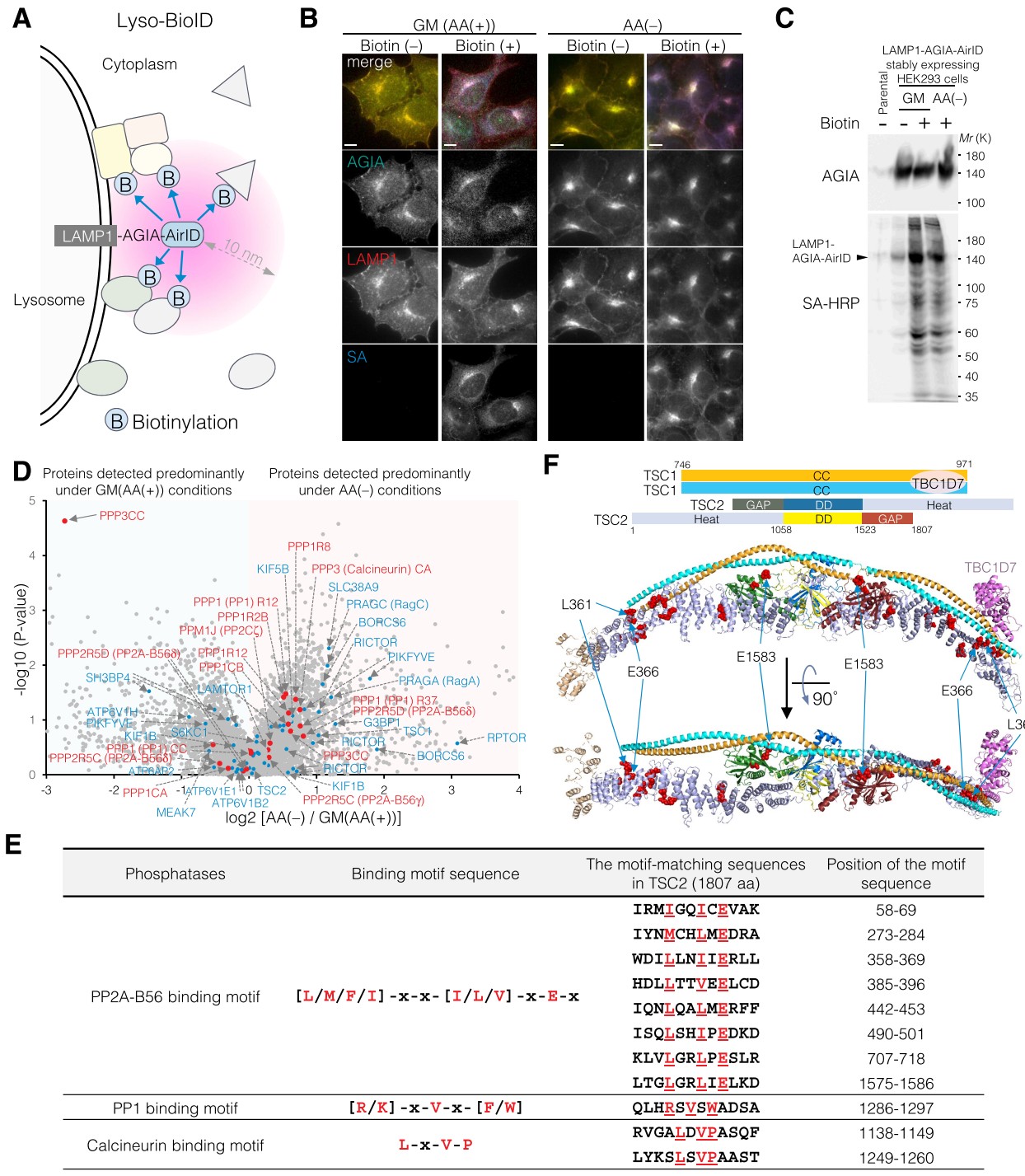

**Figure 5. Comprehensive identification of proteins on the lysosomal surface using a proximity-labeling method (Lyso-BioID).**
**(A)** Schematic representation of proximity-dependent biotin identification of lysosomal surface proteins (Lyso-BioID) using LAMP1-AGIA-AirID. **(B)** Representative immunofluorescence images of LAMP1-AGIA-AirID stably expressing HEK293 cells treated with growth medium (GM) (AA(+)) or AA-free medium (AA(-)), alone or in combination with 50 µM biotin for 10 h. AGIA, green; LAMP1, red; streptavidin (SA), blue. Scale bars, 10 µm. **(C)** LAMP1-AGIA-AirID stably expressing HEK293 cells were cultured in GM or AA(-) medium together with 50 µM biotin for 10 h. Protein biotinylation in LAMP1-AGIA-AirID stably expressing HEK293 cells was confirmed by immunoblotting using an anti-AGIA antibody and SA-HRP. Note that LAMP1 (417 aa, ~45 kD) is heavily glycosylated at the N-terminal luminal side of the lysosome inner leaflet, providing protection from proteolysis. Therefore, immunoblotting bands for (glycosylated-)LAMP1 appear at higher molecular weight (~100 kD) than expected. **(D)** Biotinylated peptides derived from LAMP1-AGIA-AirID stably expressing HEK293 cells cultured in GM or AA(-) medium were detected by LC-MS/MS. Blue dots indicate proteins known to localize on the lysosomal surface. Red dots represent Ser/Thr phosphatases such as PP2A, PP1, calcineurin, and PP2C. **(E)** Table listing the phosphatase-binding motif sequences and their positions in TSC2 (1,807 aa). TSC2 contains eight PP2A-B56 binding motifs (**[L/M/F/I]-x-x-[I/L/V]-x-E-x**), one PP1-binding motif ([R/K]-x-V-x-[F/W]), and two calcineurin-binding motifs (L-x-V-P) in TSC2. **(F)** Cryo-EM structure of the human TSC1-TSC2-TBC1D7 complex (PDB ID: 7DL2), showing a 2:2:1 stoichiometry of TSC1, TSC2, and TBC1D7, as depicted in Fig 3B. L, M, I, V, and E residues in the PP2A-B56–binding motif sites (273–284; 358–369; 385–396; 442–453;

for biotinylated proteins, we identified a total of 5,865 biotinylated peptides (Supplemental Data 2). Of these, we identified already known lysosomal proteins (e.g., p18/LAMTOR1, Rag [PRAGA and PRAGC], v-ATPase [ATP6V1E1, ATP6V1B2, ATP6V1H, and ATP6AP2], SLC38A9, S6K [S6KB1 and S6KC1], TSC1, PIKFYVE, RAB7A, RAB9A, SH3BP4, MEAK7, G3BP1, RPTOR, RICTOR, KIF1B, KIF5B, BORC complex [BORCS6 and BORCS7]) (Harrison et al, 2003; Johansson et al, 2005; Nada et al, 2009; Kim et al, 2012; Pu et al, 2015; Rebsamen et al, 2015; Wang et al, 2015; Nguyen et al, 2018; Jia & Bonifacino, 2019; Fitzian et al, 2021; Chen et al, 2022; Figlia et al, 2022; Bussi et al, 2023; Xu et al, 2023), confirming that this proteomic analysis successfully monitored lysosomal surface proteins (Fig 5D and Supplemental Data 2). In addition, although AKT was not detected in this proteomic analysis, a number of Ser/Thr phosphatases, including PP2A (PPP2R5C [B56γ] and PPP2R5D [B56δ]), PP1 (PPP1CB, PPP1CC, PPP1R12, PPP1R2B, and PPP1R37), calcineurin (PPP3CA and PPP3CC), and PPM1J (PP2Cζ), were identified as proteins localized to the lysosomal surface (Fig 5D and Supplemental Data 2). Because PP2A (B56)-, PP1-, or calcineurin-binding motif sequences have been well studied, we next examined whether these phosphatase-interacting motifs are conserved in TSC2 (Fig 5E). One of the PP1-binding motifs ([R/K]-x-V-x-[F/W]) (Hendrickx et al, 2009) in residues 1,286–1,297 and two of the calcineurin-binding motifs (L-x-V-P) (Sheftic et al, 2016) in residues 1,138–1,149 and 1,249–1,260 are located in TSC2 (Fig 5E). In contrast, for the PP2A-B56–binding motif ([L/M/F/I]-x-x-[I/L/V]-x-E-x) (Hertz et al, 2016), at least eight sequences (e.g., 58–69, 273–284, 358–369, 385–396, 442–453, 490–501, 707–718, and 1,575–1,586) are conserved within TSC2 (Fig 5E). Furthermore, structural analysis of the human TSC1-TSC2-TBC1D7 complex (PDB ID: 7DL2) showed that although structural information of PP2A-B56 (58–69), PP1 (1,286–1,297), and calcineurin (1,138–1,149 and 1,249–1,260) motifs is not resolved, seven of the eight PP2A-B56–binding motif sequences (273–284, 358–369, 385–396, 442–453, 490–501, 707–718, and 1,575–1,586) are located on the surface of the TSC2 structure (Fig 5F). These data are consistent with the results obtained from the AlphaScreen assay and CoIP experiments, showing that PP2A preferentially binds to TSC2 (Fig S4B, D, and E), and suggest why TSC2 is sensitive to OA treatment and PP2A knockdown (Fig 4D–F). Furthermore, using the Catalogue Of Somatic Mutations In Cancer (COSMIC) database (https://cancer.sanger.ac.uk/cosmic), we found one somatic mutation each in the PP1-binding motif (R1289S) and the calcineurin-binding motif (L1142P), and also two somatic TSC2(E/K) mutations of the PP2A-B56–binding motif ([L/M/F/I]-x-x-[I/L/V]-x-E-x) in three clinical cases: E366K in rectal adenocarcinoma (https://cancer.sanger.ac.uk/cosmic/mutation/overview?id=102914022) and ovarian carcinoma (Cheasley et al, 2019), and E1583K in skin carcinoma (Starrett et al, 2020) (Fig 5F and Supplemental Data 3). These

data suggest that these E366 and E1583 residues are particularly required for PP2A-mediated dephosphorylation of TSC2.

## Lysosomal TSC2 localization is required for T1462 dephosphorylation

Because locking TSC2 to lysosomes using LAMP1-Flag-TSC2 resulted in lower phosphorylation levels of T1462 residues upon Ins stimulation compared with endogenous TSC2, regardless of the presence or absence of AAs (Fig S5A–C), we next examined the effect of lysosomal TSC2 localization on T1462 phosphorylation after Ins stimulation under AA-supplemented or AA-free conditions. Immunofluorescence analysis of cells expressing or depleted of p18/LAMTOR1 (Nada et al, 2009), a lysosomal membrane-anchored scaffold protein of the Ragulator/LAMTOR-Rag complex (Fig S6) (Yonehara et al, 2017), was performed. In p18Rev (p18 expressing) cells, similar to HEK293 cells (Fig S4F), under AA-supplemented conditions, TSC2 displayed diffuse cytoplasmic localization, whereas under AA deprivation, it localized to the lysosomal surface independent of Ins stimulation (Fig 6A). Lysosomal TSC2 localization during AA starvation was primarily mediated by the Ragulator/LAMTOR-Rag GTPase, as the genetic depletion of p18 led to a marked decrease in TSC2 recruitment to the lysosomal membrane (Fig 6B and C). In addition, immunofluorescence showed that although the Ragulator/LAMTOR-Rag–mediated lysosomal localization of TSC2 correlated with decreased intracellular AA levels, a part of PP2A-Aα/β and PP2A-Cα constantly displayed lysosomal localization, regardless of intracellular AA or Ins levels (Figs 6A and D and S7A–C). This is consistent with MS data, which indicated that at least the PP2A regulatory subunits B56γ and B56δ are detected on the lysosomal surface under AA-supplemented and AA-removal conditions (Fig 5D and Supplemental Data 2). The lysosomal localization of PP2A-Aα/β and PP2A-Cα was mediated by a Ragulator/LAMTOR-Rag–independent mechanism as it was not inhibited by p18-KO (Figs 6B and D and S7D–F). Furthermore, Western blotting showed that similar to HEK293 cells (Figs 3A and S4A), TSC2 phosphorylation remained low in Ins-stimulated p18Rev cells during AA removal compared with simultaneous Ins and AA treatment, although no significant changes were observed in AKT phosphorylation levels at T308 (Figs 6E–G and S8). Ins-stimulated p18KO cells under AA-deprived conditions restored TSC2 phosphorylation to the level observed upon simultaneous Ins and AA stimulation of p18Rev cells (Figs 6E and F and S8). Our findings indicated that lysosomal TSC2 localization occurs in an AA concentration–dependent manner (Demetriades et al, 2014; Carroll et al, 2016). However, Ins-dependent release of TSC2 from the lysosomal surface during serum starvation has also been reported (Menon et al, 2014), although this was observed under prolonged (16-h) serum starvation, which is considerably longer than the serum and AA starvation conditions (1 h) used

---

490–501; 707–718; 1,575–1,586) of TSC2 in (E) are shown as red spheres. The PP2A-B56–binding motif (58–69), PP1-binding motif (1,286–1,297), and calcineurin-binding motifs (1,138–1,149 and 1,249–1,260) are not resolved because of missing structural information. The L361 residue mutated to Pro (L361P), reported in tuberous sclerosis, and the E366 and E1583 residues mutated to Lys (E366K and E1583K) in tumors are indicated by light blue arrows.
Source data are available for this figure.

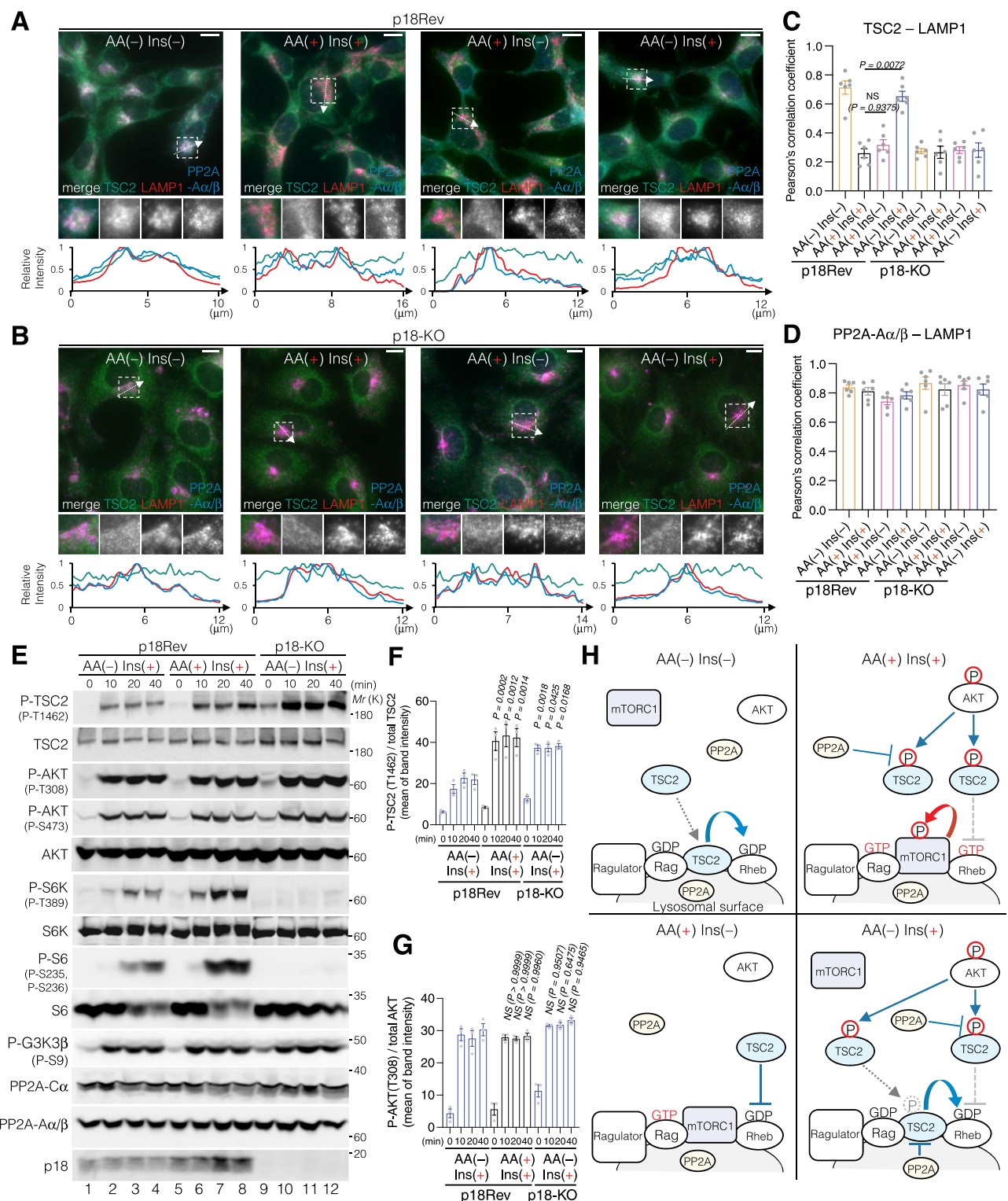

**Figure 6. Lysosomal localization of TSC2 during AA removal is required for its selective dephosphorylation at T1462.**

**(A, B)** Representative immunofluorescence images of p18Rev (A) and p18-KO (B) cells treated with AA or Ins, alone or in combination, for 15 min. The areas in the small squares are enlarged and shown in the lower panels. TSC2, green; LAMP1, red; PP2A-Aα/β, blue. Scale bars, 10 μm. Line scans of TSC2 (green), LAMP1 (red), and PP2A-Aα/β (blue) are shown at the bottom of the figure. **(A, B, C, D)** TSC2-LAMP1 (C) and PP2A-Aα/β-LAMP1 (D) colocalizations in (A, B) were measured based on Pearson's correlation coefficient. The mean values of Pearson's correlation coefficient from six cells and the SEM are shown. *P*-values were assessed using one-way ANOVA with Tukey's multiple comparison test. NS, not significant. **(E)** Phosphorylation levels of TSC2 at T1462, AKT at T308 and S473, S6K at T389, S6 at S235 and S236, and GSK3β at S9 monitored in p18Rev and p18-KO cells exposed to AA starvation and subsequently treated with Ins, alone or in combination with AA. Cell lysates were analyzed by

in our study. We performed experiments to test both possibilities. AA- and Ins-dependent TSC2 release from lysosomes are mediated by Ragulator/LAMTOR-Rag (Figs 6A–C and S7C–E) (Demetriades et al, 2014) and (Farnesylated) Rheb (Menon et al, 2014), respectively. Therefore, we used p18/Rheb-double KO (dKO) (Nada & Okada, 2020) and its derivative cells re-expressing p18 and Rheb alone or in combination to determine whether Rheb, together with Ragulator/LAMTOR-Rag, is involved in the lysosomal localization of TSC2 upon AA removal. Immunoblotting showed that the reintroduction of both p18 and Rheb in p18/Rheb-dKO cells restored low levels of Ins-stimulated TSC2 phosphorylation during AA starvation, whereas the reintroduction of either p18 or Rheb alone did not rescue this phenotype (Fig S9A and B). These data suggest that Ragulator/LAMTOR-Rag, at least partially, cooperates with Rheb on the lysosomal surface to regulate TSC2 localization during AA deprivation. Collectively, these data indicate that upon AA removal, a certain percentage of TSC2 is selectively transported to the lysosomal surface and dephosphorylated by PP2A. The dephosphorylated TSC2 with high GAP activity induces the conversion of active Rheb[GTP] to inactive Rheb[GDP] on the lysosomal surface, thereby lowering the level of the activatory phosphorylation of mTORC1 upon Ins stimulation during AA deprivation (Fig 6H).

### A compartment model describing cytoplasm–lysosome distribution of TSC2 and mTORC1, as well as lysosomal TSC2 dephosphorylation, reproduces the temporal changes of P-AKT, P-TSC2, and P-S6K

Based on all the findings from experiments, such as cytoplasm–lysosome distribution of TSC2 and mTORC1 depending on the AA concentration, and TSC2 dephosphorylation on the lysosomal surface, we developed a compartment model of the mTORC1 pathway (Fig 7A). The cytoplasm-to-lysosome (import) and lysosome-to-cytoplasm (export) transporting rates of TSC, GATOR1, and mTORC1 were defined as k25, k27, k28, l25, l27, and l28 (Fig 7A). The dephosphorylation rate of lysosomal TSC2 (TSC2_Lyso) was then defined as l26 (Fig 7A). Mathematical simulations revealed that the temporal changes of P-AKT and P-TSC2 were reproduced equally well in both the with-crosstalk and compartment models (Figs 4C and 7B and C). However, the with-crosstalk model could not reproduce all the temporal changes of P-S6K under three conditions of AA(+) Ins(+), AA(+) Ins(−), and AA(−) Ins(+) (Fig 7D and E). In contrast, compared to the with-crosstalk model, the compartment model was able to reproduce all the temporal changes of P-S6K upon AA(+) Ins(+), AA(+) Ins(−), and AA(−) Ins(+) (Fig 7D and E), supporting our conclusion that lysosomal TSC2 localization and its

selective dephosphorylation on lysosomal surfaces fine-tune TSC2 phosphorylation levels and mTORC1 activation.

## Discussion

In this study, we established a mathematical model of the mTORC1 pathway combined with AA-sensing (Ragulator/LAMTOR-Rag) and Ins-sensing (AKT-TSC-Rheb) axes. This model reproduced the temporal regulation of the mTORC1 signaling pathway after AA and/or Ins treatment, as demonstrated by the phosphorylation levels of AKT at T308 and S473, TSC2 at T1462, and S6K at T389.

Ins-induced TSC2 phosphorylation at T1462 was preferentially inhibited during AA removal relative to AA administration, despite comparable levels of AKT phosphorylation at T308 and S473 upon Ins stimulation, irrespective of intracellular AA levels. Generally, protein phosphorylation results from the balance between kinase and phosphatase activities. The ratio of kinase/phosphatase activity to lysosomal TSC2 localization during AA deprivation likely prevents TSC2 phosphorylation by cytosolic AKT, contributing to the low level of AKT-induced TSC2 phosphorylation upon Ins stimulation only during AA removal. In addition, although PP1 and PP2A have been reported to primarily exhibit cytoplasmic distributions (Xin & Deng, 2006; Junttila et al, 2007; Medina et al, 2013), we have shown that the highly conserved scaffold (PP2A-Aα/β), catalytic (PP2A-Cα), and regulatory (PP2A-B56γ/δ) subunits of PP2A are constantly localized to the lysosomes, as demonstrated by immunostaining, CoIP assay, and Lyso-BioID. Furthermore, we discovered several somatic mutations that cluster in PP2A-B56 binding sites of TSC2 among previously reported TSC2 mutations: the L361P mutation in a tuberous sclerosis case (Dabora et al, 2001), E366K mutations in rectal adenocarcinoma (https://cancer.sanger.ac.uk/cosmic/mutation/overview?id=102914022) and ovarian carcinoma (Cheasley et al, 2019), and the E1583K mutation in skin carcinoma (Starrett et al, 2020). These findings strongly suggest that PP2A serves as a key regulator for TSC2, a molecular brake on the mTORC1 pathway, and its deregulation is associated with tuberous sclerosis or tumorigenesis. Lysosomal positioning has been reported to regulate mTORC1 signaling (Korolchuk et al, 2011; Jia & Bonifacino, 2019). After nutrient starvation, lysosomes accumulate in the vicinity of the nucleus and suppress mTORC1 activity, whereas upon stimulation with nutrients, such as AA and GF (e.g., Ins), lysosomes disperse throughout the cytoplasm to participate in mTORC1 activation. In our experimental conditions, we also observed perinuclear clustering of lysosomes, which was dependent on AA removal rather than GF removal. Given that the reaction is enhanced when the enzyme and substrate accumulate locally

---

Western blotting using the indicated antibodies. Note that the anti-P-S6 antibodies added during the first immunoblotting cover the antigen S6 protein, and therefore, the newly added anti-S6 antibodies are repelled when the same membrane is reblotted. **(F, G)** Comparison of the phosphorylation levels of TSC2 at T1462 with total TSC2 (F) and AKT at T308 with total AKT (G). *P*-values were assessed using one-way ANOVA with Tukey's multiple comparison test. NS, not significant. **(H)** Schematic model of the mTORC1 pathway regulatory mechanism. AKT is robustly phosphorylated and activated upon Ins stimulation irrespective of intracellular AA levels, whereas AKT-target TSC2 is more strongly phosphorylated upon Ins stimulation in combination with AA than under AA-free conditions. Mechanistically, under AA-supplemented conditions, TSC2 is distributed in the cytoplasm and is likely to be a poor target of lysosomal PP2A and more likely to be a target of cytosolic AKT. Alternatively, during AA removal, TSC2 localizes to the lysosomal membrane in a Ragulator-Rag–dependent manner. Accumulation of TSC2 and PP2A on the lysosomal surface enables PP2A to effectively dephosphorylate TSC2, leading to a low level of inhibitory TSC2 phosphorylation with high GAP activity during AA deprivation relative to that observed in the presence of intracellular AAs (see text for details). Source data are available for this figure.

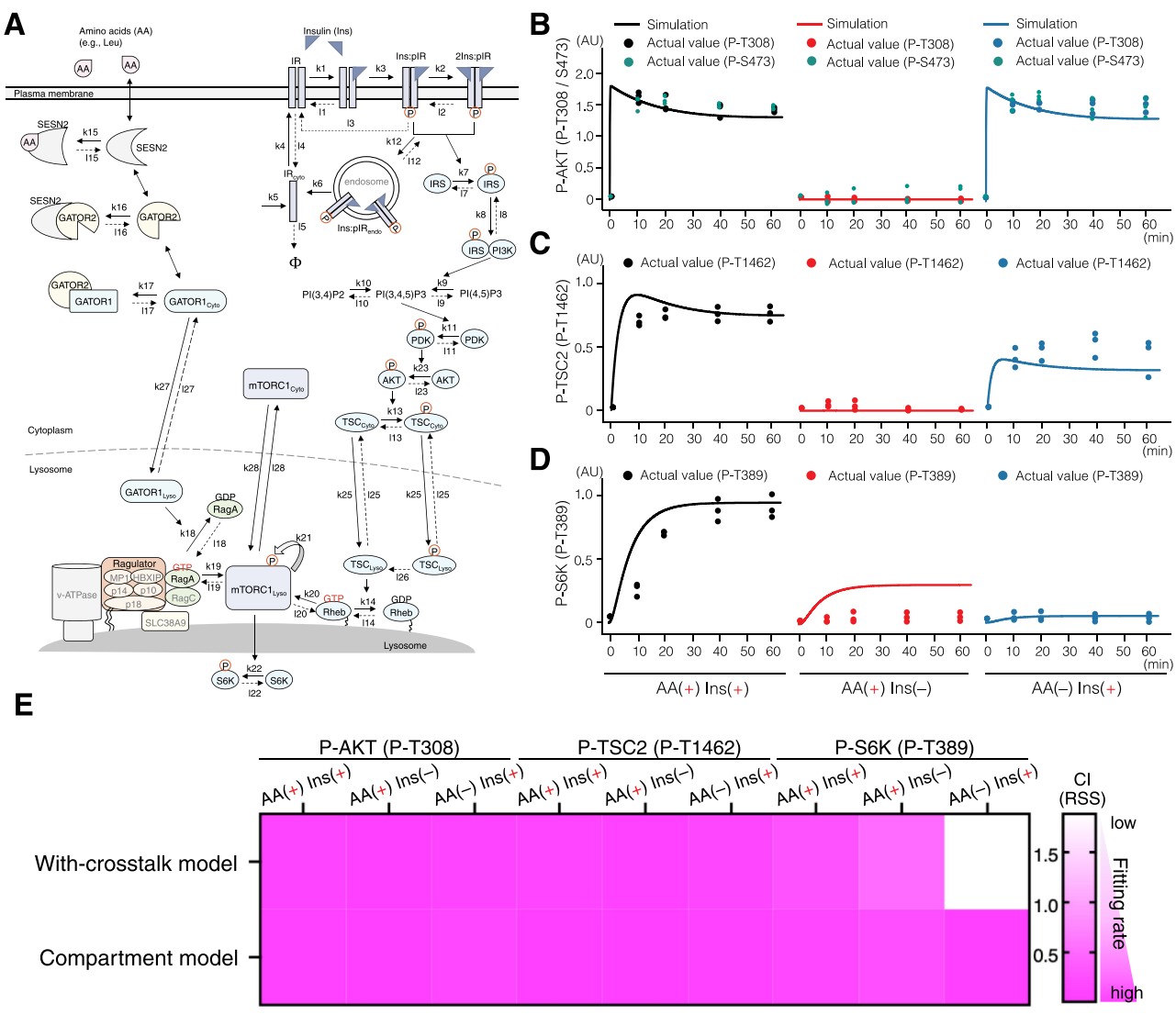

**Figure 7. Compartment model of the mTORC1 pathway reproduces the temporal changes of P-AKT, P-TSC, and P-S6K better than the crosstalk model.**
**(A)** Schematic representation of reactions involved in the spatiotemporal regulation of the mTORC1 pathway. A compartment model of the mTORC1 pathway was described based on the without-crosstalk model (Fig 2A) with the following modifications. Parameters k25, k27, and k28 were defined as the transport (import) rates for TSC, GATOR1, and mTORC1 from the cytoplasm to the lysosomes, whereas l25, l27, and l28 were defined as their export rates from the lysosomes to the cytoplasm. Parameter l26 was also defined as the dephosphorylation rate of TSC2 on the lysosomal surfaces. **(B, C, D)** Simulation of phospho-AKT at T308 and S473 (B), phospho-TSC2 at T1462 (C), and phospho-S6K at T389 (D) in the compartment model. The dots indicate the experimental values. AU, arbitrary units. **(E)** Correlations between the simulation and actual values of P-AKT(T308) (left), P-TSC2 (T1462) (middle), and P-S6K(T389) (right) were evaluated by the correlation indexes (CI) based on the residual sum of squares (RSS). A smaller CI (RSS) indicates a higher fitting rate of the model to the actual data.

(Sokolova et al, 2013; Banjade & Rosen, 2014; Banani et al, 2017; Zhang et al, 2021), the perinuclear clustering of lysosomes upon AA removal likely enhances the TSC2 association with PP2A on lysosomal surface, leading to PP2A-mediated dephosphorylation of TSC2. This is another reason for the low phosphorylation levels of TSC2 upon Ins stimulation during AA removal (Fig 6H). Although weakly phosphorylated at T1462, TSC2 continued to display relatively high GAP activity, which contributed to a decrease in active Rheb$^{GTP}$ and mTORC1 levels on the lysosomal surface during AA deprivation. OA is a specific PP2A and PP1 inhibitor with $IC_{50}$ values of 0.1–0.3 and 15–50 nM, respectively (Swingle et al, 2007). However, OA also

potently inhibits the activity of other phosphatases, such as PP4 ($IC_{50}$: 0.1 nM), PP5 ($IC_{50}$: 3.5 nM), and PP6 (Cohen, 1991; Swingle et al, 2007; Medina et al, 2013). In addition, our data showed that Ins-induced TSC2 phosphorylation at T1462 was elevated upon Ins and OA stimulation, relative to that upon Ins stimulation alone, despite comparable levels of AKT phosphorylation at T308 and S473 and S6K phosphorylation at T389, regardless of OA addition. Given that PP2A is involved in the dephosphorylation of AKT, S6K, and TSC2 (Ugi et al, 2004; Kuo et al, 2008; Xia et al, 2008; Hahn et al, 2010; Nada & Okada, 2020), and supported by proximity biotinylation (Lyso-BioID) data, it is possible that (a) phosphatase(s) other than PP2A (e.g., PP1, PP4,

PP5, calcineurin, and PP2C) may also contribute to the dephosphorylation of TSC2 at T1462 under AA starvation. Although further studies are required to fully understand the molecular mechanism of how PP2A and other phosphatases cooperatively regulate the mTORC1 signal transduction, our findings highlight PP2A-mediated TSC2 dephosphorylation on the lysosomal surface as a key regulatory mechanism that maintains low mTORC1 activation upon Ins stimulation during AA deprivation (Fig 6H). In addition to our data and those of previous studies (Demetriades et al, 2014; Carroll et al, 2016), a recent study has provided additional evidence of lysosomal localization upon AA removal by showing that Ragulator/LAMTOR-Rag mediates the AA starvation–induced lysosomal localization of MAP4K3, which is responsible for autophagic regulation (Hsu et al, 2018). Furthermore, intriguing evidence has shown that lysosomal lipid composition (e.g., PI(3)P, PI(4)P, and PI(3,5)P2) switches depending on the sensing of intracellular nutrient levels, including AAs, which leads to the dictation of anabolic or catabolic induction (Hong et al, 2017; Fitzian et al, 2021; Ebner et al, 2023). Thus, increasing evidence suggests that lysosomes are not only a site of degradation but also a signaling platform for the precise regulation of mTORC1-mediated metabolic processes or autophagy. In addition, given that mTORC1 signaling deregulation is associated with the etiology and pathophysiology of various human diseases, including cancer and diabetes, intense investigation of signaling regulation on the lysosomal surface is required for a deeper understanding of complex mTORC1 signaling transduction and the development of new therapeutic targets for disease treatment.

HeLa cells, a cervical cancer cell line resulting from papillomavirus (HPV) infection, have been widely used for experiments on mTORC1 signaling transduction. HPV-E6 oncoprotein induces the degradation of p53, a well-known inhibitor of the mTORC1 pathway (Horton et al, 2002; Feng et al, 2005; Budanov & Karin, 2008; Sengupta et al, 2010), leading to its down-regulation in HeLa cells (Buitrago-Pérez et al, 2009). We showed that HeLa cells express higher levels of Rheb (responsible for the serum-free–induced lysosomal localization of TSC2) and relatively lower levels of p18 (required for the AA-removal–induced lysosomal localization of TSC2) than HEK293 cells, p18Rev MEFs, and control MEFs (Fig S3B). This is likely why the suppression of Ins-induced TSC2 phosphorylation at T1462 was readily observed in HEK293 cells, p18Rev MEFs, and control MEFs (Ragulator(LAMTOR)-Rag/Rheb ratio: p18 ≥ Rheb) during AA starvation, whereas low suppression was observed in HeLa cells (Ragulator(LAMTOR)-Rag/Rheb ratio: p18 < Rheb) (Fig S3B). Thus, these data suggest that the Ragulator(LAMTOR)-Rag/Rheb ratio on the lysosomal surface likely regulates the direction of either AA- or Ins-induced release of TSC2 from lysosomes.

Consistent with the finding that lysosomal localization of TSC2 is primarily mediated by Ragulator/LAMTOR-Rag GTPase, our findings most likely explain why the activation level of mTORC1 was higher during AA deprivation in RagA KO or RagA/B KO cells than in WT cells (Efeyan et al, 2014; Kim et al, 2014). In addition to phosphatase-mediated crosstalk regulation of the mTORC1 pathway, another example is that of AA-induced Rheb polyubiquitination, which facilitates Rheb association with mTORC1 on the lysosomal surface, leading to mTORC1 activation (Yao et al, 2020). Furthermore, after the limitation of intracellular AA levels, Rap1-GTPase confines lysosomes to the perinuclear region and reduces lysosome abundance, leading to the suppression of mTORC1 activation (Mutvei et al, 2020). Therefore, incorporating the regulatory mechanisms through crosstalk, such as Rheb polyubiquitination, and the Rap1-mediated reduction of lysosomal abundance into the mathematical model of the mTORC1 signaling pathway will improve the mathematical model of the mTORC1 signaling pathway.

Given that the mathematical simulations of the mTORC1 pathway align well with the experimental data, our mathematical model enhances the quantitative understanding of the temporal regulation within the intricate mTORC1 signaling pathway, integrating both the AA- and GF-sensing axes. This underscores the vital role of mathematical modeling in surpassing the limitations of experimental approaches alone. Furthermore, given that PP2A-mediated TSC2 dephosphorylation on the lysosomal surface after AA removal is a regulatory mechanism of mTORC1 signaling, AA depletion (Butler et al, 2021) or the pharmaceutical promotion of TSC2 and/or PP2A lysosomal localization serves as a promising therapeutic strategy for the treatment of tumors with constitutive mTORC1 activation because of mutation-driven hyperactivation of the Ras and PI3K-AKT pathways.

# Materials and Methods

### Cell lines

p18Rev cells were generated by reintroducing p18 into p18-KO MEFs as previously described (Nada et al, 2009). Normal/parental MEF (Nakamura et al, 2013), p18-KO, p18Rev, p18/Rheb-dKO, and its derivative MEFs (Nada & Okada, 2020), HEK293, and HeLa cells were cultured in DMEM (4.5 g/liter glucose [08458-16; Nacalai Tesque] for MEFs; 1.0 g/liter glucose [08456-36; Nacalai Tesque] for HEK293 and HeLa) supplemented with 10% (vol/vol) fetal bovine serum (NICHIREI BIOSCIENCES INC) and penicillin/streptomycin (09367-34; Nacalai Tesque) at 37°C in a humidified atmosphere containing 5% $CO_2$. To analyze the temporal activation/phosphorylation status of the mTORC1 pathway in an AA- or Ins-dependent manner, semi-confluent cells were washed with PBS (27575-31; Nacalai Tesque) and incubated in DMEM (high glucose) without AAs (048-33575; Fujifilm Wako) for 1 h. The cells were Ins-stimulated through Ins (099-06473; Fujifilm Wako) (10 μg/ml) addition into the culture medium. For AA stimulation, starved cells were incubated in DMEM (high glucose) with AAs (08458-45; Nacalai Tesque) for designated time periods. OA (#5934; Cell Signaling Technology) (100 nM) was used to block the enzymatic activation of phosphatases PP2A and PP1. AKT inhibitor VIII (Akti-1/2) (S7776; Selleck) (1 μM), rapamycin (R8781; Sigma-Aldrich) (100 nM), Torin1 (4247; Tocris Bioscience) (100 nM), and LY2584702 (S7698; Selleck) (100 nM) were used to block AKT1/2, mTORC1, mTORC1/2, and S6K activation, respectively.

### Plasmids and transfection

cDNAs of TSC2-1740 aa (NM_001077183.3), LAMP1 (NM_005561.4), PPP2CA (PP2A-Cα) (NM_002715.4), PPP2CB (PP2A-Cβ) (NM_001009552.2), PPP2R1A (PP2A-Aα) (NM_014225.6), PPP2R1B (PP2A-Aβ) (NM_002716.5),

PPP2R2A (B55$\alpha$) (NM_002717.4), PPP2R2C (B55$\gamma$) (NM_181876.3), PPP2R2D (B55$\delta$) (NM_018461.5), PPP2R5A (B56$\alpha$) (NM_006243.4), PPP2R5B (B56$\beta$) (NM_006244.4), PPP2R5D (B56$\delta$) (NM_006245.4), PPP2R5E (B56$\varepsilon$) (XM_024449647.2), PPP2R3A (PR72) (XM_006713686.5), and PPP2R4 (PR53) (NM_178001.3) were obtained from Ehime/Kazusa Human cDNA clone library (https://www.pros.ehime-u.ac.jp/proteodrugdiscovery/technology/proteinarray2/). TSC2, GFP, PPP2CA, PPP2CB, PPP2R1A, PPP2R1B, PPP2R2A, PPP2R2C, PPP2R2D, PPP2R5A, PPP2R5B, PPP2R5D, PPP2R5E, PPP2R3A, and PPP2R4 were subcloned into pEU-bls, pEU-Flag (Sawasaki et al, 2002), pcDNA3-Flag, or pcDNA4-Myc (Nakamura et al, 2013). To create a pQCXIP-LAMP1-AGIA-AirID construct, AGIA-AirID was amplified from pcDNA3.1-AGIA-AirID (Kido et al, 2020) by PCR and fused in-frame to the 3′ terminal of the LAMP1 cDNA. The resulting recombinant LAMP1-AGIA-AirID sequence was subcloned into pQCXIP (Clontech). TSC2 phospho-deficient (T1395A identical to T1462A in TSC2-1807 aa) and phospho-mimetic (T1395D and T1395E) mutants were generated using PCR mutagenesis. TSC2(T1395A), TSC2(T1395D), and TSC2(T1395E) were subcloned into pcDNA3-Flag. To create a pQCXIP-LAMP1-Flag-TSC2 and its derivatives, Flag-TSC2, Flag-TSC2(T1395A), Flag-TSC2(T1395D), and Flag-TSC2(T1395E) were amplified from pcDNA3-Flag-TSC2 (WT, T1395A, T1395D, and T1395E) by PCR and fused to the 3′ terminal of the LAMP1 cDNA. Each resulting recombinant LAMP1-Flag-TSC2 (WT, T1395A, T1395D, and T1395E) sequence was subcloned into pQCXIP. Transfections were performed using X-tremeGENE 9 (6365809001; Merck). The total amount of plasmid DNA was adjusted to 1 $\mu$g/35 mm and 2.5 $\mu$g/60 mm dish with an empty vector. To generate LAMP1-AGIA-AirID stably expressing HEK293 cells, HEK293 cells were transfected with pQCXIP LAMP1-AGIA-AirID. After 2 wk of culture, the cells stably expressing LAMP1-AGIA-AirID were selected by puromycin (19752-64; Nacalai Tesque) treatment.

## Western blotting

Cells were lysed using ODG buffer (20 mM Tris–HCl [pH 7.4], 0.15 M NaCl, 1 mM EDTA, 1% NP-40, 2% octyl-$\beta$-D-glucoside, 5% glycerol, 1 mM Na$_3$VO$_4$, 10 mM NaF, and protease inhibitors) or lysis buffer (20 mM Tris–HCl [pH 7.5], 1% Triton X-100, 0.5% deoxycholate [DOC], 10% glycerol, 137 mM NaCl, 2 mM EDTA, 50 mM $\beta$-glycerophosphate, 10 mM NaF, 1 mM Na$_3$VO$_4$, 1 mM PMSF, 10 $\mu$g/ml leupeptin, and 10 $\mu$g/ml aprotinin). Equal amounts of total protein were separated using sodium dodecyl sulfate–polyacrylamide gel electrophoresis and transferred onto polyvinylidene difluoride or nitrocellulose membranes. Membranes were blocked and subsequently incubated with primary antibodies, followed by incubation with HRP-conjugated secondary anti-rabbit (#7074; Cell Signaling Technology) and anti-mouse (#7076; Cell Signaling Technology) antibodies. Signals were visualized using a WSE6200H LuminoGraph II (ATTO) or LAS-4000 (GE).

The following antibodies were used: anti-phospho-S6K (T389) rabbit monoclonal (108D2) (#9234; Cell Signaling Technology); anti-phospho-mTOR (S2481) rabbit polyclonal (#2974; Cell Signaling Technology); anti-phospho-AKT (T308) rabbit monoclonal (244F9) (#4056; Cell Signaling Technology); anti-phospho-AKT (S473) rabbit monoclonal (D9E) (#4060; Cell Signaling Technology); anti-phospho-TSC2 (T1462) rabbit monoclonal (5B12) (#3617; Cell Signaling Technology); anti-phospho-TSC2 (S664) rabbit monoclonal (D3B9Z) (#40729; Cell Signaling Technology); anti-phospho-TSC2

(S939) rabbit polyclonal (#3615; Cell Signaling Technology); anti-phospho-S6 (S235/S236) rabbit monoclonal (D57.2.2E) (#4858; Cell Signaling Technology); anti-phospho-ERK (T202/Y204) rabbit polyclonal (#9101; Cell Signaling Technology); anti-phospho-eIF2$\alpha$ (S51) rabbit polyclonal (#9721; Cell Signaling Technology); anti-phospho-GSK3$\beta$ (S9) rabbit monoclonal (D85E12) (#5558; Cell Signaling Technology); anti-S6K rabbit polyclonal (#9202; Cell Signaling Technology); anti-mTOR rabbit polyclonal (#2972; Cell Signaling Technology); anti-AKT1 rabbit monoclonal (C73H10) (#2938; Cell Signaling Technology); anti-AKT rabbit monoclonal (C67E7) (#4691; Cell Signaling Technology); anti-TSC2 rabbit polyclonal (#3612; Cell Signaling Technology); anti-TSC2 rabbit monoclonal (D93F12) (#4308; Cell Signaling Technology); anti-p18/LAMTOR1/C11orf59 rabbit monoclonal (D11H6) (#8975; Cell Signaling Technology); anti-p53 mouse monoclonal (1C12) (#2524; Cell Signaling Technology); anti-ERK1/2 mouse monoclonal (C-9) (sc-514302; Santa Cruz Biotechnology); anti-eIF2$\alpha$ mouse monoclonal (D-3) (sc-133132; Santa Cruz Biotechnology); anti-PP2A-A$\alpha$/$\beta$ mouse monoclonal (4G7) (sc-13600; Santa Cruz Biotechnology); anti-p70 S6K $\alpha$ mouse monoclonal (H9) (sc-8418; Santa Cruz Biotechnology); anti-S6 mouse monoclonal (C8) (sc-74459; Santa Cruz Biotechnology); anti-Rheb mouse monoclonal (B12) (sc-271509; Santa Cruz Biotechnology); anti-Myc mouse monoclonal (9E10) (sc-40; Santa Cruz Biotechnology); anti-PP2A-C$\alpha$ mouse monoclonal (46/PP2A; Catalytic a) (610555; BD Transduction Laboratories); anti-$\beta$-actin mouse monoclonal (6D1) (010-27841; Fujifilm Wako); anti-Flag mouse monoclonal (M2) antibody (F3165; Sigma-Aldrich). An anti-AGIA rabbit monoclonal antibody was made in-house (Yano et al, 2016). All antibodies were used at a dilution of 1:1,000, except anti-phospho-AKT (T308) (diluted 1:2,000), anti-phospho-AKT (S473) (diluted 1:2,000), anti-phospho-TSC2 (T1462) (diluted 1:2,000), anti-phospho-TSC2 (S939) (diluted 1:2,000), anti-phospho-S6K (T389) (diluted 1:2,000), anti-phospho-S6 (S235/S236) (diluted 1:5,000), anti-phospho-GSK3$\beta$ (S9) (diluted 1:5,000), and anti-AGIA (diluted 1:5,000). Streptavidin–HRP (02517-61; Nacalai Tesque) was used to detect biotinylated proteins at a dilution of 1:5,000.

## Co-immunoprecipitation assay

Cell lysates were prepared in CoIP lysis buffer (lysis buffer without DOC; 20 mM Tris–HCl [pH 7.5], 1% Triton X-100, 10% glycerol, 137 mM NaCl, 2 mM EDTA, 50 mM $\beta$-glycerophosphate, 10 mM NaF, 1 mM sodium vanadate, 1 mM dithiothreitol, 1 mM PMSF, 10 $\mu$g/ml leupeptin, and 10 $\mu$g/ml aprotinin) and incubated with the anti-Flag mouse monoclonal (M2) antibody (F3165; Sigma-Aldrich) for 5 h at 4°C with gentle rotation. Immunoprecipitates were recovered with protein G Sepharose beads (17061801; Cytiva) and washed four times with CoIP lysis buffer. Proteins were then separated by SDS–PAGE and immunoblotted with the indicated antibodies.

## Immunofluorescence

Immunofluorescence staining was performed as previously described (Nakamura et al, 2013). Briefly, cells grown on coverslips were fixed in 1% PFA (160-16061; Fujifilm Wako) or ice-cold methanol (21915-93; Nacalai Tesque), permeabilized with 50 $\mu$g/

ml digitonin (044-02121; Fujifilm Wako) or 0.1% Triton X-100 (35501-15; Nacalai Tesque) in PBS, blocked with Blocking One reagent (03953-95; Nacalai Tesque) or PBS containing 0.05% Triton X-100 and 1% globulin-free BSA (01281-26; Nacalai Tesque) for 1 h, and sequentially probed with primary antibodies.

The antibodies used were as follows: anti-mTOR rabbit monoclonal (7C10) (#2983; Cell Signaling Technology); anti-LAMP1 mouse monoclonal (H4A3) (sc-20011; Santa Cruz Biotechnology); anti-LAMP1 rat monoclonal (1D4B) (sc-19992; Santa Cruz Biotechnology); anti-TSC2 rabbit monoclonal (D93F12) (#4308; Cell Signaling Technology); anti-AKT1 rabbit monoclonal (C73H10) (#2938; Cell Signaling Technology); anti-PP2A-A$\alpha/\beta$ mouse monoclonal (4G7) (sc-13600; Santa Cruz Biotechnology); anti-PP2A-C$\alpha$ mouse monoclonal (46/PP2A; Catalytic a) (610555; BD Biosciences); anti-DDDDK mouse monoclonal (FLA-1) (M185-3L; MBL).

After washing with PBS, the cells were incubated with the appropriate secondary antibodies: Alexa Fluor Plus 405–conjugated anti-rabbit IgG (A-48254; Thermo Fisher Scientific), Alexa Fluor 488–conjugated anti-rabbit IgG (A-11008; Thermo Fisher Scientific), Alexa Fluor 488–conjugated anti-rabbit IgG (711-545-152; Jackson ImmunoResearch Laboratory), Alexa Fluor 488–conjugated anti-mouse IgG1 (A-21121; Thermo Fisher Scientific), Alexa Fluor 568–conjugated anti-mouse IgG (A-11019; Thermo Fisher Scientific), Alexa Fluor 568–conjugated anti-mouse IgG2a (A-21134; Thermo Fisher Scientific), Alexa Fluor 568–conjugated anti-rat IgG (A-11077; Thermo Fisher Scientific), and Alexa Fluor 647–conjugated anti-mouse IgG1 (A-21240; Thermo Fisher Scientific). All primary and secondary antibodies were used at a dilution of 1:1,000. For visualization of the biotinylated protein, cells were stained with Alexa Fluor 647–conjugated streptavidin (016-600-084, 1:2,000; Jackson ImmunoResearch Laboratory).

Next, the specimens were mounted on glass slides using Pro-Long Gold Antifade Reagent (P36930; Invitrogen) or FluorSave Reagent (345789; Merck) and examined using an FV1000 laser scanning microscope (Evident), a Nikon TiE inverted microscope (Nikon) equipped with an ORCA-Fusion BT Digital CMOS camera C15440 (Hamamatsu), or a BZ-X800 fluorescence microscope system (KEYENCE). To quantify the images, line scanning of the fluorescent signals was performed using Fiji/ImageJ software (NIH, https://imagej.net). For analysis of colocalization, Pearson's correlation coefficient was calculated using the Fiji/ImageJ Coloc2 plug-in.

### Absolute quantification of protein concentrations

Absolute quantification of protein concentrations, with in vitro proteome-assisted multiple reaction monitoring for protein absolute quantification (iMPAQT) (Matsumoto et al, 2017), was performed as follows.

### *Sample preparation for shotgun proteomics*
p18Rev cells were lysed in lysis buffer containing 1% SDS buffer and 50 mM Tris, pH 8.8, and sonicated using a Bioruptor (UCD-200; SONIC BIO Co.). The protein concentration of the lysates was determined using Pierce BCA Protein Assay Kit (Thermo Fisher Scientific) and adjusted to 1 mg/ml. Cysteine residues were reduced with 2.5 mM Tris(2-carboxyethyl)phosphine hydrochloride (Thermo

Fisher Scientific) for 30 min at 37°C and alkylated with 5 mM 2-iodoacetamide (Sigma-Aldrich) for 30 min at 25°C. Proteins were precipitated with acetone, dispersed via sonication in 25 mM triethylammonium bicarbonate, and then digested with trypsin (LAURUS Bio).

### *Preparation of internal standard peptides of signaling proteins*
To identify proteotypic peptides, protein digests were injected (200 ng/sample) into a pre-column L-column2 micro (Chemicals Evaluation and Research Institute [CERI], Japan) and separated using an in-house–fabricated 20-cm column packed with 2-$\mu$m octadecyl silane particles (CERI). Elution was performed with a linear gradient of 5–35% solvent B over 70 min at a flow rate of 200 nl/min (solvent A = 0.1% formic acid; solvent B = 0.1% formic acid in acetonitrile) using Dionex Ultimate 3000 HPLC System (Thermo Fisher Scientific). Raw DIA data were processed with DIA-NN (ver. 1.8) using the library search mode against mouse UniProt sequences for identification.

Proteotypic peptide sequences for the mTOR signaling pathway were concatenated into two synthetic artificial protein sequences (mTOR_concat01 and mTOR_concat02), and the corresponding cDNAs were synthesized (Thermo Fisher Scientific), subcloned into the pENTR2B vector, and subjected to recombination with pET21b-Qco-deAA-DEST by the LR reaction. Artificial protein sequences (mTOR_concat01 and mTOR_concat02) are listed in Table 2. To obtain isotopically labeled recombinant proteins with maximum labeling efficiency, we generated expression strains using P1 transduction of *Escherichia coli* BL21(DE3), which is auxotrophic for arginine and lysine and is derived from argA and lysA deletion mutants in the Keio Collection of single-gene knockouts. Recombinant proteins were expressed as His6-tagged molecules in the auxotrophic cells cultured in the presence of [$^{13}C_6/^{15}N_2$]Lys and [$^{13}C_6/^{15}N_4$]Arg and purified with Ni-resin.

### *Determination of cellular protein concentration by spike-in DIA proteomics*
Isotopically labeled concatemers were spiked into cell lysates (20 $\mu$g each) with four threefold serial dilutions from 5,000 fmol. The lysates were subjected to enzymatic digestion as described above. For absolute quantification of proteins, the digests containing isotopically labeled recombinant proteins were separated on a column (EV1106, Evosep) using a pre-programmed gradient (15 samples per day) at a flow rate of 200 nl/min using the Evosep One system (Evosep). The eluted peptides were sprayed with an emitter (FortisTip, AMR) interfaced with an Exploris 480 equipped with a front-end high-field asymmetric waveform ion mobility spectrometer (FAIMS) Pro. Exploris 480 was operated in the DIA mode. All data were acquired in profile mode using positive polarity. The spray voltage was set to 1.7 kV, high-field asymmetric waveform ion mobility spectrometry CV to –45 V, funnel RF level at 40, and heated capillary temperature at 275°C. For complete MS spectrum acquisition, the resolution was set to 120,000 at m/z 200, and the complete MS AGC target was 200%, with a maximum IT of 45 msec. The mass range was set to 480–880. For fragment spectrum acquisition, the resolution was set to 30,000, IT to auto, and the mass range to 145–1,450 m/z. The AGC target value for the fragment spectra was set at 1,000%. Twenty-six DIA windows of

15 units ranging from 480 to 880 with an overlap of one unit were used. The normalized collision energy was set to 28%. Raw MS files were processed with DIA-NN with a spectrum library containing the targeted peptides with and without $[^{13}C_6/^{15}N_2]$Lys and $[^{13}C_6/^{15}N_4]$Arg labeling. The absolute abundance of proteins was determined based on the height ratio of light (sample) to heavy (internal standard) peaks in the chromatogram. To calculate the total protein concentration per cell, p18Rev cells ($2 \times 10^6$ cells) were lysed with 200 μl of lysis buffer, and the protein concentration in the lysate (1.69 μg/μl) was measured using an absorption spectrometer. Because 200 μl of cell lysate is made from $2 \times 10^6$ cells, the total protein per cell is 169 pg/cell from the formula [1.69 (μg/μl) × 200 (μl) × 1/($2 \times 10^6$ (cells))]. The average volume per cell was defined as $3.4 \times 10^{-12}$ L/cell ($2.8 \times 10^{-12}$ L/cell [cytoplasm] + $0.6 \times 10^{-12}$ L/cell [nucleus]), according to a previous report (Aoki et al, 2011).

### RPPA

p18Rev cells were amino acid–starved for 1 h and subsequently stimulated with amino acids and insulin for the indicated periods. Cells were then lysed using the T-PER protein extraction reagent (78510; Thermo Fisher Scientific), supplemented with inhibitors (100 mM NaF, 1 mM $Na_3VO_4$, 10 mM NaPi-IIb, 1 mM EDTA, PhosSTOP, 4906845001; Sigma-Aldrich) and protease inhibitor cocktails (1.04 mM AEBSF, 800 nM aprotinin, 40 μM bestatin, 14 μM E-64, 20 μM leupeptin, and 15 μM pepstatin A; Sigma-Aldrich). The lysates were scraped from the culture dish, collected in each tube, and centrifuged at 15,000$g$ at 4°C for 20 min; the supernatant was subjected to the following analyses: after adjusting the protein concentration to ~1.0 mg/ml according to the Bradford protein assay (5000006; Bio-Rad), the lysates were manually serially diluted twofold with an extraction buffer. The diluted lysates were boiled in 2% SDS and 2.5% β-mercaptoethanol and printed onto nitrocellulose-coated slides in four replicates (Grace Bio-Labs) using Aushon Biosystems 2470 Arrayer. After blocking with an ODYSSEY blocking buffer (927-40100; LI-COR Biosciences) supplemented with 0.1% Tween-20, the blotted slides were probed with validated primary antibodies (anti-phospho-mTOR [S2481] rabbit monoclonal [EPR427(N)] [ab137133; Abcam]; anti-phospho-S6K [T389] rabbit monoclonal [108D2] [#9234; Cell Signaling Technology]), followed by secondary antibodies conjugated to infrared dyes, IRDye 680RD and 800CW (LI-COR Biosciences). Six replicate slides were scanned using an ODYSSEY scanner (LI-COR Biosciences). The signal intensity of each spot was quantified using Image Studio (LI-COR Biosciences), according to the manufacturer's instructions. For each slide, the RPPA data of protein expression and phosphorylation for each cell line were normalized to the RPPA intensity of the house-keeping protein γ-tubulin, log2-transformed, and z-score–standardized across all cell lines.

### Mathematical simulation

We developed a mathematical model of the insulin-dependent AKT-TSC-Rheb pathway, consisting of 25 Equations (46), (47), (48), (49), (50), (51), (52), (53), (54), (55), (56), (57), (58), (59), (60), (61), (62), (63), (64), (65), (66), (67), (68), (69), and (70), based on previous reports (Sedaghat et al, 2002; Jain & Bhalla, 2009) that described the dynamic regulation

of post-insulin receptor events, including IRS1 phosphorylation, PI3K activation, and subsequent AKT activation.

We extended the Ins receptor (IR) life cycle subsystem (encompassing synthesis, degradation, exocytosis, and both basal and ligand-induced endocytosis of IRs) so that ligand-induced endocytosis is applied only to phosphorylated cell surface IRs. Both once- and twice-bound phosphorylated IRs were treated identically with respect to internalization. An additional step, including dephosphorylation of internalized phosphorylated IRs and their incorporation into the intracellular pool, was added. State variables representing free surface IRs and phosphorylated surface IRs (Ins:IR and 2Ins:IR) are shared with the binding and recycling subsystems. Thus, the differential equations for these coupled subsystems are as follows:

$$\frac{d[Ins]}{dt} = -k1[Ins][IR] + l1[Ins:IR] - k2[Ins][Ins:pIR] + l2[2Ins:pIR] + l3[Ins:pIR] \quad (46)$$

$$\frac{d[IR]}{dt} = l1[Ins:IR] + l3[Ins:pIR] - k1[Ins][IR] + l4[IR_{cyto}] - k4[IR] \quad (47)$$

$$\frac{d[Ins:IR]}{dt} = k1[Ins][IR] - l1[Ins:IR] - k3[Ins:IR] \quad (48)$$

$$\frac{d[2Ins:pIR]}{dt} = k2[Ins][Ins:pIR] - l2[2Ins:pIR] + l12[2Ins:pIR_{endo}] - k12[2Ins:pIR] \quad (49)$$

$$\frac{d[Ins:pIR]}{dt} = k3[Ins:IR] - k2[Ins][Ins:pIR] + l2[2Ins:pIR] - l3[Ins:pIR] - k12[Ins:pIR] + l12[Ins:pIR_{endo}] \quad (50)$$

$$\frac{d[IR_{cyto}]}{dt} = k5 - l5[IR_{cyto}] + k6([2Ins:pIR_{endo}] + [Ins:pIR_{endo}]) + k4[IR] - l4[IR_{cyto}] \quad (51)$$

$$\frac{d[2Ins:pIR_{endo}]}{dt} = k12[2Ins:pIR] - l12[2Ins:pIR_{endo}] - k6[2Ins:pIR_{endo}] \quad (52)$$

$$\frac{d[Ins:pIR_{endo}]}{dt} = k12[Ins:pIR] - l12[Ins:pIR_{endo}] - k6[Ins:pIR_{endo}] \quad (53)$$

The post-receptor signaling model comprises key elements of the metabolic Ins signaling pathway, assuming that this is a closed subsystem in which synthesis and degradation of signaling molecules are not explicitly represented. The concentration of phosphorylated surface IRs serves as the input signal. Activated IRs phosphorylate IRS, which subsequently binds and activates PI3K. The dependence of IRS phosphorylation on phosphorylated surface IRs is modeled as a linear function. The rate constant for IRS phosphorylation (pIRS), k7, is modulated by the fraction of phosphorylated surface IRs, calculated as ([2Ins:pIR] + [Ins:pIR])/($8.97 \times 10^{-13}$ M), where $8.97 \times 10^{-13}$ M is the concentration of phosphorylated surface IRs achieved after maximal insulin stimulation. The association of phosphorylated IRS (pIRS) with PI3K was assumed to occur at a 1:1 stoichiometry. The differential equations describing the dynamics of pIRS and the subsequent formation of the pIRS/activated PI3K complex are as follows:

$$\frac{d[IRS]}{dt} = l7[pIRS] - \frac{k7[IRS]([2Ins:pIR] + [Ins:pIR])}{8.97 \times 10^{-13}} \quad (54)$$

$$\frac{d[pIRS]}{dt} = \frac{k7[IRS]([2Ins:pIR] + [Ins:pIR])}{8.97 \times 10^{-13}}$$
$$+ l8[pIRS:PI3K] - l7[pIRS] - k8[pIRS][PI3K] \quad (55)$$

$$\frac{d[PI3K]}{dt} = l8[pIRS:PI3K] - k8[pIRS][PI3K] \quad (56)$$

$$\frac{d[pIRS:PI3K]}{dt} = k8[pIRS][PI3K] - l8[pIRS:PI3K] \quad (57)$$

Active PI3K-mediated conversion of phosphatidylinositol [PI(4,5)P2] to PI(3,4,5)P3 is modeled as a linear function so that k9 (1.259 × [pIRS1:PI3K]/(5 × 10$^{-13}$) + 0.131), the rate constant for the generation of PI(3,4,5)P3, is dependent on the (pIRS:PI3K) complex, which represents activated PI3K. The differential equations describing the interconversion between PI(3,4,5)P3, PI(4,5)P2, and PI(3,4)P3 are as follows:

$$\frac{d[PI(3,4,5)P3]}{dt} = k9[PI(4,5)P2] + k10[PI(3,4)P2] - l9[PI(3,4,5)P3]$$
$$- l10[PI(3,4,5)P3] \quad (58)$$

$$k9 = \frac{1.259[pIRS1:PI3K]}{1 \times 10^{-13}} + 0.131 \quad (59)$$

$$\frac{d[PI(4,5)P2]}{dt} = l9[PI(3,4,5)P3] - k9[PI(4,5)P2] \quad (60)$$

$$\frac{d[PI(3,4)P2]}{dt} = l10[PI(3,4,5)P3] - k10[PI(3,4)P2] \quad (61)$$

Activation of the downstream Ser/Thr kinases PDK (PI(3,4,5)P3: PDK) and AKT (pAKT) depends on PI(3,4,5)P3 levels. In the model, this activation is governed by rate constants (k11) that are functions of PI(3,4,5)P3 concentration. The corresponding differential equations are as follows:

$$\frac{d[PDK]}{dt} = l11[PI(3,4,5)P3:PDK] - k11[PDK] \quad (62)$$

$$\frac{d[PI(3,4,5)P3:PDK]}{dt} = k11[PDK] - l11[PI(3,4,5)P3:PDK] \quad (63)$$

$$k11 = \frac{0.6931([PI(3,4,5)P3] - 0.31)}{2.79} \quad (64)$$

$$\frac{d[AKT]}{dt} = l23[pAKT] - \frac{k23[PI(3,4,5)P3:PDK][AKT]}{0.4 \times 10^{-6} + [AKT]} \quad (65)$$

$$\frac{d[pAKT]}{dt} = \frac{k23[PI(3,4,5)P3:PDK][AKT]}{0.4 \times 10^{-6} + [AKT]} - l23[pAKT] \quad (66)$$

Active AKT phosphorylates TSC2 within the TSC1-TSC2-TBC1D7 complex (pTSC). The unphosphorylated TSC1-TSC2-TBC1D7 complex (TSC) exhibits GAP activity, promoting the conversion of active Rheb$^{GTP}$, an activator of mTORC1, to inactive Rheb$^{GDP}$, thereby leading to mTORC1 inactivation. The corresponding differential equations are as follows:

$$\frac{d[TSC]}{dt} = l13[pTSC] - \frac{k13[pAKT][TSC]}{10.3 \times 10^{-6} + [TSC]} \quad (67)$$

$$\frac{d[pTSC]}{dt} = \frac{k13[pAKT][TSC]}{10.3 \times 10^{-6} + [TSC]} - l13[pTSC] \quad (68)$$

$$\frac{d\left[Rheb^{GTP}\right]}{dt} = l14\left[Rheb^{GDP}\right] - \frac{k14[TSC]\left[Rheb^{GTP}\right]}{0.3 \times 10^{-6} + \left[Rheb^{GTP}\right]}$$
$$- k20\left[Ragulator:RagA^{GTP}:mTORC1\right]\left[Rheb^{GTP}\right]$$
$$+ l20\left[Ragulator:RagA^{GTP}:mTORC1:Rheb^{GTP}\right] \quad (69)$$

$$\frac{d\left[Rheb^{GDP}\right]}{dt} = \frac{k14[TSC]\left[Rheb^{GTP}\right]}{0.3 \times 10^{-6} + \left[Rheb^{GTP}\right]} - l14\left[Rheb^{GDP}\right] \quad (70)$$

In addition, we developed an AA-sensing Ragulator/LAMTOR-Rag pathway model based on standard reaction network system characteristics, consisting of nine Equations (71), (72), (73), (74), (75), (76), (77), (78), and (79) (Liu & Sabatini, 2020). Cytosolic AA signals are relayed via AA-sensing (e.g., SESN, CASTOR)–GATOR2-GATOR1-Rag axes. GATOR1, composed of DEPDC5, NPRL2, and NPRL3, exhibits GAP activity through the NPRL2 subunit, which hydrolyzes the GTP on Rag, inhibiting mTORC1 activation. As GATOR1 physically interacts with GATOR2 (a pentameric complex of WDR59, WDR24, MIOS, SEH1, and SEC13), the GATOR2 complex antagonizes GATOR1 function and serves as a positive regulator for mTORC1 activation upon AA sensing. Upon AA starvation (e.g., leu starvation), the cytosolic leu sensor SESN2 binds and inhibits GATOR2, preventing lysosomal mTORC1 localization. Refeeding with AA restores cytosolic leu levels, allowing AA to bind to a pocket on SESN2, which dissociates from GATOR2, thereby relieving GATOR1-mediated mTORC1 inhibition and activating mTORC1. The corresponding differential equations are as follows:

$$\frac{d[AA]}{dt} = -k15[AA][SESN2] + l15[AA:SESN2] \quad (71)$$

$$\frac{d[SESN2]}{dt} = -k15[AA][SESN2] + l15[AA:SESN2] - k16[SESN2][GATOR2]$$
$$+ l16[SESN2:GATOR2] \quad (72)$$

$$\frac{d[AA:SESN2]}{dt} = k15[AA][SESN2] - l15[AA:SESN2] \quad (73)$$

$$\frac{d[GATOR2]}{dt} = -k16[SESN2][GATOR2] + l16[SESN2:GATOR2]$$
$$- k17[GATOR2][GATOR1] + l17[GATOR2:GATOR1] \quad (74)$$

$$\frac{d[SESN2:GATOR2]}{dt} = k16[SESN2][GATOR2] - l16[SESN2:GATOR2]$$
$$\quad (75)$$

$$\frac{d[GATOR1]}{dt} = -k17[GATOR2][GATOR1] + l17[GATOR2:GATOR1] \quad (76)$$

$$\frac{d[GATOR2:GATOR1]}{dt} = k17[GATOR2][GATOR1] - l17[GATOR2:GATOR1]$$
$$\quad (77)$$

$$\frac{d\left[Ragulator:Rag^{GTP}\right]}{dt} = -\frac{k18[GATOR1]\left[Ragulator:Rag^{GTP}\right]}{1 \times 10^{-7} + \left[Ragulator:Rag^{GTP}\right]}$$
$$+ l18\left[Ragulator:Rag^{GDP}\right] - k19\left[Ragulator:Rag^{GTP}\right][mTORC1]$$
$$+ l19\left[Ragulator:Rag^{GTP}:mTORC1\right] \quad (78)$$

$$\frac{d\left[Ragulator:Rag^{GDP}\right]}{dt} = \frac{k18[GATOR1]\left[Ragulator:Rag^{GTP}\right]}{1 \times 10^{-7} + \left[Ragulator:Rag^{GTP}\right]}$$
$$- l18\left[Ragulator:Rag^{GDP}\right] \quad (79)$$

Based on the data, complete mTORC1 activation (pmTORC1) is required for lysosomal mTOR localization, which is mediated by the Ragulator/LAMTOR-Rag complex (Ragulator:Rag$^{GTP}$) upon AA treatment and subsequent association of mTORC1 with lysosomal Rheb$^{GTP}$. Active pmTORC1 phosphorylates its target S6K (pS6K). We defined these mTORC1 activation reactions on the lysosomal surface using six Equations (80), (81), (82), (83), (84), and (85) and subsequently integrated the two models to generate the combined mTORC1 pathway model. The corresponding differential equations are as follows:

$$\frac{d[mTORC1]}{dt} = -k19[Ragulator:Rag^{GTP}][mTORC1] + l19[Ragulator:Rag^{GTP}:mTORC1] \quad (80)$$

$$\frac{d[Ragulator:Rag^{GTP}:mTORC1]}{dt} = k19[Ragulator:Rag^{GTP}][mTORC1] - l19[Ragulator:Rag^{GTP}:mTORC1]$$

$$- k20[Ragulator:Rag^{GTP}:mTORC1][Rheb^{GTP}]$$

$$+ l20[Ragulator:Rag^{GTP}:mTORC1:Rheb^{GTP}] \quad (81)$$

$$\frac{d[Ragulator:Rag^{GTP}:mTORC1:Rheb^{GTP}]}{dt} = k20[Ragulator:Rag^{GTP}:mTORC1][Rheb^{GTP}] - l20[Ragulator:Rag^{GTP}:mTORC1:Rheb^{GTP}] - k21[Ragulator:Rag^{GTP}:mTORC1:Rheb^{GTP}]$$

$$+ l21[Ragulator:Rag^{GTP}:pmTORC1:Rheb^{GTP}] \quad (82)$$

$$\frac{d[Ragulator:Rag^{GTP}:pmTORC1:Rheb^{GTP}]}{dt} = k21[Ragulator:Rag^{GTP}:mTORC1:Rheb^{GTP}] - l21[Ragulator:Rag^{GTP}:pmTORC1:Rheb^{GTP}] \quad (83)$$

$$\frac{d[S6K]}{dt} = -\frac{k22[Ragulator:Rag^{GTP}:pmTORC1:Rheb^{GTP}][S6K]}{0.8 \times 10^{-6}+[S6K]} + \frac{l22 \times 0.15 \times 10^{-6}[pS6K]}{8.8 \times 10^{-6}+[pS6K]} \quad (84)$$

$$\frac{d[pS6K]}{dt} = \frac{k22[Ragulator:Rag^{GTP}:pmTORC1:Rheb^{GTP}][S6K]}{0.8 \times 10^{-6}+[S6K]} - \frac{l22 \times 0.15 \times 10^{-6}[pS6K]}{8.8 \times 10^{-6}+[pS6K]} \quad (85)$$

Simulations and parameter estimations were performed using Mathematica software version 14.1 (Wolfram Research, Champaign, IL, USA). The phosphorylation level of TSC2 at the T1462 residue significantly changed upon simultaneous stimulation with AA and Ins (AA(+) Ins(+)) compared with Ins stimulation alone (AA(−) Ins(+)). To account for this, we apply the reactions involving the AA-sensitive PPase that dephosphorylates TSC2 at the T1462 residue and incorporate five additional equations into the mathematical model with crosstalk (Equations (86), (87), (88), (89), and (90)).

$$\frac{d[TSC]}{dt} = \frac{l13[PPase][pTSC]}{8.8 \times 10^{-6}+[pTSC]} - \frac{k13[pAKT][TSC]}{10.3 \times 10^{-6}+[TSC]} \quad (86)$$

$$\frac{d[pTSC]}{dt} = \frac{k13[pAKT][TSC]}{10.3 \times 10^{-6}+[TSC]} - \frac{l13[PPase][pTSC]}{8.8 \times 10^{-6}+[pTSC]} \quad (87)$$

$$\frac{d[AA]}{dt} = -k15[AA][SESN2] + l15[AA:SESN2] - k24[AA][PPase] + l24[AA:PPase] \quad (88)$$

$$\frac{d[PPase]}{dt} = -k24[AA][PPase] + l24[AA:PPase] \quad (89)$$

$$\frac{d[AA:PPase]}{dt} = k24[AA][PPase] - l24[AA:PPase] \quad (90)$$

A compartment model was defined to represent the AA concentration–dependent cytoplasm–lysosome distribution of TSC (TSC1-TSC2-TBC1D7) and mTORC1 (Fig 7A). The compartment model was described based on the without-crosstalk model (Fig 2A) with modifications as follows: k25, k27, and k28 were defined as the import rates for cytoplasmic TSC, GATOR1, and mTORC1 to the lysosomes, whereas l25, l27, and l28 were defined as their export rates from the lysosomes to the cytoplasm. Parameter l26 was also defined as the dephosphorylation rate of TSC2 on the lysosomal surfaces. Because lysosomes are reported to constitute 1–5% of the intracellular volume (Wartosch et al, 2015), protein concentrations in lysosomes (e.g., Ragulator-Rag and Rheb) were calculated by multiplying the protein concentration in the intracellular cell by 20 (100/5) – 100 (100/1) (Table 3).

$$\frac{d[TSC_{Cyto}]}{dt} = \frac{l13[PPase_{Cyto}][pTSC_{Cyto}]}{8.8 \times 10^{-6}+[pTSC_{Cyto}]} - \frac{k13[pAKT][TSC_{Cyto}]}{10.3 \times 10^{-6}+[TSC_{Cyto}]} - k25[TSC_{Cyto}] + l25[TSC_{Lyso}] \quad (91)$$

$$\frac{d[pTSC_{cyto}]}{dt} = \frac{k13[pAKT][TSC_{Cyto}]}{10.3 \times 10^{-6}+[TSC_{Cyto}]} - \frac{l13[PPase_{Cyto}][pTSC_{Cyto}]}{8.8 \times 10^{-6}+[pTSC_{Cyto}]} - k25[pTSC_{Cyto}] + l25[pTSC_{Lyso}] \quad (92)$$

$$\frac{d[TSC_{Lyso}]}{dt} = \frac{l26[PPase_{Lyso}][pTSC_{Lyso}]}{8.8 \times 10^{-6}+[pTSC_{Lyso}]} + k25[TSC_{Cyto}] - l25[TSC_{Lyso}] \quad (93)$$

$$\frac{d[pTSC_{Lyso}]}{dt} = -\frac{l26[PPase_{Lyso}][pTSC_{Lyso}]}{8.8 \times 10^{-6}+[pTSC_{Lyso}]} + k25[pTSC_{Cyto}] - l25[pTSC_{Lyso}] \quad (94)$$

$$[PPase_{Cyto}] \ (constant) \quad (95)$$

$$[PPase_{Lyso}] \ (constant) \quad (96)$$

$$[PPase_{Cyto}] + [PPase_{Lyso}] = 0.15 \times 10^{-6} \ (M) \quad (97)$$

$$\frac{d[Rheb^{GTP}]}{dt} = l14[Rheb^{GDP}] - \frac{k14[TSC_{Lyso}][Rheb^{GTP}]}{0.3 \times 10^{-6}+[Rheb^{GTP}]} - k20[Ragulator:RagA^{GTP}:mTORC1_{Lyso}][Rheb^{GTP}] + l20[Ragulator:RagA^{GTP}:mTORC1_{Lyso}:Rheb^{GTP}] \quad (98)$$

$$\frac{d[Rheb^{GDP}]}{dt} = \frac{k14[TSC_{Lyso}][Rheb^{GTP}]}{0.3 \times 10^{-6}+[Rheb^{GTP}]} - l14[Rheb^{GDP}] \quad (99)$$

$$\frac{d[GATOR1_{Cyto}]}{dt} = -k17[GATOR2][GATOR1_{Cyto}] + l17[GATOR2:GATOR1_{Cyto}] - k27[GATOR1_{Cyto}] + l27[GATOR1_{Lyso}] \tag{100}$$

$$\frac{d[GATOR2:GATOR1_{Cyto}]}{dt} = k17[GATOR2][GATOR1_{Cyto}] - l17[GATOR2:GATOR1_{Cyto}] \tag{101}$$

$$\frac{d[GATOR1_{Lyso}]}{dt} = k27[GATOR1_{Cyto}] - l27[GATOR1_{Lyso}] \tag{102}$$

$$\frac{d[Ragulator:Rag^{GTP}]}{dt} = -\frac{k18[GATOR1_{Lyso}][Ragulator:Rag^{GTP}]}{1 \times 10^{-7} + [Ragulator:Rag^{GTP}]} + l18[Ragulator:Rag^{GDP}] - k19[Ragulator:Rag^{GTP}][mTORC1_{Lyso}]$$
$$+ l19[Ragulator:Rag^{GTP}:mTORC1_{Lyso}] \tag{103}$$

$$\frac{d[Ragulator:Rag^{GDP}]}{dt} = \frac{k18[GATOR1_{Lyso}][Ragulator:Rag^{GTP}]}{1 \times 10^{-7} + [Ragulator:Rag^{GTP}]} - l18[Ragulator:Rag^{GDP}] \tag{104}$$

$$\frac{d[mTORC1_{Cyto}]}{dt} = -k28[mTORC1_{Cyto}] + l28[mTORC1_{Lyso}] \tag{105}$$

$$\frac{d[mTORC1_{Lyso}]}{dt} = -k19[Ragulator:Rag^{GTP}][mTORC1_{Lyso}] + l19[Ragulator:Rag^{GTP}:mTORC1_{Lyso}] + k28[mTORC1_{Cyto}]$$
$$- l28[mTORC1_{Lyso}] \tag{106}$$

$$\frac{d[Ragulator:Rag^{GTP}:mTORC1_{Lyso}]}{dt} = k19[Ragulator:Rag^{GTP}][mTORC1_{Lyso}] - l19[Ragulator:Rag^{GTP}:mTORC1_{Lyso}]$$
$$- k20[Ragulator:Rag^{GTP}:mTORC1_{Lyso}][Rheb^{GTP}]$$
$$+ l20[Ragulator:Rag^{GTP}:mTORC1_{Lyso}:Rheb^{GTP}] \tag{107}$$

$$\frac{d[Ragulator:Rag^{GTP}:mTORC1_{Lyso}:Rheb^{GTP}]}{dt} =$$
$$k20[Ragulator:Rag^{GTP}:mTORC1_{Lyso}][Rheb^{GTP}]$$
$$- l20[Ragulator:Rag^{GTP}:mTORC1_{Lyso}:Rheb^{GTP}]$$
$$- k21[Ragulator:Rag^{GTP}:mTORC1_{Lyso}:Rheb^{GTP}]$$
$$+ l21[Ragulator:Rag^{GTP}:pmTORC1_{Lyso}:Rheb^{GTP}] \tag{108}$$

$$\frac{d[Ragulator:Rag^{GTP}:pmTORC1_{Lyso}:Rheb^{GTP}]}{dt} =$$
$$k21[Ragulator:Rag^{GTP}:mTORC1_{Lyso}:Rheb^{GTP}]$$
$$- l21[Ragulator:Rag^{GTP}:pmTORC1_{Lyso}:Rheb^{GTP}] \tag{109}$$

$$\frac{d[S6K]}{dt} = -\frac{k22[Ragulator:Rag^{GTP}:pmTORC1_{Lyso}:Rheb^{GTP}][S6K]}{0.8 \times 10^{-6} + [S6K]}$$
$$+ \frac{l22 \times [PPase_{Lyso}][pS6K]}{8.8 \times 10^{-6} + [pS6K]} \tag{110}$$

$$\frac{d[pS6K]}{dt} = \frac{k22[Ragulator:Rag^{GTP}:pmTORC1_{Lyso}:Rheb^{GTP}][S6K]}{0.8 \times 10^{-6} + [S6K]}$$
$$- \frac{l22 \times [PPase_{Lyso}][pS6K]}{8.8 \times 10^{-6} + [pS6K]} \tag{111}$$

## Structure visualization

The cryo-EM structural data for the human TSC (TSC1-TSC2-TBC1D7) complex (PDB ID: 7DL2) and the crystal structure of Ragulator/LAMTOR in complex with the roadblock domains of RagA/C (PDB ID: 5X6V) were obtained from Protein Data Bank Japan (https://pdbj.org). The protein structures of the human TSC1-TSC2-TBC1D7 complex and the Ragulator/LAMTOR-Rag complex were visualized using PyMOL (The PyMOL Molecular Graphics System, version 3.1.0; Schrödinger LLC).

## siRNA knockdown experiment

HEK293 cells were transfected with siRNAs targeting human PP2A-Aα (SASI_Hs01_00232389; Merck), PP2A-Aβ (SASI_Hs01_00040968; Merck), PP2A-Cα (SASI_Hs01_00107753; Merck), AKT1 (SASI_Hs01_00205545; Merck), AKT2 (SASI_Hs02_00332190; Merck), TSC2 (SASI_Hs01_00127335; Merck), or mouse TSC2 (SASI_Mm01_00052652; Merck) using Lipofectamine RNAiMAX Transfection Reagent (13778030; Thermo Fisher Scientific). Two days after transfection, the medium was replaced with a fresh medium, and the cells were further cultured for 4 h before subsequent experiments. The targeting sequences were as follows: siPP2A-Aα (human), 5′-GUGAUCAUGUCCCAGAUCUdTdT-3′; siPP2A-Aβ (human), 5′-CUAAUUACUUGCAUAGAAUdTdT-3′; siPP2A-Cα (human), 5′-CUGGUUACACCUUUGGGCAdTdT-3′; siAKT1 (human), 5′-GUGCCAUGAUCUGUAUUUAdTdT-3′; siAKT2 (human), 5′-CCAUGAA-GAUCCUGCGGAAdTdT-3′; siTSC2 (human), 5′-CCAUCAAGGGCCA-GUUCAAdTdT-3′; siTSC2 (mouse), 5′-GAGAUUGUUCUGUCCAUAAdTdT-3′. AllStars Negative Control siRNA (1027281; QIAGEN) was used as a control.

## AlphaScreen-based biochemical assays using recombinant proteins

The biotin ligation site (bls)-TSC2, Flag-tagged PP2A subunits, TSC2-Flag, and GFP-Flag were synthesized using a Disulfide Bond PLUS Expression kit (#SFS-EDX-DB; CellFree Science). Biotinylated bls-TSC2 (1 μl) and 1 μl each of the Flag-PP2A subunits were mixed in 18 μl of reaction buffer (100 mM Tris–HCl, pH 8.0, 1 mg/ml BSA, 100 mM NaCl, and 0.01% Tween-20). The TSC2-PP2A subunit mixture was then mixed with 10 μl of the detection mixture (containing 1 mg/ml BSA, 100 mM NaCl, 0.01% Tween-20, 0.03 μl of anti-Flag mouse monoclonal [M2] antibody [F3165; Sigma-Aldrich], 0.1 μl protein A acceptor beads [#6760137; PerkinElmer], and 0.1 μl streptavidin donor beads [#6760002; PerkinElmer]). After incubation in a 1/2 AreaPlate-96 shallow well microplate (#6005569; PerkinElmer) for 24 h at 25°C

under shaded conditions, the luminescence signals of each well were detected using an EnSpire plate reader (PerkinElmer). The EnSpire plate reader detected light between 520 and 620 nm at an excitation wavelength of 680 nm.

### Comprehensive screening for proteins on lysosomal surface

HEK293 cells stably expressing LAMP1-AGIA-AirID in 10-cm dishes were treated with 50 $\mu$M D-biotin (04822-91; Nacalai Tesque) in GM containing AAs or AA-free medium (AA (−)) for 10 h. Cells were washed twice with PBS and then lysed in 700 $\mu$l Gdm-TCEP buffer (6 M guanidine HCl, 100 mM Hepes-NaOH, pH 7.5, 10 mM TCEP, and 40 mM chloroacetamide). Three independent lysates were pooled in one tube and then divided into three tubes of 700 $\mu$l each. The cell lysates in Gdm-TCEP buffer were dissolved by heating and sonication and then centrifuged at 20,000$g$ for 15 min at 4°C. The supernatants were recovered, and proteins were purified by methanol–chloroform precipitation and solubilized using PTS buffer (12 mM SDC, 12 mM SLS, 100 mM Tris–HCl, pH 8.0). The protein solution was diluted fivefold with 100 mM Tris–HCl, pH 8.0, and digested with trypsin (MS grade, Thermo Fisher Scientific) at 37°C overnight. The peptide solutions were then diluted twofold with TBS (50 mM Tris–HCl, pH 7.5, 150 mM NaCl). Biotinylated peptides were captured on a 15 $\mu$l slurry of MagCapture HP Tamavidin 2-REV magnetic beads (#133-18611; Fujifilm Wako) after 3 h at 4°C. After washing five times with TBS, the biotinylated peptides were eluted twice with 100 $\mu$l of 1 mM biotin in TBS for 15 min at 37°C. The combined eluates were desalted using GL-Tip SDB (#7820-11200; GL Sciences), evaporated in a SpeedVac concentrator (Thermo Fisher Scientific), and redissolved in 0.1% TFA and 3% acetonitrile. LC-MS/MS analysis of the resultant peptides was performed on an EASY-nLC 1,200 ultra-high-performance liquid chromatography connected to an Orbitrap Fusion mass spectrometer using a nanoelectrospray ion source (Thermo Fisher Scientific). Raw data were directly analyzed against the Swiss-Prot database, restricted to *Homo sapiens*, using Proteome Discoverer version 2.5 (Thermo Fisher Scientific) with the Sequest HT search engine.

### Statistical analysis

All experiments were repeated independently at least three times with consistent results. Data are expressed as the mean ± SEM. The statistical significance of the difference between mean values was analyzed by one-way ANOVA with Tukey's multiple comparison test using GraphPad Prism 10.4.2 software (GraphPad, Inc.).

## Data Availability

Mass spectrometry proteomics data have been deposited in the ProteomeXchange Consortium via the jPOST repository with the dataset identifier PXD042413 (JPST002164). Structural data for the human TSC (TSC1-TSC2-TBC1D7) complex (PDB ID: 7DL2) and the Ragulator/LAMTOR-Rag complex (PDB ID: 5X6V) were obtained from the Protein Data Bank. The source data are provided in this study.

### Code availability

All the mathematical scripts used in the mathematical simulation in this study are available upon request.

## Supplementary Information

## Acknowledgements

This work was supported by a Grant-in-Aid for Scientific Research from the Japan Society for the Promotion of Science (JPSP) (21K06147 and 24K11046 to T Nakamura; 20K06623 to S Nada; 23K27635 to J Masumoto; 16H06576 to T Suzuki; 19H03504 and 19H04962 to M Okada); by JST CREST (JPMJCR2022) to M Takekawa and T Suzuki; by AMED BINDS (22ama121010j0001 and 23ama121010j0002) to T Sawasaki; by grants from Takeda Science Foundation, Mochida Memorial Foundation for Medical and Pharmaceutical Research, and the Princess Takamatsu Cancer Research Fund to T Nakamura; and by the JSPS Core-to-Core Program, A. Advanced Research Networks, to T Nakamura and T Suzuki. We thank Dr. Noriko Tokai-Nishizumi (The University of Tokyo) for technical support related to microscopy, Dr. Tatsuki Mori (Musashino University) for technical assistance with computational simulation, Dr. Hiroyuki Takeda (Ehime University) for providing cDNAs from the Ehime Kazusa Human Protein Array Library, and Dr. Naoe Kaneko (Ehime University) for technical advice regarding the AlphaScreen-based biochemical assay. We also thank Editage (www.editage.com) for English-language editing.

### Author Contributions

T Nakamura: conceptualization, formal analysis, supervision, funding acquisition, visualization, project administration, and writing—original draft, review, and editing.
S Nada: conceptualization, resources, formal analysis, funding acquisition, and investigation.
M Matsumoto: investigation and methodology.
N Loling Othman: investigation and methodology.
H Kosako: investigation, methodology, and resources.
K Ikeda: investigation and methodology.
N Koshikawa: investigation and methodology.
J Masumoto: resources, funding acquisition, and writing—review and editing.
T Sawasaki: resources, funding acquisition, and writing—review and editing.
M Takekawa: resources, funding acquisition, and writing—review and editing.
T Suzuki: conceptualization, funding acquisition, investigation, and writing—review and editing.
M Okada: conceptualization, resources, funding acquisition, and writing—review and editing.

### Conflict of Interest Statement

The authors declare that they have no conflict of interest.

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
