## [Reviewer comments · Life Science Alliance]

Life Science Alliance

Amino acid-dependent TSC2 dephosphorylation by lysosome-PP2A regulates mTORC1 signaling transduction

Takanori Nakamura, Shigeyuki Nada, Masaki Matsumoto, Nuha Loling Othman, Hidetaka Kosako, Kazuki Ikeda, Naohiko Koshikawa, Junya Masumoto, Tatsuya Sawasaki, Mutsuhiro Takekawa, Takashi Suzuki, and Masato Okada

DOI: <https://doi.org/10.26508/lsa.202503206>

Corresponding author(s): Takanori Nakamura, Ehime University and Masato Okada, Osaka University, Research Institute for Microbial Diseases

Review Timeline:

Submission Date:	2025-01-07
Editorial Decision:	2025-01-08
Revision Received:	2025-06-30
Editorial Decision:	2025-07-29
Revision Received:	2025-08-11
Accepted:	2025-08-11

Scientific Editor: Tim Fessenden

Transaction Report:

Please note that the manuscript was previously reviewed at another journal and the reports were taken into account in the decision-making process at *Life Science Alliance*.

Referee #1 Review

Report for Author:

In this study, Nakamura et al. investigate the re-activation of mTORC1 in response to growth factor (GF) and amino acid (AA) stimulation, focusing primarily on the regulation of TSC2 phosphorylation by AKT. Using cell culture models and mathematical modelling, they show that, in cells starved for both GF and AA, Insulin stimulation alone (in the absence of AA) drives the AKT-mediated phosphorylation of TSC2 at T1462 less efficiently, compared to stimulation with both Insulin and AA. Based on this observation, the authors conclude that the GF and AA signaling branches upstream of TSC2-mTORC1 do not work independently, and propose that a lysosomal PP2A fraction is responsible for the enhanced dephosphorylation of TSC2 upon AA starvation.

Although the regulation of TSC2 phosphorylation by PP2A [or by another okadaic acid (OA)-sensitive phosphatase] is potentially interesting, the manuscript suffers from a number of conceptual, technical and study design-related issues that greatly limit my enthusiasm. For instance, the existence of cross-talk between GF and AA signaling upstream of mTORC1 is not novel. Furthermore, the role of PP2A in the regulation of TSC2 localization and mTORC1 activation is not clear and not properly assessed. Finally, a number of false statements and claims that are not supported by the data shown in this study or by the relevant literature, are present in the manuscript. Specific comments are listed below.

Major comments

1) Throughout the manuscript (including in the abstract), the authors refer to two independent, parallel signaling axes that converge on mTORC1, namely GF signaling and AA signaling. However, several studies in the recent years have clearly demonstrated that intense interplay exists between these two signaling branches. Therefore, it is pretty clear that these two "axes" are, in fact, part of an intricate signaling network. Some examples are the phosphorylation of RagC (classically considered an AA signaling component) by mTORC1 in response to GF stimulation (PMID: 30552228) or the Rag-dependent recruitment of TSC (classically considered a GF signaling component) to lysosomes in response to AA starvation (PMID: 26742086, 24529380). Therefore, the identification of a point of cross-talk between the GF and AA signaling branches is neither surprising nor novel (although the authors' narrative presents it as such). Ultimately, this notion about the extensive cross-talk between the two branches is also confirmed in the present study, as the data indicate that TSC2 phosphorylation by AKT is influenced by the state of AA availability. The existence of cross-talk between GF and AA signaling in the mTORC1 network should be properly acknowledged in the manuscript (abstract, introduction and discussion) and the narrative of the manuscript to be adjusted accordingly, to better reflect the relevant literature.

This is an important point also because the two branches of "AA-sensing and GF-sensing" were considered by definition distinct and independent by the authors to generate the initial mathematical model of temporal mTORC1 regulation, therefore this false assumption likely influences the outcomes of the model in various ways.

2) The role of PP2A (or of another OA-sensitive phosphatase) in AA signaling, TSC localization or mTORC1 activation is not clear.

a) Fig. 4C: The authors claim that AA removal prevents the complete TSC2 phosphorylation by AKT because it activates an OA-sensitive phosphatase or promotes the interaction between PP2A and TSC2 on lysosomes. However, OA treatment increases p-TSC2 further both in cells stimulated with Ins+AA and in cells stimulated with Ins alone. These data would go against a model of PP2A regulation downstream of AA signaling. If that was true, the pharmacological inhibition of PP2A should be epistatic to AA starvation, and should increase TSC2 phosphorylation to the levels of Ins+AA when applied to cells treated with Ins alone.

b) Fig. 4C: Furthermore, is OA sufficient to increase TSC2 phosphorylation, also in the absence of both Ins and AA?

c) Fig. 4C: The samples shown in lanes 1-3 and 8-10 (and the additional controls as described above) should be run side-by-side in the same gel to allow for direct comparisons.

d) Fig. 4E,F: In line with the previous points, how does OA treatment influence lysosomal localization of TSC2 in "-AA -Ins", "+AA -Ins", and "+AA +Ins" treated cells? How does this correlate with TSC2 phosphorylation at T1462?

e) Fig. 4E,F: Additionally, it is hard to explain why the Pearson's correlation coefficient values approach 1.0 in the

starved samples. Such values would indicate near-perfect colocalization of TSC2 with LAMP1, which is definitely not the case here given the widespread distribution pattern of TSC2 in cells. How do the authors explain this?

f) Fig. 4E: It is advisable that the authors also show single channels for TSC2 and LAMP1 in microscopy panels to allow for a proper assessment of localization of each protein in cells. Showing only the merged image to indicate colocalization can be misleading, as the colors of overlapping and non-overlapping pixels can also be affected by relative signal intensities. This point is not specific to this panel, but actually applies to all colocalization analyses (for TSC2, mTOR, AKT, etc.)

g) Fig. 4C: Although OA treatment causes dramatically higher TSC2 phosphorylation, both in the presence and in the absence of AA, this effect is not translated to changes in mTORC1 activation (as S6K and S6 phosphorylation is similar between OA-treated and -untreated samples). This indicates that, while PP2A likely plays an important role in the phosphorylation of TSC2 at T1462, the importance and relevance of this mode of regulation in mTORC1 signaling remains rather unclear.

3) Is the (de)phosphorylation effect specific for the T1462 site of AT on TSC2, or it works the same also for the other AKT-dependent phospho-sites? As good phospho-specific antibodies exist for a few of them, it should be straight forward to test this.

4) Fig. 3B: The panel does not really offer any insight as it doesn't show the actual position of T1462 on TSC2. Even if it did, what is the intended purpose of showing the already published TSC1-TSC2 structure? How does this add to the study? Similarly, for Fig. EV5: The structure of the Rag-LAMTOR complex has been resolved in different studies, and this panel serves no purpose in the context of this manuscript.

5) Fig. EV4A: The assay used to indicate interactions between TSC2 and PP2A is not sufficient to conclusively support binding between these proteins, and the observation would need to be confirmed also by independent methods like co-IPs. Importantly, if TSC2 alone gives values around 3000, how do the authors explain the even lower values observed for some of the PP2A subunits? One would expect that the signal would be either equal to TSC2 alone (when there is no interaction) or higher (when there is an interaction with a PP2A subunit).

6) Important controls or quantifications are missing from a number of experiments.

a) Fig. 5A,B and EV6: Quantification of colocalization is needed to assess potential changes in PP2A localization to lysosomes in response to the various treatments and in the different genotypes. Also, the presence of a fraction of PP2A on lysosomes should be confirmed independently by a different method that does not depend on microscopy, like lyso-IP experiments.

b) Fig. 5C: Quantifications of p-TSC2 signals over multiple independent replicate experiments, including appropriate statistical analyses, are required to support the authors' claims. In contrast to their statement in the text, the insulin-induced TSC2 phosphorylation in p18-KO cells does not resemble TSC2 phosphorylation upon double +AA +Ins treatment in p18Rev cells, but is rather enhanced further and also starts from increased levels even in the absence of Ins in starved cells.

c) Fig. EV1A: Appropriate controls for the TSC2 KO lines are missing. The p18Rev cells are reconstituted MEFs re-expressing p18 in a p18 KO background, that are not the right isogenic control for the TSC2 KO clones. Also, it is not clear from the figure legend or the methods if the TSC2 KO clones are also MEFs or something else. Finally, how do the authors explain the phosphorylation on AKT-S473, given that TSC2 is known to be required for proper mTORC2 activity towards AKT (PMID: 18411301)?

d) Accordingly, the choice of p18Rev MEFs for most experiments is not justified. Using p18 KO cells that were later reconstituted by re-expressing p18 is not the same as using parental (control) MEFs.

e) Fig. EV4D: Controls for specificity of the AKT signal in microscopy are missing.

f) Fig. 5A,B: Were the samples in panels 5A and 5B processed and imaged side-by-side? The fact that the two genotypes are shown in separate panels implies that these are two separate experiments, which would not allow for direct comparisons. Can the authors please clarify this point, and provide quantifications of colocalization from experiments comparing TSC2 localization to lysosomes in p18-control and -deficient cells?

g) Fig. 4C: Why are total S6 levels substantially lower in the samples where p-S6 is high?

7) Unfortunately, a number of unsupported claims and false statements is present in the manuscript.

a) The authors state: "complete mTORC1 activation required signaling induction of both the Ragulator-Rag and AKT-TSC1/2-Rheb axes" however no data investigating the contribution of any of these signaling components are shown. This claim is solely based on the activation of mTORC1 in response to treatment with AAs or Ins. Appropriate KO models would be required to make this point.

- b) In the introduction, the authors state: "In response to stimuli such as [...] nutrient amino acids (AAs), mTORC2 localizes to the plasma membrane". What is the evidence from the literature or this manuscript for AA regulating mTORC2 localization to the plasma surface?
- c) In the introduction, the authors state: "depending on the availability of GFs, hormones, nutrient AAs, or energy, mTORC1 localizes to the lysosomal membrane". This statement is false, as mTORC1 localization to lysosomes was shown to be controlled by AAs and glucose, but not energy or GFs.
- d) The authors state: "the precise integration of numerous signals and their relative impact on mTORC1 activity remains poorly understood". However, this is not correct as the interplay of AAs and GFs on TSC localization and mTORC1 inactivation has been studied previously (PMID: 24529379, 26868506).
- e) In the discussion, the authors refer to "a decrease in the lysosomal localization of active RhebGTP and mTORC1 during AA deprivation". Which are the data supporting changes in Rheb localization upon starvation?

8) Fig. 2A: In the model, do the authors imply that GDP-bound Rheb is not farnesylated?

9) The logic behind the mathematical equations that compose the models is completely unclear and should be explained in sufficient detail in the text. Most importantly, it is not clear how the mathematical modeling has added anything to this study. The authors could have already worked on a model that involves an additional layer on top of the AKT-mediated phosphorylation of TSC2, solely based on the immunoblotting data shown in Fig. 3A.

10) It is advisable to refer to the "Ragulator" complex as "LAMTOR complex" (or at least as "LAMTOR/Ragulator complex") mainly for two reasons: 1) historical - this is the original name of this complex; 2) the exclusive use of the term 'Ragulator' can be misleading as it restricts the description of its function to the regulation of the Rags. This ignores completely additional molecular functions of this protein complex, such as the regulation of MAPK signaling on the lysosomal surface, as was previously demonstrated by the work of Lukas Huber and others.

11) The discussion section lacks coherence, and does not add much to the manuscript.

Minor comments

12) Using page and line numbers would greatly assist the reviewer's work.

13) Vertical dotted lines are present in many of the immunoblot panels, separating the different conditions. Does this indicate splicing of lanes that were run on separate gels? Or were all samples of each blot run side-by-side on the same gel? Can the authors please clarify?

14) Introduction: The term "protein translation" is not correct. It is either "mRNA translation" or "protein synthesis".

15) The authors call the pattern of S6K phosphorylation "slow", however this happens within minutes after AA+Ins stimulation.

16) In the introduction, the authors fail to acknowledge that mTORC1 phosphorylates 4E-BP1 also at Ser65.

Referee #2 Review

Report for Author:

In this MS, the authors report the identification of a novel control of the regulation of mTORC1 activation that links nutrient and growth-factor signaling. More specifically, the authors identify a mechanism of de-activation of the GTPase Rheb by its GTPase activating protein TSC2 via PPA-dependent dephosphorylation (and activation) of TSC2 at the lysosomal surface. The authors support their conclusions on biochemical and signal transduction experiments, immunofluorescence, and mathematical modelling of the kinetics of phosphorylation of the regulatory components of the growth factor signaling cascade. While the conclusions are provocative, the MS seems to be thin in experimental support, with a mathematical model that has little value and an important fraction of experiments with results that are already known in the field. Below are some specific comments.

-The mathematical model is of very little value, and the model does not seem to suit the experimental data provided by many, including the figures in the MS. Moreover, the model is so simple, aside of multiple equations, that makes such modeling rather superficial, and more so, of little support to the specific concept to be raised (TSC2 dephosphorylation in AA (-) conditions).

-in line with the previous comment, it is unclear why the authors claim that "Mathematical simulations revealed that compared to the model without crosstalk (Fig. 3C), the simulation values of P-AKT (P-T308 and P-S473) and P-TSC2 (P-T1462) obtained from the model with crosstalk fit relatively well (better) with the experimental values (Fig. 4B)". What is the numerical basis for this statement?

-All experiments have been done with stimulation of PI3K and mTORC1 activity, and not on the kinetics following deprivation of insulin and nutrients. Because dephosphorylation is the phenomenon to be demonstrated, these deactivation experiments are critical to demonstrate the phenomenon and are surprisingly absent in the MS.

-The use of the p18Rev cells is problematic. The genetic makeup of these cells makes them prone to multiple selective pressures that result in signal transduction adaptation to the lack of p18, and to its reconstitution. Several labs have shown that mTORC1 activation and inhibition heavily impacts on GF signaling by multiple (known and unknown) mechanisms, but all in all, limits the potential physiological relevance of such phenomenon. Several additional non-transformed human and mouse cells are available for signal transduction experiments. Finally, is this phenomenon observed in cell lines with constitutive PI3K-Akt signaling?

-In addition, nutrient and growth factor signaling are known to functionally regulate one another. For example, work by C Thompson has shown that GF signaling is critical for nutrient uptake. Activation of nutrient signaling by AA esters would provide additional support to the model.

-According to the model, a prediction is that locking TSC2 to the cytosolic lysosomal surface would result in increased PP2A-dependent dephosphorylation of TSC2, and presumably independently of AA conditions and Ragulator.

-The authors intended to manipulate the nutrient signaling cascade upstream of mTORC1 to assess its impact on PP2A and TSC2, gain- and loss-of-function variants of the Rag GTPases (Rag-BGTP/RagC-GDP, and vice versa) constitute a reliable system to test this idea.

-Experiments with phospho-deficient and phospho-mimetic TSC2 mutants are required to demonstrate the relevance of this site on the regulation under AA starvation.

-The IF data is at odds with plenty of literature that has shown that localization of TSC at the lysosomal surface depends on GF signaling, and not on nutrient signaling, and the alternative control of TSC2 by amino acids is controversial. With the limited quality of images and the selected regions for PCA quantification, the conclusions are far from supported and additional experimental robustness would be ideal. This particularly worrisome for the IF on Figure 5, where the PP2A stains and TSC2 stains are uninterpretable and visually weak, and in general for p18KO and rev cells, where lysosomal markers do not seem to have a lysosomal pattern, very evident in Figure 4E.

-The specificity of the control of phosphorylation on TSC2 is at least surprising, given the well-established promiscuity of phosphatases. Are other targets of Akt (presumably not dynamically localizing to the lysosomal surface) also affected?

-Why does Figure 4c lack the +AA + Ins conditions?

Referee #3 Review

Report for Author:

summary

The manuscript investigates the synergistic activation of mTORC1 by amino acids (AA) and insulin, focusing on the relief of inhibitory TSC2 phosphorylation upon AA stimulation. Using genetic knockouts, colocalization analysis, pharmacological approaches, and meticulously designed western blot experiments, the authors attribute this relief to PP2A-mediated dephosphorylation facilitated by lysosomal localization after AA stimulation. Additionally, the authors develop a mathematical model to formalize their findings and compare simulations against experimental data.

The paper is well-written and the experimental work is thorough, yielding plausible and interesting findings. However, the mathematical model feels somewhat disconnected from the experimental narrative. It (i) incorporates numerous molecular mechanisms not emphasized in the main manuscript, (ii) omits key mechanisms identified experimentally, and (iii) does not align well with the experimental data. This gives the impression that the experimental and modeling efforts were developed in isolation.

In conclusion, the work shows potential, but its impact would be significantly enhanced by better integrating the experimental and mathematical components or removing the mathematical model from the manuscript.

major concerns

Model-Experiment Comparison

The authors claim improved agreement between simulations and experiments after adding crosstalk via PPase (Fig. 3C vs. Fig. 4). However, this improvement is not evident upon visual inspection. The authors should:

- Provide a quantitative comparison using metrics such as explained variance, R^2 , or correlation.
- Estimate model parameters directly from their experimental data, rather than relying on literature-based or arbitrary values, which may not suit both models equally.

Simplification in the Model

Experimentally, the authors show that PP2A dephosphorylates TSC2 at the lysosome, and AA starvation induces lysosomal localization of TSC2. However, the model simplifies this mechanism to AA-dependent sequestration of PP2A. This stark simplification contrasts with the complexity of other model components. The authors should either:

- Simplify the rest of the model for consistency.
- Add more detail to the crosstalk mechanism in line with the experimental findings.

Insufficient Experimental Validation of the Model

The model is only validated against AA+ and Ins+ conditions, despite the manuscript presenting diverse experimental datasets (e.g., Ins dose-response measurements, pharmacological inhibition, and knockout experiments). Without comparing model simulations to all relevant experimental conditions, the claim that the model enhances quantitative understanding is unsubstantiated. The model does not appear to yield insights beyond what could be inferred from the data alone, contrary to the claims in the discussion.

minor concerns

- Clarify the timepoints post-AA/Ins stimulation for the immunofluorescence experiments.
- Update the reference in "Western blot analysis using phospho-TSC2- and TSC2-specific antibodies (Fig. EV1A, B)" to refer to Fig. 1.
- Revise the use of "reaction-diffusion," which typically refers to partial differential equations, as its usage here may be misleading.

additional suggestions

none

January 8, 2025

Re: Life Science Alliance manuscript #LSA-2025-03206-T

Dr. Takanori Nakamura
Proteo-Science Center, Ehime University
Department of Pathology
454 Shitsukawa, Toon
Toon, Ehime 791-0295
Japan

Dear Dr. Nakamura,

Thank you for submitting your manuscript entitled "Amino acid-dependent TSC2 dephosphorylation by lysosome-PP2A regulates mTORC1 signaling transduction" to Life Science Alliance. We invite you to submit a revised manuscript addressing the Reviewer comments.

Thank you for this interesting contribution to Life Science Alliance. We are looking forward to receiving your revised manuscript.

Sincerely,

Eric Sawey, PhD
Executive Editor
Life Science Alliance
<http://www.lsa-journal.org>

B. MANUSCRIPT ORGANIZATION AND FORMATTING:

Re: *Life Science Alliance* manuscript #LSA-2025-03206-T
Nakamura et al.

Point-by-point Responses to the Referees' Comments

We sincerely thank all the referees for their thoughtful comments on our manuscript. Their constructive suggestions were instrumental in guiding us to generate new data that further strengthened our manuscript. We are deeply grateful for their insightful feedback.

Referee #1:

1) Throughout the manuscript (including in the abstract), the authors refer to two independent, parallel signaling axes that converge on mTORC1, namely GF signaling and AA signaling. However, several studies in the recent years have clearly demonstrated that intense interplay exists between these two signaling branches. Therefore, it is pretty clear that these two "axes" are, in fact, part of an intricate signaling network. Some examples are the phosphorylation of RagC (classically considered an AA signaling component) by mTORC1 in response to GF stimulation (PMID: 30552228) or the Rag-dependent recruitment of TSC (classically considered a GF signaling component) to lysosomes in response to AA starvation (PMID: 26742086, 24529380). Therefore, the identification of a point of cross-talk between the GF and AA signaling branches is neither surprising nor novel (although the authors' narrative presents it as such). Ultimately, this notion about the extensive cross-talk between the two branches is also confirmed in the present study, as the data indicate that TSC2 phosphorylation by AKT is influenced by the state of AA availability. The existence of crosstalk between GF and AA signaling in the mTORC1 network should be properly acknowledged in the manuscript (abstract, introduction and discussion) and the narrative of the manuscript to be adjusted accordingly, to better reflect the relevant literature.

This is an important point also because the two branches of "AA-sensing and GF-sensing" were considered by definition distinct and independent by the authors to generate the initial mathematical model of temporal mTORC1 regulation, therefore this

false assumption likely influences the outcomes of the model in various ways.

Response:

We appreciate the referee's insightful comment. We have cited as many papers on crosstalk as possible in the Introduction and Discussion sections, including the report mentioned by referee-#1 (PMID: 30552228). In addition, the other reports noted (PMID: 26742086, 24529380) had already been cited. To clearly acknowledge the existence of interplay between GF and AA signaling in the mTORC1 network, we have added the following sentence to the manuscript:

“Although several studies in the recent years have reported that interplay exists between the AA and GF signaling branches (Yang *et al.*, 2019; Demetriades *et al.*, 2014; Carroll *et al.*, 2016), to the best of our knowledge, an integrated mathematical model of mTORC1 regulation through two distinct AA- and GF-sensing axes has not yet been proposed.” (pages 6, lines 136–140).

2) The role of PP2A (or of another OA-sensitive phosphatase) in AA signaling, TSC localization or mTORC1 activation is not clear.

a) Fig. 4D (former Fig. 4C): The authors claim that AA removal prevents the complete TSC2 phosphorylation by AKT because it activates an OA-sensitive phosphatase or promotes the interaction between PP2A and TSC2 on lysosomes. However, OA treatment increases p-TSC2 further both in cells stimulated with Ins+AA and in cells stimulated with Ins alone. These data would go against a model of PP2A regulation downstream of AA signaling. If that was true, the pharmacological inhibition of PP2A should be epistatic to AA starvation, and should increase TSC2 phosphorylation to the levels of Ins+AA when applied to cells treated with Ins alone.

Response:

In accordance with the referee's request, we performed the suggested western blotting experiments within the same membrane, and the data have been added as the new Fig. 4E. These data demonstrates that Ins-plus-OA treatment (PP2A inhibition) increases TSC2 phosphorylation levels to those observed in cells treated with AA and Ins.

b) Fig. 4D (former Fig. 4C): Furthermore, is OA sufficient to increase TSC2 phosphorylation, also in the absence of both Ins and AA?

Response:

We appreciate the referee's insightful comment. We performed the suggested experiments, and the data have been added as the new Fig. 4E. This figure demonstrates that OA treatment induces a moderate (but insufficient) increase in TSC2 phosphorylation under AA(-) Ins(-) conditions, consistent with the result that PP2A knockdown (for 2 days) increases TSC2 phosphorylation under AA(-) Ins(-) conditions, as shown in Fig. 4F.

c) Fig. 4D (former Fig. 4C): The samples shown in lanes 1-3 and 8-10 (and the additional controls as described above) should be run side-by-side in the same gel to allow for direct comparisons.

Response:

In accordance with the referee's request, we performed the suggested experiments, and the data have been added as the new Fig. 4E. These data indicates that Ins-plus-OA treatment increases TSC2 phosphorylation levels to those observed in cells treated with AA and Ins.

d) Fig.4E,F(the new Fig.4G,H): In line with the previous points, how does OA treatment influence lysosomal localization of TSC2 in "-AA -Ins", "+AA -Ins", and "+AA +Ins" treated cells? How does this correlate with TSC2 phosphorylation at T1462?

Response:

We appreciate the referee's insightful comment. We performed the suggested immunostaining experiments, and the data have been added as the new Fig. 4G and H. These data demonstrate that OA treatment does not influence lysosomal TSC2 localization. In addition, we performed the suggested immunoblot experiments monitoring the phosphorylation levels of TSC2 at T1462 under AA(-) Ins(-), AA(+) Ins(-), AA(-) Ins(+), and AA(+) Ins(+) conditions, or together with OA. The data have been added as the new Fig. 4E. The immunoblot data demonstrate that although OA treatment induces a slight increase in TSC2 phosphorylation under AA(-) Ins(-) conditions, OA treatment induces the largest increase in TSC2 phosphorylation levels under the AA(-) Ins(+) conditions, reaching the TSC2 phosphorylation levels comparable to those observed in cells treated with AA(+) Ins(+).

e) Fig.4E,F (the new Fig.4G,H): Additionally, it is hard to explain why the Pearson's correlation coefficient values approach 1.0 in the starved samples. Such values would indicate near-perfect colocalization of TSC2 with LAMP1, which is definitely not the case here given the widespread distribution pattern of TSC2 in cells. How do the authors explain this?

Response:

We appreciate the referee's insightful comment. The Pearson correlation coefficient for Fig. 4G (previously Fig. 4E) had adopted the no-threshold value, and the overall value therefore was high. Consequently, we therefore recalculated the value above the threshold, and the data have been added as the new Fig. 4H.

f) Fig.4E (the new Fig.4G): It is advisable that the authors also show single channels for TSC2 and LAMP1 in microscopy panels to allow for a proper assessment of localization of each protein in cells. Showing only the merged image to indicate colocalization can be misleading, as the colors of overlapping and non-overlapping pixels can also be affected by relative signal intensities. This point is not specific to this panel, but actually applies to all colocalization analyses (for TSC2, mTOR, AKT, etc.)

Response:

We appreciate the referee's thoughtful comment. We have added single-channel images for TSC2, mTOR, AKT, and LAMP1 in the new Fig. 4G, 5B, S1C, D, S4F, I, J, and S5C.

g) Fig. 4D (former Fig. 4C): Although OA treatment causes dramatically higher TSC2 phosphorylation, both in the presence and in the absence of AA, this effect is not translated to changes in mTORC1 activation (as S6K and S6 phosphorylation is similar between OA-treated and -untreated samples). This indicates that, while PP2A likely plays an important role in the phosphorylation of TSC2 at T1462, the importance and relevance of this mode of regulation in mTORC1 signaling remains rather unclear.

Response:

Full activation of mTORC1 requires the involvement of the AA and Ins signaling pathways: 1) AA-stimulated recruitment of mTORC1 to lysosomes via the Ragulator/LAMTOR-Rag complex, and 2) Ins-stimulated mTORC1 autophosphorylation via binding to Rheb^{GTP} on the lysosomal surface (PMID:

22177716, 304253336, 31937935).

Thus, upon AA(+) Ins(-) stimulation, mTORC1 is recruited to lysosomes by the Ragulator/LAMTOR-Rag complex, but its autophosphorylation on the lysosomal surface assisted by Rheb^{GTP} is not induced, resulting in insufficient mTORC1 activation. Conversely, during AA(-) Ins(+) stimulation, Ins stimulation promotes the accumulation of active Rheb^{GTP} on the lysosomal surface, but does not induce mTORC1 autophosphorylation on the lysosomes because the recruitment of mTORC1 to the lysosomes (mediated by the Ragulator/LAMTOR-Rag complex) does not occur, again leading to insufficient mTORC1 activation. As shown in Fig. 4D and E, OA treatment under AA(-) Ins(+) conditions markedly increased TSC2 phosphorylation at T1462 compared to AA(-) Ins(+) conditions without OA, but phosphorylation levels of S6K at T389 and S6 at S235/S236 were only slightly elevated. This is most likely because, under AA deprivation, mTORC1 is not localized to lysosomes and therefore cannot bind Rheb^{GTP} on the lysosomal surface, thereby resulting in insufficient mTORC1 activation even under AA(-) Ins(+) OA(+) conditions.

3) Is the (de)phosphorylation effect specific for the T1462 site of AKT on TSC2, or it works the same also for the other AKT-dependent phospho-sites? As good phospho-specific antibodies exist for a few of them, it should be straight forward to test this.

Response:

In accordance with the referee's request, we performed the suggested experiments using a rabbit polyclonal antibody (CST, #3615) against phospho-TSC2 at S939, another major AKT phosphorylation site (PMID:12150915). The data have been added as the new Fig. 1B–D, S1A and B. Although the sensitivity and specificity of this antibody are lower compared to the anti-phospho-TSC2(T1462) monoclonal antibody (CST, #3617), the data demonstrated a similar pattern of AKT-dependent TSC2 phosphorylation at S939 and T1462 after stimulation with AA(+) Ins(+) (Fig. 1B), AA(-) Ins(+) (Fig. 1C), and AA(+) Ins(-) (Fig. 1D).

4) Fig. 3B: The panel does not really offer any insight as it doesn't show the actual position of T1462 on TSC2. Even if it did, what is the intended purpose of showing the already published TSC1-TSC2 structure? How does this add to the study? Similarly, for

Fig. EV5 (the new Fig.S6): The structure of the Rag-LAMTOR complex has been resolved in different studies, and this panel serves no purpose in the context of this manuscript.

Response:

We appreciate the referee's thoughtful comment. The 3D structures of the T1462 and S939 residues have not yet been deciphered, even when using AlphaFold2 (<https://alphafold.ebi.ac.uk>). However, we believe that positional information for the T1462, S939, and S664 residues in the TSC1–TSC2–TBC1D7 complex is necessary to understand the physiological function of TSC2, even if only approximately. Indeed, the analysis of the conformation of the TSC1–TSC2–TBC1D7 complex (PDB: 7DL2) indicates that the T1462 residue in the dimerization domain (DD) of TSC2 is likely located at or near the interaction surface with the TSC1 coiled-coil (CC) structure. The S939 residue in the Heat domain, which is phosphorylated by AKT, is positioned opposite to the TSC1(CC) binding surface, whereas the S664 residue, which is phosphorylated by ERK, is predicted to be at the interaction surface with the TSC1-CC domain, distinct from the TSC2(DD)–TSC1(CC) interaction interface. Therefore, in addition to the T1462 residue, we have included the approximate positions of the S939 and S664 residues in the human TSC1–TSC2–TBC1D7 complex (PDB: 7DL2) in Fig. 3B. Furthermore, we performed proximity-dependent biotinylation of lysosomal proteins (Lyso-BioID) (Fig. 5A–D) and identified phosphatases, such as PP2A-B56 subunits (B56 γ and B56 δ). Notably, we found that eight PP2A-B56 binding motifs ([L/M/F/I]-x-x-[I/L/V]-x-E-x) are conserved within TSC2 (Fig. 5E). Structural data for seven of the eight PP2A-B56 binding motifs have been solved, and the amino acids required for binding to PP2A-B56 in TSC2 (e.g., M, L, I, V, and E) are highlighted in red. This information has been added in the new Fig. 5F.

The structure of the Ragulator/LAMTOR-Rag complex (PDB: 5X6V) in Fig. S6 is the structure we previously elucidated (Nat. Commun. 2017, PMID: 29158492). This structure of the Ragulator/LAMTOR-Rag complex shows that p18, one of the Ragulator components, interacts not only with each of the other Ragulator/LAMTOR components (p14, MP1, p10, and HBXIP) but also with RagA and RagC. Therefore, to help readers understand that p18 knockout effectively inhibits the function of the Ragulator/LAMTOR-Rag complex, we believe that the structural information (PDB: 5X6V) for the Ragulator/LAMTOR-Rag complex in Fig. S6 is necessary.

5) Fig. EV4B (the new Fig. S4B): The assay used to indicate interactions between TSC2 and PP2A is not sufficient to conclusively support binding between these proteins, and the observation would need to be confirmed also by independent methods like co-IPs. Importantly, if TSC2 alone gives values around 3000, how do the authors explain the even lower values observed for some of the PP2A subunits? One would expect that the signal would be either equal to TSC2 alone (when there is no interaction) or higher (when there is an interaction with a PP2A subunit).

Response:

We apologize for the insufficient explanation. We have changed Fig. S4B (previously Fig. EV4B) to improve its clarity. Since TSC2 is known to form a dimer (PMID: 33307091, 33436626), the leftmost panel (with a value of $\sim 3,000$) in the AlphaScreen data monitors the interaction (dimer formation) between TSC2-Flag and bls-TSC2, which serves as the positive control. Actually, the dimeric formation between Myc-TSC2 and Flag-TSC2 was confirmed by co-IP (Fig. S5C). The green fluorescent protein (GFP)-Flag (second from the left) acts as the negative control in the AlphaScreen assay. As its value was $\sim 1,300$, values $\leq 1,300$ were regarded as the background signals in the assay. Under these conditions, stronger signals were detected for PP2A-C α -Flag, PP2A-C β -Flag, B55 δ -Flag, B56 β -Flag, and B56 δ -Flag than for TSC2-Flag, suggesting particularly strong direct binding of TSC2 to the PP2A subunits. Supporting this, co-IP assay confirmed that TSC2 bound to PP2A in a manner dependent on T1462/T1395 phosphorylation or its phospho-mimetic (Fig. S5D and E). Furthermore, these results are consistent with the observation that eight PP2A-B56 binding sites are conserved in TSC2 (Fig. 5E and F).

6) Important controls or quantifications are missing from a number of experiments.

a) Fig. 5A,B (the new Fig. 6A,B) and EV6 (the new Fig. S7): Quantification of colocalization is needed to assess potential changes in PP2A localization to lysosomes in response to the various treatments and in the different genotypes. Also, the presence of a fraction of PP2A on lysosomes should be confirmed independently by a different method that does not depend on microscopy, like lyso-IP experiments.

Response:

In accordance with the referee's request, we performed the suggested quantification experiments, and the data have been added to the new Fig. 6C, D, S7E and F. These figures demonstrate that, as anticipated, lysosomal TSC2 localization during AA starvation is primarily mediated by the Ragulator/LAMTOR-Rag GTPase (Fig. 6C and S7E), whereas a part of PP2A-A α / β and -C α constantly display lysosomal localization in a manner independent of intracellular AA or Ins levels, or Ragulator/LAMTOR-Rag GTPase (Fig. 6D and S7F).

Furthermore, a CoIP assay was performed using LAMP1-fused Flag-TSC2(1740 aa)-WT and its phospho-deficient (T1395A, identical to T1462A in TSC2-1807 aa) and phospho-mimetic (T1395D and T1395E) mutants. The results demonstrate that, although LAMP1-Flag-TSC2-WT, anchored to the lysosomal membrane, does not bind to PP2A subunits under AA starvation (because it is not phosphorylated), it interacts with PP2A subunits in a TSC2 phosphorylation-dependent manner upon AA(-) Ins(+) stimulation (Fig. S4E). In addition, even under AA starvation, the phospho-mimetic LAMP1-Flag-TSC2-T1395D and -T1395E mutants bind to PP2A subunits, whereas the phospho-deficient TSC2-T1395A does not. Moreover, using LAMP1-AGIA-AirID stably expressing HEK293 cells, we have performed MS-based identification of proteins on the lysosomal surface under AA-supplementation or AA-removal. The data demonstrate that a number of phosphatases, including PP2A-B56 γ and -B56 δ , constantly localize on lysosomal surfaces, regardless of intracellular AA levels (Fig. 5D). These findings further support our conclusion that, during AA starvation, TSC2 is localized to the lysosomes and subjected to a dephosphorylation target for lysosomal PP2A.

b) Fig. 5C (the new Fig. 6E): Quantifications of p-TSC2 signals over multiple independent replicate experiments, including appropriate statistical analyses, are required to support the authors' claims. In contrast to their statement in the text, the insulin-induced TSC2 phosphorylation in p18-KO cells does not resemble TSC2 phosphorylation upon double +AA +Ins treatment in p18Rev cells, but is rather enhanced further and also starts from increased levels even in the absence of Ins in starved cells.

Response:

In accordance with the referee's request, we performed the suggested quantification experiments, and the data have been added to the new Fig. 6E–G and S8. These figures demonstrate that, while there were no significant differences in the phosphorylation level of AKT at T308 between each stimulus (Fig. 6G), the phosphorylation level of TSC2 at T1462 was higher in p18Rev cells stimulated with AA(+) Ins(+) than in those stimulated with AA(–) Ins(+), reaching the level observed in p18-KO cells stimulated with AA(–) Ins(+) (Fig. 6F).

c) Fig. S1A (former Fig. EV1A): Appropriate controls for the TSC2 KO lines are missing. The p18Rev cells are reconstituted MEFs re-expressing p18 in a p18 KO background, that are not the right isogenic control for the TSC2 KO clones. Also, it is not clear from the figure legend or the methods if the TSC2 KO clones are also MEFs or something else. Finally, how do the authors explain the phosphorylation on AKT-S473, given that TSC2 is known to be required for proper mTORC2 activity towards AKT (PMID: 18411301)?

Response:

In Fig. S1A, we used TSC2-KO MEFs, and modified the figure legend as follows: “p18Rev and TSC2-KO MEFs (#1 and #2) were exposed to AA deprivation and subsequent AA and Ins treatment.”

Furthermore, in accordance with the referee's request, we performed the knockdown experiments using siRNAs specific for TSC2. The data have been added to the new Fig. S1B–D. These data demonstrate that TSC2 phosphorylation at T1462 (and S939) was either lost or markedly decreased in TSC2-knockdown cells, confirming the specificity of the anti-phospho-TSC2 (T1462 and S939) antibodies used in this study. TSC2-depletion for 2 days or more also led to a decrease in AKT phosphorylation at T308 and S473 (Fig. S1B), supporting the previous finding that TSC2 is required for proper mTORC2 activity towards AKT (PMID: 18411301). However, it should be noted that this result monitors TSC2-mediated positive feedback regulation of mTORC2-AKT when cells are genetically deleted of TSC2 for more than 2 days, which differs in time scale from our experiments monitoring intracellular mTORC1 signaling within 1 h after AA or Ins addition.

d) Accordingly, the choice of p18Rev MEFs for most experiments is not justified. Using p18KO cells that were later reconstituted by re-expressing p18 is not the same as using parental (control) MEFs.

Response:

In accordance with the referee's request, we performed the experiments using parental (control) MEFs, and the data have been added to the new Figs. S1B and S1C, and S3B. These data confirm that the similar responsivity to AA and Ins was observed in MEFs and p18Rev MEFs. In this sense, we consider p18Rev MEFs to be equivalent to control MEFs in their responsivity to AA and Ins in this study.

e) Fig. S4I (previously Fig. EV4D): Controls for specificity of the AKT signal in microscopy are missing.

Response:

In accordance with the referee's request, we performed the knockdown experiments with siRNAs specific for AKT1 and AKT2. The data have been added in the new Fig. S4G and H. The data demonstrate that AKT1/2-knockdown led to the loss of AKT signal observed via microscopy.

f) Fig. 6A,B (former Fig. 5A,B): Were the samples in panels 5A and 5B processed and imaged side-by-side? The fact that the two genotypes are shown in separate panels implies that these are two separate experiments, which would not allow for direct comparisons. Can the authors please clarify this point, and provide quantifications of colocalization from experiments comparing TSC2 localization to lysosomes in p18-control and -deficient cells?

Response:

In accordance with the referee's request, we performed the quantification of the TSC2, LAMP1, and PP2A-A α / β in Fig. 6A and B, and the data have been added to the new Fig. 6C and D. Under AA-supplementation, TSC2 displayed diffuse cytoplasmic localization, whereas under AA deprivation, it localized to the lysosomal surface independent of Ins stimulation (Fig. 6C). Lysosomal TSC2 localization during AA starvation was primarily mediated by the Ragulator/LAMTOR-Rag GTPase, because the genetic p18 depletion resulted in a marked decrease in TSC2 recruitment to the lysosomal membrane (Fig. 6C). In contrast to TSC2, PP2A-A α / β constantly displayed

some lysosomal localization, regardless of intracellular AA or Ins levels (Fig. 6D).

g) Fig. 4C (the new Fig 4D): Why are total S6 levels substantially lower in the samples where p-S6 is high?

Response:

The anti-phospho antibodies added during the first immunoblotting cover the antigen protein, and therefore, the newly added antibodies are repelled when the same membrane is reblotted. Although we washed membrane with a stripping solution for at least 1 h before the reblotting, the P-S6 antibody binds particularly strongly to the S6 antigen and cannot be completely removed by the stripping solution. Thus, the S6 level appears low at first glance when the phosphorylation is stronger. To avoid the misleading impression that the S6 levels are lower when the phosphorylation level is higher, the contrast of the S6 band was corrected so that the S6 protein levels appear similar in Figs. 3A, 3D, 4D, 4E, and 6E (previously 5C).

7) Unfortunately, a number of unsupported claims and false statements is present in the manuscript.

a) The authors state: "complete mTORC1 activation required signaling induction of both the Ragulator-Rag and AKT-TSC1/2-Rheb axes" however no data investigating the contribution of any of these signaling components are shown. This claim is solely based on the activation of mTORC1 in response to treatment with AAs or Ins.

Appropriate KO models would be required to make this point.

Response:

We appreciate the referee's thoughtful comment. We agree with referee #1's opinion that loss-of-function experiments (KO models) are the best way to confirm whether both the Ragulator/LAMTOR-Rag and AKT-TSC1/2-Rheb axes are required for complete mTORC1 activation. However, there have already been numerous previous studies, including our own work with knockout experiments, demonstrating that both the AKT-TSC1/2-Rheb and Ragulator/LAMTOR-Rag axes are required for mTORC1 activation (PMID:

26449471, 26235620, 24529380, 16959574, 14651849, 36357396, 26742086, 28880149, 35945201, 36097072, 32184389, 32653916, 29158492, 20381137, 39163330). In addition, we have demonstrated that genetic depletion of both p18 and Rheb, key

regulators of the AA- and GF-sensing pathways, results in the complete loss of mTORC1 activation monitored by the phosphorylation levels of P-S6K(T389) and P-S6(S235/S236) (Fig. S9B lanes 1-4). Therefore, we believe that the requirement for the Ragulator-Rag and AKT-TSC1/2-Rheb axes in complete mTORC1 activation has already been conclusively demonstrated by the previous studies, including our own using KO models. Consequently, we do not think that the additional KO experiments suggested by the referee are necessary.

b) In the introduction, the authors state: "In response to stimuli such as [...] nutrient amino acids (AAs), mTORC2 localizes to the plasma membrane". What is the evidence from the literature or this manuscript for AA regulating mTORC2 localization to the plasma surface?

Response:

In accordance with the referee's suggestion, we have revised the sentence in the Introduction section as follows:

"In response to stimuli such as extracellular growth factors (GFs) [e.g., insulin (Ins) and insulin-like GF (IGF)], mTORC2 localizes to the plasma membrane and phosphorylates AKT, also known as protein kinase B (PKB), at S473 (Saxton & Sabatini, 2017; Yang et al, 2015), protein kinase C (PKC) (Saxton & Sabatini, 2017), and serum- and glucocorticoid-induced protein kinase (SGK) (Saxton & Sabatini, 2017), regulating cytoskeletal dynamics and cellular survival." (pages 3-4, lines 77–82).

c) In the introduction, the authors state: "depending on the availability of GFs, hormones, nutrient AAs, or energy, mTORC1 localizes to the lysosomal membrane". This statement is false, as mTORC1 localization to lysosomes was shown to be controlled by AAs and glucose, but not energy or GFs.

Response:

In accordance with the referee's suggestion, we changed the sentence in the Introduction section as follows:

"depending on the availability of GFs, hormones, nutrient AAs, and energy, mTORC1 localizes to the lysosomal membrane in response to AAs or glucose" (page 4, lines 82–84).

d) The authors state: "the precise integration of numerous signals and their relative impact on mTORC1 activity remains poorly understood". However, this is not correct as the interplay of AAs and GFs on TSC localization and mTORC1 inactivation has been studied previously (PMID: 24529379, 26868506).

Response:

In accordance with the referee's suggestion, we changed the sentence in the Introduction section as follows:

"the precise integration of numerous signals and their relative impact on mTORC1 activity is not fully understood" (page 5, lines 129–130).

e) In the discussion, the authors refer to "a decrease in the lysosomal localization of active RhebGTP and mTORC1 during AA deprivation". Which are the data supporting changes in Rheb localization upon starvation?

Response:

In accordance with the referee's suggestion, we changed the sentence in the discussion as follows:

"a decrease in the active Rheb^{GTP} and mTORC1 levels on the lysosomal surface during AA deprivation" (pages 24, lines 611–612).

8) Fig. 2A: In the model, do the authors imply that GDP-bound Rheb is not farnesylated?

Response:

To keep the figure as simple as possible, the farnesylation line of Rheb^{GDP} was not originally included in Fig. 2A. However, in accordance with the referee's advice, we have described the farnesylated linker to Rheb^{GDP} in the new Fig. 2A.

9) The logic behind the mathematical equations that compose the models is completely unclear and should be explained in sufficient detail in the text. Most importantly, it is not clear how the mathematical modeling has added anything to this study. The authors could have already worked on a model that involves an additional layer on top of the

AKT mediated phosphorylation of TSC2, solely based on the immunoblotting data shown in Fig.3A.

Response:

We appreciate the referee's insightful comment. We have described the logic behind the equations in the "Mathematical simulation" subsection of the Methods section. Among the three models we developed (without-crosstalk, with-crosstalk, and compartment models), the compartment model best reproduced the temporal changes of P-AKT, P-TSC2 and P-S6K upon all AA(+) Ins(+), AA(+) Ins(-), AA(-) Ins(+) conditions (Fig. 4C, 7B-E). These data demonstrate that the factors, such as TSC2 dephosphorylation on the lysosomal surface and the cytoplasm-lysosome distribution of TSC2 and mTOR depending on the AA concentration, are required for the fine-tuning of TSC2 phosphorylation levels and mTORC1 activation.

10) It is advisable to refer to the "Ragulator" complex as "LAMTOR complex" (or at least as "LAMTOR/Ragulator complex") mainly for two reasons: 1) historical - this is the original name of this complex; 2) the exclusive use of the term 'Ragulator' can be misleading as it restricts the description of its function to the regulation of the Rags. This ignores completely additional molecular functions of this protein complex, such as the regulation of MAPK signaling on the lysosomal surface, as was previously demonstrated by the work of Lukas Huber and others.

Response:

In accordance with the referee's advice, we have changed Ragulator-Rag to Ragulator/LAMTOR-Rag throughout the text.

11) The discussion section lacks coherence, and does not add much to the manuscript.

Response:

In accordance with the referee's advice, we have modified the Discussion section to enhance its informativeness and effectiveness.

12) Using page and line numbers would greatly assist the reviewer's work.

Response:

In accordance with the referee's advice, we have added page and line numbers to the

manuscript.

13) Vertical dotted lines are present in many of the immunoblot panels, separating the different conditions. Does this indicate splicing of lanes that were run on separate gels? Or were all samples of each blot run side-by-side on the same gel? Can the authors please clarify?

Response:

As already partially shown in Fig. S3, all the immunoblot data with vertical dotted lines in Fig. 3A and 3D show samples that were run side-by-side on the same membrane. The dotted lines were initially added to enhance clarity and understanding, but they have been removed from the revised Fig. 3A and D to avoid potential misunderstandings or conjectures.

14) Introduction: The term "protein translation" is not correct. It is either "mRNA translation" or "protein synthesis".

Response:

In accordance with the referee's advice, we have re-written this sentence as follows (Page 4, lines 82–87):

“In contrast, depending on the availability of GFs, hormones, nutrient amino acids (AAs), and energy, mTORC1 localizes to the lysosomal membrane in response to AAs or glucose and integrates multiple signals to promote protein synthesis and cell growth by phosphorylating its substrates, including eukaryotic initiation factor 4E-binding proteins (4E-BP) at T37/T46/S65/T70 and p70 S6 kinase (S6K) at T389.”

15) The authors call the pattern of S6K phosphorylation "slow", however this happens within minutes after AA+Ins stimulation.

Response:

As far as we could determine in our experiments, AKT was fully phosphorylated within 3 min after AA+Ins stimulation (Fig. S1F). In contrast, S6K phosphorylation was only slightly observed 10 min after AA+Ins treatment and reached its maximum after 20–40 min (Fig. S1F). Based on these observations, we stated that S6K phosphorylation is relatively slow relative to AKT phosphorylation.

16) In the introduction, the authors fail to acknowledge that mTORC1 phosphorylates 4EBP1 also at Ser65.

Response:

In accordance with the referee's advice, we have re-written this sentence as follows (Page 4, lines 82–87):

“In contrast, depending on the availability of GFs, hormones, nutrient AAs, and energy, mTORC1 localizes to the lysosomal membrane in response to AAs or glucose and integrates multiple signals to promote protein synthesis and cell growth by phosphorylating its substrates, including eukaryotic initiation factor 4E-binding proteins (4E-BP) at T37/T46/S65/T70 and p70 S6 kinase (S6K) at T389.”

Referee #2

-The mathematical model is of very little value, and the model does not seem to suit the experimental data provided by many, including the figures in the MS. Moreover, the model is so simple, aside of multiple equations, that makes such modeling rather superficial, and more so, of little support to the specific concept to be raised (TSC2 dephosphorylation in AA (-) conditions).

Response:

We appreciate the referee's insightful comment. In accordance with the referee's advice, we have developed a compartment model by adding as many experimental details as possible, such as lysosomal localization of TSC2 and mTORC1 depending on intracellular AA or Ins levels (Fig. 7A). The mathematical simulation demonstrates that the compartment model best reproduced the temporal changes of P-AKT, P-TSC2 and P-S6K upon all AA(+) Ins(+), AA(+) Ins(-), and AA(-) Ins(+) conditions, compared to the with-crosstalk model (Figs. 4C, and 7B–E).

-in line with the previous comment, it is unclear why the authors claim that "Mathematical simulations revealed that compared to the model without crosstalk (Fig. 3C), the simulation values of P-AKT (P-T308 and P-S473) and P-TSC2 (P-T1462)

obtained from the model with crosstalk fit relatively well (better) with the experimental values (Fig. 4B)". What is the numerical basis for this statement?

Response:

We appreciate the referee's insightful comment. We calculated the correlation indexes (CI) to estimate fitting rates, and the values have been added to the Fig. 4C. The correlation analysis confirmed that compared to the without-crosstalk model, the with-crosstalk model better reproduced the temporal changes of P-AKT and P-TSC2 (Fig. 4C). In addition to without- and with-crosstalk models, we further developed a compartment model of mTORC1 pathway (Fig. 7A), based on all the findings from experiments, such as cytoplasm-lysosome distribution of TSC2 and mTORC1 depending on the AA concentration, and TSC2 dephosphorylation on the lysosomal surface. Among the three models we developed (without-crosstalk, with-crosstalk, and compartment models), the compartment model best reproduced the temporal changes of P-AKT, P-TSC2 and P-S6K upon all AA(+) Ins(+), AA(+) Ins(-), AA(-) Ins(+) conditions (Fig. 4C, 7B-E). These data demonstrates that the factors, such as TSC2 dephosphorylation on the lysosomal surface and the cytoplasm-lysosome distribution of TSC2 and mTOR depending on the AA concentration, are required for the fine-tuning of TSC2 phosphorylation levels and mTORC1 activation.

-All experiments have been done with stimulation of PI3K and mTORC1 activity, and not on the kinetics following deprivation of insulin and nutrients. Because dephosphorylation is the phenomenon to be demonstrated, these de-activation experiments are critical to demonstrate the phenomenon and are surprisingly absent in the MS.

Response:

In accordance with the referee's request, we performed the mathematical simulation with crosstalk by adjusting k_{13} , the rate constant for PP2A (PPase), from 2.0 to 6.0. The data have been added to the new Fig. 4C. Compared to the without-crosstalk model with $k_{13}=0.6$, increased k_{13} in the with-crosstalk model improved the fitting rate. Among them, the simulation with $k_{13}=4.0$ exhibited the mostly high fitting. Therefore, we adopted $k_{13}=4.0$ in the with-crosstalk model (Table 1).

-The use of the p18Rev cells is problematic. The genetic makeup of these cells makes them prone to multiple selective pressures that result in signal transduction adaptation to the lack of p18, and to its reconstitution. Several labs have shown that mTORC1 activation and inhibition heavily impacts on GF signaling by multiple (known and unknown) mechanisms, but all in all, limits the potential physiological relevance of such phenomenon. Several additional non-transformed human and mouse cells are available for signal transduction experiments. Finally, is this phenomenon observed in cell lines with constitutive PI3K-Akt signaling?

Response:

In accordance with the referee's request, we performed experiments using non-transformed HEK293 and parental (control) MEF cells, and the data have been added to the new Figs. 1B, 3A, 4F, 5B–D, S1B–D, S3A, B, S4A, C–J, S5A–C, and S7A, B. The data demonstrate that the responsiveness to AA or Ins observed in p18Rev cells is conserved in HEK293 and MEF cells.

We identified eight PP2A-B56 binding sites ([L/M/F/I]-x-x-[I/L/V]-x-E-x) in TSC2, and for two of these sites, loss-of-function mutations E366K and E1583K have been found in skin, ovary, and rectal cancer cells (Fig. 5E, 5F, and Dataset 3). These E/K mutated cancer cells also have constitutive active mutations in PIK3CA and AKT, suggesting that once TSC2(E/K) is phosphorylated, it is may not be susceptible to dephosphorylation by PP2A. As the referee suggested, we plan to examine whether the reactivity to AA and Ins in cancer cells with constitutively activated PI3K-AKT pathways differs from that in p18Rev, control MEFs, and HEK293 cells. However, considering the limited time available for additional experiments, we intend to pursue these experiments as future work.

-In addition, nutrient and growth factor signaling are known to functionally regulate one another. For example, work by C Thompson has shown that GF signaling is critical for nutrient uptake. Activation of nutrient signaling by AA esters would provide additional support to the model.

Response:

We appreciate the referee's insightful comment and have carefully considered it. Since the referee did not provide the detailed reference for C. Thompson and their studies on the critical role of GF signaling in nutrient uptake, we assume that the referee is

referring to Dr. Craig B. Thompson from Memorial Sloan Kettering Cancer Center and his research, based on the available literature. According to Dr. Craig B Thompson's paper (PMID: 31628187; 12134068), the statement "growth factor (GF) signaling is critical for nutrient uptake" indicates that following GF stimulation in the presence of amino acids (AA(+), GF(+)), nutrient uptake (e.g., AA and glucose) is facilitated by mTORC1 activated downstream of AKT. Therefore, the statement mentioned by the referee appears to describe a positive feedback regulation via activated mTORC1 rather than crosstalk regulation from GF signaling pathway to nutrient (AA) signaling pathway. Nevertheless, as the referee pointed out an important mTORC1-mediated positive feedback mechanism, we plan to incorporate this mechanism to the mathematical simulation to better capture the dynamics of the mTORC1 signaling transduction in the future studies.

-According to the model, a prediction is that locking TSC2 to the cytosolic lysosomal surface would result in increased PP2A-dependent dephosphorylation of TSC2, and presumably independently of AA conditions and Ragulator.

Response:

In accordance with the referee's request, we performed the experiments, and the data have been added to the new Fig. S5A and B. As expectedly, the data demonstrate that artificially anchoring TSC2 to the lysosomal surface results in lower phosphorylation levels of the T1462 residues under Ins stimulation compared to endogenous TSC2, irrespective of the presence or absence of AAs.

-The authors intended to manipulate the nutrient signaling cascade upstream of mTORC1 to assess its impact on PP2A and TSC2, gain- and loss-of-function variants of the Rag GTPases (Rag-BGTP/RagC-GDP, and vice versa) constitute a reliable system to test this idea.

Response:

We appreciate the referee's constructive comment. However, given the limited 3-month revisional period, establishing the cell lines for gain- and loss-of-function variants of the Rag GTPases (RagB^{GTP}/RagC^{GDP}, and vice versa), as requested by the referee, is not feasible. Therefore, we plan to address this as future work. Nevertheless, considering all

the new experiments we have included in this revision, we believe that the data we have gathered sufficiently support our claims, even in the absence of the additional experiments requested by the referee.

-Experiments with phospho-deficient and phospho-mimetic TSC2 mutants are required to demonstrate the relevance of this site on the regulation under AA starvation.

Response:

In accordance with the referee's request, we performed the CoIP experiments using phospho-deficient and phospho-mimetic TSC2 mutants, and the data have been added in the new Fig. S4D and S4E. Using human TSC2 (1740 aa), we generated mutants with the T1395 residue (equivalent to T1462 residue in TSC2 (1807 aa)) changed to phospho-deficient Ala (T1395A) or phospho-mimetic Asp (T1395D) or Glu (T1395E). The CoIP experiments demonstrate that while TSC2-WT does not bind to PP2A because it is not phosphorylated under AA starvation (AA(-) Ins(-)), it interacts with PP2A subunits in a phosphorylation-dependent manner upon Ins stimulation (AA(-) Ins(+)). In contrast, the phospho-mimetic TSC2 mutants (T1395D and T1395E) bind to PP2A subunits even under AA starvation (AA(-) Ins(-)), whereas the phospho-deficient TSC2 mutant (T1395A) does not.

-The IF data is at odds with plenty of literature that has shown that localization of TSC at the lysosomal surface depends on GF signaling, and not on nutrient signaling, and the alternative control of TSC2 by amino acids is controversial. With the limited quality of images and the selected regions for PCA quantification, the conclusions are far from supported and additional experimental robustness would be ideal. This particularly worrisome for the IF on Figure 5 (the new Figure 6), where the PP2A stains and TSC2 stains are uninterpretable and visually weak, and in general for p18KO and rev cells, where lysosomal markers do not seem to have a lysosomal pattern, very evident in Figure 4G (former Figure 4E).

Response:

We appreciate the referee's thoughtful comment.

The GF signaling-dependent lysosomal TSC2 localization mentioned by the referee was performed under conditions different from ours. Although Ins-dependent release of TSC2 from the lysosomal surface during serum starvation has indeed been reported

(PMID: 24529379; 26868506), this was observed after prolonged (16 h) serum starvation, which is considerably longer than the serum and AA starvation conditions (1 h) used in our study.

Serum in culture medium contains high levels of various growth factors (GFs) (PMID: 36133630), which activate mTORC1-mediated upregulation of the surface expression of amino acid (AA) transporters, thereby increasing AA uptake (PMID: 31628187; 12134068). Given concerns about side effects, such as a decrease in AA transporters on cell membranes and a decline in mitochondrial membrane potential after prolonged serum starvation, we limit the duration of AA and serum removal to 1 hour. Under these experimental conditions, in which we monitored the responsiveness to AA or Ins 1 h after AA and serum removal, lysosomal TSC2 localization was primarily inhibited by the AA addition, whereas the effect of Ins was small.

-The specificity of the control of phosphorylation on TSC2 is at least surprising, given the well-established promiscuity of phosphatases. Are other targets of Akt (presumably not dynamically localizing to the lysosomal surface) also affected?

Response:

In accordance with the referee's request, we performed the experiments using an antibody against GSK3 β (at S9) phosphorylation mediated by AKT, and the data have been added in the new Figs. 1B–D, 4D–F, 6E and S3B. These data demonstrate that since GSK3 β localizes only to cytoplasm or nucleus in contrast to TSC2, its phosphorylation level is correlated with AKT phosphorylation, but not with TSC2 phosphorylation.

-Why does Figure 4D (previously Figure 4c) lack the +AA + Ins conditions?

Response:

We apologize for the insufficient explanation. However, we have already conducted the experiments under AA(+) Ins(+) conditions, as shown in Fig. 4D (previously Fig. 4C). To clarify this, we have changed the description of the AA(+) Ins(+) stimuli in Fig. 4D to ensure consistency with the terminology used throughout the entire paper.

Referee #3

Model-Experiment Comparison

The authors claim improved agreement between simulations and experiments after adding crosstalk via PPase (Fig. 3C vs. Fig. 4). However, this improvement is not evident upon visual inspection. The authors should:

- Provide a quantitative comparison using metrics such as explained variance, R^2 , or correlation.

Response:

In accordance with the referee's advice, we calculated the correlation indexes based on the residual sum of squares, and the values have been added to the new Fig. 4C. The data demonstrate that the correlation indexes in the mathematical simulation with crosstalk were higher than those in the simulation without crosstalk (Fig. 4C). We further developed a compartment model depicting cytoplasm-lysosome distribution of TSC2 and mTORC1 depending on the AA concentration, and TSC2 dephosphorylation on the lysosomal surface, and compared with the with-crosstalk model. Mathematical simulations revealed that the temporal changes of P-AKT and P-TSC2 were reproduced equally well in both the with-crosstalk and compartment models (Fig. 4C, 7B, C, E). However, the with-crosstalk model could not reproduce all the temporal changes of P-S6K under three conditions of AA(+) Ins(+), AA(+) Ins(-), and AA(-) Ins(+) (Fig. 7D, E). In contrast, compared to the with-crosstalk model, the compartment model was able to reproduce all the temporal changes of P-S6K upon AA(+) Ins(+), AA(+) Ins(-), and AA(-) Ins(+) (Fig. 7D, E).

- Estimate model parameters directly from their experimental data, rather than relying on literature-based or arbitrary values, which may not suit both models equally.

Response:

We appreciate the referee's constructive comment. In response to the referee's advice, we have estimated the parameters (e.g., k_{13} , l_{13} , k_{25} , l_{25} , l_{26} , k_{28} , and l_{28}) from our experiments or simulation, and these values have been added in the new Table 1.

Simplification in the Model

Experimentally, the authors show that PP2A dephosphorylates TSC2 at the lysosome, and AA starvation induces lysosomal localization of TSC2. However, the model

simplifies this mechanism to AA-dependent sequestration of PP2A. This stark simplification contrasts with the complexity of other model components. The authors should either:

- Simplify the rest of the model for consistency.
- Add more detail to the crosstalk mechanism in line with the experimental findings.

Response:

In accordance with the referee's advice, we have added more experimental details, such as lysosomal localization of TSC2 and mTORC1 depending on intracellular AA or Ins levels, and TSC2 dephosphorylation on the lysosomal surface, to the mathematical model, referred to as the compartment model (Fig. 7A). The mathematical simulation demonstrates that the compartment model best reproduced the temporal changes of P-AKT, P-TSC2 and P-S6K upon all AA(+) Ins(+), AA(+) Ins(-), and AA(-) Ins(+) conditions, compared to the with-crosstalk model (Figs. 4C and 7B–E).

The model is only validated against AA+ and Ins+ conditions, despite the manuscript presenting diverse experimental datasets (e.g., Ins dose-response measurements, pharmacological inhibition, and knockout experiments). Without comparing model simulations to all relevant experimental conditions, the claim that the model enhances quantitative understanding is unsubstantiated. The model does not appear to yield insights beyond what could be inferred from the data alone, contrary to the claims in the discussion.

Response:

We appreciate the referee's constructive comment.

Although the referee mentioned that "The model is only validated against AA+ and Ins+ conditions," we have already presented simulation data under AA(+)Ins(-) and AA(-)Ins(+) conditions in Figs. 3C and 4B, in addition to the simulation data under AA(+)Ins(+) condition.

In accordance with the referee's suggestion, we have now established a more comprehensive mathematical model (a compartment model) incorporating as many experimental results as possible, including the lysosomal localization of TSC2 and mTORC1 depending on the AA concentration and TSC2 dephosphorylation on the lysosomal surface (Fig. 7A). The simulation demonstrates that among the models, the compartment model best reproduced the temporal changes of P-AKT, P-TSC2 and P-

S6K upon all AA(+) Ins(+), AA(+) Ins(-), and AA(-) Ins(+) conditions (Fig. 7B–E).

- Clarify the timepoints post-AA/Ins stimulation for the immunofluorescence experiments.

Response:

In accordance with the referee's request, we have described the time points after stimulation with AA, Ins, or in their combination. The details have been added in the legends of the new Figs. 1E, 4G, 5B, 6A, B, S4F, I, J, S5C, S7C, and D.

- Update the reference in "Western blot analysis using phospho-TSC2- and TSC2-specific antibodies (Fig. EV1A, B (the new Fig. S1A, C))" to refer to Fig. 1.

Response:

We appreciate the referee's thoughtful comment.

We attempted to move Figs. S1A and C to Fig. 1, in accordance with the referee's request. However, because of space limitations and the inclusion of S1B and D (related to S1A and C) in Fig. S1, we were unable to accommodate S1A and C in Fig. 1. Therefore, Fig. S1A and C remain in Supplementary Fig. S1.

- Revise the use of "reaction-diffusion," which typically refers to partial differential equations, as its usage here may be misleading.

Response:

In accordance with the referee's advice, and to avoid any potential confusion, we have changed "reaction-diffusion" to "reaction network system."

July 29, 2025

RE: Life Science Alliance Manuscript #LSA-2025-03206-TR

Dr. Takanori Nakamura
Ehime University
Proteo-Science Center, Department of Pathology
454 Shitsukawa
Toon, Toon 791-0295
Japan

Dear Dr. Nakamura,

Thank you for submitting your revised manuscript entitled "Amino acid-dependent TSC2 dephosphorylation by lysosome-PP2A regulates mTORC1 signaling transduction". Overall both reviewers commend the improvements in place in this revised manuscript. Reviewer 1 made suggestions to clarify the text and figures, which we invite you to consider. We concur with this reviewer on western blotting quantification made in point 4, which must be calculated based on phospho-/total protein. Related to the following point, please also adjust the contrast on protein blots as requested. The remaining points are left to your discretion. We appreciate the concern of this reviewer over the usefulness of the mathematical modeling, however in view of the support from Reviewer 2 we do not suggest removing these data from the manuscript. However please consider the suggestion by Reviewer 2 to improve the presentation of the modeling. We would be happy to publish your paper in Life Science Alliance pending resolution of these requests and final revisions necessary to meet our formatting guidelines.

- Please upload your main manuscript text as an editable .doc file.
- Please upload your main and supplementary figures as single files.
- Please add your main, supplementary figure, and table legends to the main manuscript text after the references section and remove them from the figures.
- Please add the X and Bluesky handles of your host institute/organization as well as your own and/or one of the authors in our system.
- Please be sure that the authorship listing and order are correct and match between the system and the manuscript file.
- Please upload your Tables in editable .doc or Excel format.
- The protein blots in Figure 3 and Figure 6 are duplicated in Figure S3 and S8, respectively. Please note this duplication in the legends for Figures S3 and S8.
- In Figure S4, the PP2A-Ca row in panel D is duplicated in panel E. Please correct this or note this duplication.

A. FINAL FILES:

- An editable version of the final text (.DOC or .DOCX) is needed for copyediting (no PDFs).
- High-resolution figure, supplementary figure and video files uploaded as individual files: See our detailed guidelines for preparing your production-ready images, <https://www.life-science-alliance.org/authors>
- Summary blurb (enter in submission system): A short text summarizing in a single sentence the study (max. 200 characters)

including spaces). This text is used in conjunction with the titles of papers, hence should be informative and complementary to the title. It should describe the context and significance of the findings for a general readership; it should be written in the present tense and refer to the work in the third person. Author names should not be mentioned.

B. MANUSCRIPT ORGANIZATION AND FORMATTING:

Sincerely,

Reviewer #1 (Comments to the Authors (Required)):

I appreciate the authors' efforts to revise the manuscript for LSA. The authors have sufficiently addressed most of my previous concerns. The new data and textual changes strengthen the manuscript further and improve its clarity. Few remaining comments that should be addressed prior to acceptance are listed below.

Major comments

1) Regarding my previous comment #1g, the authors provide a plausible explanation about why the increased TSC2 phosphorylation caused upon OA treatment does not translate into changes in mTORC1 activity (ie S6K phosphorylation) when either AA or Ins is absent. However, this does not explain the lack of an effect when both AA and Ins are present. For instance, in the new panel 4E, lanes 7-8 show a strong increase in p-TSC2(T1462) signal, however p-S6K seems unaffected (in this case, one would expect higher TSC2 phosphorylation to cause further mTORC1 activation). Therefore, as commented before, while PP2A seems to play an important role in the phosphorylation of TSC2 at T1462, the importance and relevance of this mode of regulation in mTORC1 signaling remains rather unclear. Minimally, this should be clearly acknowledged by the authors in the manuscript as a limitation of this study.

2) The labeling of new Fig. 1C is confusing. Does the time (min) refer to addition of the various insulin concentrations or something else? Are lanes 2, 6, and 10, AA starvation alone or double starvation of AA and Ins? If it is the former, why p-Akt and p-TSC2 are low? If it is the latter, the insulin labelling at the top should correspond only to the lanes where insulin was present, not to the 'zero' lanes.

(This comment also applies to additional panels, in which time-course of treatments is performed. Please check for clarity.)

3) Regarding my previous comment #3 "Is the (de)phosphorylation effect specific for the T1462 site of AKT on TSC2, or it works the same also for the other AKT-dependent phospho-sites":

The authors added new data (Fig. 1C,D) showing the response of TSC2(S939) to AA and Ins treatments. However, my question was clearly about whether OA affects only T1462 or the other Akt sites on TSC2 (like S939) as well. Ideally, that would be panel 4E, also showing p-TSC2(S939) blots. Therefore, my previous comment was not addressed by the authors in the revised version of the manuscript.

4) Related to my previous comment #6b: "Fig. 5C (new 6E): In contrast to their statement in the text, the insulin-induced TSC2 phosphorylation in p18-KO cells does not resemble TSC2 phosphorylation upon double +AA +Ins treatment in p18Rev cells, but is rather enhanced further and also starts from increased levels even in the absence of Ins in starved cells."

The new graphs added in the revised manuscript (6F) do not seem to match the visual impression from the blots (6E). For instance, I am not convinced at all that the p-TSC2 bands in lanes 10-12 in the blots have the same intensity as those in lanes 6-8, as shown in the accompanying graph.

Also, how do the statistics look when comparing p-TSC2 in lanes 5 and 9?

Finally, ALL quantifications should be performed using the ratios of phospho to total intensities of each protein, to take into account potential changes in total protein levels.

Minimally, the authors should rephrase their statement to acknowledge that, although +AA +Ins caused stronger TSC phosphorylation compared to -AA +Ins in the p18Rev cells, this was still inferior to TSC2 phosphorylation in p18 KO cells, that demonstrate higher basal and Ins-induced p-TSC2 levels.

4) Related to my previous comment #6g: "Fig. 4C (new 4D): Why are total S6 levels substantially lower in the samples where p-S6 is high?"

Increasing the intensity of the total S6 blots to 'hide' the differences is bad scientific practice and should be avoided!

The authors should instead stick to the initial, lower exposures and acknowledge very clearly these technical limitations in the results and methods sections.

5) Related to my previous comment #7c: "In the introduction, the authors state: "depending on the availability of GFs, hormones, nutrient AAs, or energy, mTORC1 localizes to the lysosomal membrane". This statement is false, as mTORC1 localization to lysosomes was shown to be controlled by AAs and glucose, but not energy or GFs."

Check again the revised text "depending on the availability of GFs, hormones, nutrient AAs, and energy, mTORC1 localizes to the lysosomal membrane in response to AAs or glucose" (page 4, lines 82-84)" for syntax and clarity. Also, again, energy does not regulate the localization of mTORC1 to lysosomes.

6) I remain extremely skeptical about the inclusion of the mathematical modeling in the manuscript. I would insist that there is no added value and it offers no additional insight, on top of the immunoblotting data. Moreover, I do not find the numbers used for 'total proteins', 'volume per cell', intracellular protein concentrations, etc, fully justified. I leave this up to the other expert reviewers and to the editors to decide, however my humble suggestion would be to completely remove this part from the final version of the manuscript.

Reviewer #2 (Comments to the Authors (Required)):

The authors have reasonably addressed all of my concerns.

It might be helpful to mention in the caption that Figure 4C shows fits for different values of the parameter I13, i.e. feedback strengths. Took me a moment to understand.

Re: *Life Science Alliance* manuscript #LSA-2025-03206-TR
Nakamura et al.

Point-by-point Responses to the Reviewers' Comments

We sincerely thank all the reviewers for their thoughtful comments and suggestions on our manuscript.

Reviewer #1 (Comments to the Authors (Required)):

I appreciate the authors' efforts to revise the manuscript for LSA. The authors have sufficiently addressed most of my previous concerns. The new data and textual changes strengthen the manuscript further and improve its clarity. Few remaining comments that should be addressed prior to acceptance are listed below.

Major comments

1) Regarding my previous comment #1g, the authors provide a plausible explanation about why the increased TSC2 phosphorylation caused upon OA treatment does not translate into changes in mTORC1 activity (ie S6K phosphorylation) when either AA or Ins is absent. However, this does not explain the lack of an effect when both AA and Ins are present. For instance, in the new panel 4E, lanes 7-8 show a strong increase in p-TSC2(T1462) signal, however p-S6K seems unaffected (in this case, one would expect higher TSC2 phosphorylation to cause further mTORC1 activation). Therefore, as commented before, while PP2A seems to play an important role in the phosphorylation of TSC2 at T1462, the importance and relevance of this mode of regulation in mTORC1 signaling remains rather unclear. Minimally, this should be clearly acknowledged by the authors in the manuscript as a limitation of this study.

Response:

We appreciate the reviewer's insightful comment.

Based on our experimental and simulation data, we conclude that both 1) lysosomal PP2A-mediated TSC2 dephosphorylation upon AA removal and 2) lysosomal localization of TSC2 and mTOR dependent on intracellular AA levels are required for

fine-tuning full mTORC1 activation.

Given that PP2A constantly distributes to lysosomal surfaces regardless of the intracellular AA or Ins levels (Fig 5, 6A, D, S7C, and F), whereas TSC2 localizes to lysosomes upon AA removal (Fig 4G, H, 6A, C, S4F, S7C, and E), lysosomal PP2A-mediated TSC2 dephosphorylation occurs mainly in the AA(-) Ins(+) condition. Therefore, the increase in TSC2 phosphorylation upon OA treatment was most pronounced under AA(-) Ins(+) stimulation (lanes 5 and 6 in Fig 4E). In contrast, under AA(+) Ins(+) condition, TSC2 does not localize to the lysosomes (Fig 4G, H, 6A, C, S4F, S7C, and E) and is therefore unlikely to be a target for lysosomal PP2A. On the other hand, under AA(+) Ins(+) condition, mTORC1 distributes to lysosomes (where TSC complex are not distributed anymore) and is activated by autophosphorylation via binding to Rheb^{GTP}. We think that this is most likely a reason why lanes 7-8 in Fig 4E show a strong increase in P-TSC2(T1462) signal, but P-S6K (mTORC1 activation) seems unaffected at glance.

However, we believe that the above conclusions are not sufficient to fully answer the reviewer's question regarding the importance and relevance of lysosomal PP2A-mediated TSC2 dephosphorylation in mTORC1 signaling regulation, but rather further experiments are needed. However, we plan to address this point in a future work. In accordance with the reviewer's request, we added the following sentence.

"Although further studies are required to fully understand the molecular mechanism of how PP2A and other phosphatases cooperatively regulate the mTORC1 signal transduction, our findings highlight PP2A-mediated TSC2 dephosphorylation on the lysosomal surface as a key regulatory mechanism that maintains low mTORC1 activation upon Ins stimulation during AA deprivation (Fig 6H)." (page 25, lanes 623–628)

2) The labeling of new Fig. 1C is confusing. Does the time (min) refer to addition of the various insulin concentrations or something else? Are lanes 2, 6, and 10, AA starvation alone or double starvation of AA and Ins? If it is the former, why p-Akt and p-TSC2 are low? If it is the latter, the insulin labelling at the top should correspond only to the lanes where insulin was present, not to the 'zero' lanes.

(This comment also applies to additional panels, in which time-course of treatments is

performed. Please check for clarity.)

Response:

We apologize for the insufficient explanation. To clarify this, we have changed the description of the AA and Ins stimulation, or Ins alone with various concentration from 10^{-5} to 10 $\mu\text{g}/\text{mL}$ in Fig 1C.

3) Regarding my previous comment #3 "Is the (de)phosphorylation effect specific for the T1462 site of AKT on TSC2, or it works the same also for the other AKT-dependent phospho-sites":

The authors added new data (Fig. 1C,D) showing the response of TSC2(S939) to AA and Ins treatments. However, my question was clearly about whether OA affects only T1462 or the other Akt sites on TSC2 (like S939) as well. Ideally, that would be panel 4E, also showing p-TSC2(S939) blots. Therefore, my previous comment was not addressed by the authors in the revised version of the manuscript.

Response:

In accordance with the reviewer's request, we had performed the suggested experiments using a rabbit polyclonal antibody (CST, #3615) against phospho-TSC2 at S939, another major AKT phosphorylation site (PMID:12150915). As we have already mentioned, the P-TSC2(S939) antibody could detect the phosphorylated TSC2 at S939 (Fig S1A, B), but its sensitivity and specificity were apparently lower compared to the anti-phospho-TSC2(T1462) monoclonal antibody (CST, #3617). Because the quantification of phosphorylation levels in Fig 4E require sufficient sensitivity of anti-phosphorylation antibodies but no other antibody was available, except the P-S939 antibody (CST, #3615), we had no choice but to abandon the quantification using the P-S939 antibody with low sensitivity.

4) Related to my previous comment #6b: "Fig. 5C (new 6E): In contrast to their statement in the text, the insulin-induced TSC2 phosphorylation in p18-KO cells does not resemble TSC2 phosphorylation upon double +AA +Ins treatment in p18Rev cells, but is rather enhanced further and also starts from increased levels even in the absence of Ins in starved cells."

The new graphs added in the revised manuscript (6F) do not seem to match the visual

impression from the blots (6E). For instance, I am not convinced at all that the p-TSC2 bands in lanes 10-12 in the blots have the same intensity as those in lanes 6-8, as shown in the accompanying graph.

Also, how do the statistics look when comparing p-TSC2 in lanes 5 and 9?

Finally, ALL quantifications should be performed using the ratios of phospho to total intensities of each protein, to take into account potential changes in total protein levels.

Minimally, the authors should rephrase their statement to acknowledge that, although +AA +Ins caused stronger TSC phosphorylation compared to -AA +Ins in the p18Rev cells, this was still inferior to TSC2 phosphorylation in p18 KO cells, that demonstrate higher basal and Ins-induced p-TSC2 levels.

Response:

In accordance with the reviewer's request, we examined whether total protein levels of AKT, TSC2, and S6K would change under AA starvation, or followed by stimulation with AA and Ins alone, or in combination, but there were not significant changes in Fig 2B, 3B, C, at least under our experimental conditions.

However, as the reviewer pointed out, there were several lanes in Fig 6E where the protein levels of TSC2 and AKT were not equal. We therefore calculated the ratios of phospho to total intensities of TSC2 and AKT from the triplicate immunoblotting data (Fig S8), and the data have been added as the new Fig. 6F and G.

These figures demonstrate that, while there were no significant differences in the phosphorylation level of AKT at T308 between each stimulus (Fig. 6G), the phosphorylation level of TSC2 at T1462 was higher in p18Rev cells stimulated with AA(+) Ins(+) than in those stimulated with AA(-) Ins(+), which was comparable to the level observed in p18-KO cells stimulated with AA(-) Ins(+) (Fig. 6F). In accordance with the reviewer's request, we also calculated *P*-values between lanes 5 and 9 using one-way ANOVA with Tukey's multiple comparison test. However, there were no significant differences in either case (*P*-TSC2(T1462), *P*=0.9928; *P*-AKT(T308), *P*=0.2348).

4) Related to my previous comment #6g: "Fig. 4C (new 4D): Why are total S6 levels substantially lower in the samples where p-S6 is high?"

Increasing the intensity of the total S6 blots to 'hide' the differences is bad scientific

practice and should be avoided!

The authors should instead stick to the initial, lower exposures and acknowledge very clearly these technical limitations in the results and methods sections.

Response:

In accordance with the reviewer's request, we reversed the re-blotting of total S6 in Figs 4D, 4E and 6E to the initial, lower exposure version data.

In addition, we added the following sentence in the legend of Fig 4D and 6E.

"Note that the anti-P-S6 antibodies added during the first immunoblotting cover the antigen S6 protein, and therefore, the newly added anti-S6 antibodies are repelled when the same membrane is reblotted." (pages 65–66, lanes 1681–1683; page 68, lanes 1750–1753)

5) Related to my previous comment #7c: "In the introduction, the authors state:

"depending on the availability of GFs, hormones, nutrient AAs, or energy, mTORC1 localizes to the lysosomal membrane". This statement is false, as mTORC1 localization to lysosomes was shown to be controlled by AAs and glucose, but not energy or GFs."

Check again the revised text "depending on the availability of GFs, hormones, nutrient AAs, and energy, mTORC1 localizes to the lysosomal membrane in response to AAs or glucose" (page 4, lines 82-84)" for syntax and clarity. Also, again, energy does not regulate the localization of mTORC1 to lysosomes.

Response:

We appreciate the reviewer's dedication for reading our manuscript enough.

In accordance with the reviewer's request, we changed the sentence in the Introduction section as follows:

"In contrast, mTORC1 localizes to the lysosomal membrane in response to nutrient amino acids (AAs) or glucose and integrates multiple signals to promote protein synthesis and cell growth by phosphorylating its substrates, including eukaryotic initiation factor 4E-binding proteins (4E-BP) at T37/T46/S65/T70 and p70 S6 kinase (S6K) at T389." (page 4, lines 82–86)

6) I remain extremely skeptical about the inclusion of the mathematical modeling in the

manuscript. I would insist that there is no added value and it offers no additional insight, on top of the immunoblotting data. Moreover, I do not find the numbers used for 'total proteins', 'volume per cell', intracellular protein concentrations, etc, fully justified. I leave this up to the other expert reviewers and to the editors to decide, however my humble suggestion would be to completely remove this part from the final version of the manuscript.

Response:

We appreciate the reviewer's thoughtful comment.

The procedure for calculating 'the total protein per cell' was as follows. First, the p18Rev cells cultured were once stripped from the dish with trypsin and the cell count was measured. Next, 2×10^6 cells were lysed with 200 μL of lysis buffer, and the protein concentration in the cell lysate was measured by the Bradford assay using a spectrophotometer, and the value was 1.69 $\mu\text{g}/\mu\text{L}$. Since 200 μL of cell lysate is made from 2×10^6 cells, the total protein per cell is 169 pg/cell from the formula [$1.69 (\mu\text{g}/\mu\text{L}) \times 200 (\mu\text{L}) \times 1/(2 \times 10^6 (\text{cells}))$].

'The volume per cell' was based on the value from the previous report (PMID: 21768338). Once cells were trypsinized, the diameter (radius) of the sphere-shaped cell is calculated by observing cells with differential interference contrast microscopy. Since the volume of a sphere can be calculated as $(4/3) \times \pi \times (\text{radius})^3$, the average volume per cell is 3.4×10^{-12} L/cell. At the same time, the nucleus (sphere) can also be calculated by staining the nucleus with H \ddot{o} echst 33258, and the average value is calculated to be 0.6×10^{-12} L/cell. The average volume of cytoplasm can also be calculated as 3.4×10^{-12} (L/cell) $- 0.6 \times 10^{-12}$ (L/cell) = 2.8×10^{-12} (L/cell).

'The intracellular protein concentrations' were calculated based on *in vitro* proteome-assisted multiple reaction monitoring (MRM) for protein absolute quantification (iMPAQT) (PMID: 28267743). Proteotypic (Internal standard) peptide sequences for the mTOR signaling pathway were concatenated into two synthetic artificial protein sequences (mTOR_concat01 and mTOR_concat02) (Table 2), and the corresponding cDNAs were synthesized, subcloned into pET21b-QcodeAA-DEST. Isotopically labeled recombinant (artificial) proteins were expressed as His6-tagged molecules in the auxotrophic cells cultured in the presence of [$^{13}\text{C}_6/^{15}\text{N}_2$]Lys and [$^{13}\text{C}_6/^{15}\text{N}_4$]Arg and purified with Ni-resin. Isotopically labeled artificial proteins (concatemers), used as internal standard in mass spectrometry (MS), were spiked into

cell lysates (20 μg each) with four three-fold serial dilutions from 5,000 fmol. The lysates were subjected to enzymatic digestion, and the digested peptides were loaded to MS. Raw MS files were processed with DIA-NN with a spectra library containing the targeted peptides with and without [$^{13}\text{C}_6/^{15}\text{N}_2$]Lys and [$^{13}\text{C}_6/^{15}\text{N}_4$]Arg labeling. Since the internal standard peptide mass (9.26, 27.8, 83.3 and 250 fmol) is known, the absolute abundance of proteins (fmol/ μg) was determined based on the height ratio of light (sample) to heavy (internal standard) peaks in the chromatogram (Data set 1). For example, the absolute amount of AKT1-derived peptide (VTMMNEFEYLK) is calculated to average 26.95 fmol/ μg protein (top row in Data set 1). Since the total amount of protein per cell is 169 pg/cell and the volume per cell is 3.4×10^{-12} L/cell as shown above, the absolute amount (mol/L) of AKT1-derived peptide (VTMMNEFEYLK) is calculated to 1.34×10^{-6} mol/L from the formula $[26.95 \text{ (fmol}/\mu\text{g}) \times 169 \text{ (pg/cell)} \times 1/(3.4 \times 10^{-12} \text{ (L/cell)})]$.

Given that data on intracellular concentrations of regulators of the mTORC1 signaling pathway are not well established, we believe that our intracellular concentration data calculated in this study will help in the quantitative understanding of the mTORC1 pathway.

In addition, in accordance with the reviewers' request (on first revision), we developed the compartment model including all the findings from experiments, such as cytoplasm-lysosome distribution of TSC2 and mTORC1 depending on the AA concentration, and TSC2 dephosphorylation on the lysosomal surface (Fig 7A).

Compared to the without- and with-crosstalk models, the compartment model solely could reproduce all the temporal changes of P-AKT, P-TSC2 and P-S6K upon AA(+), Ins(+), AA(+) Ins(-), and AA(-) Ins(+) (Fig 7B-E).

Reviewer-#1 suggest that the part of mathematical model is removed from the final version of the manuscript. However, we strongly believe that although there may be room to improve our current mathematical model, our mathematical model enhances the quantitative understanding of the temporal regulation within the intricate mTORC1 signaling pathway. In this point, Reviewer-#2 and Editors support our thought and appreciate some credit for our current model. Therefore, we sincerely hope that the mathematical model will remain in the final version.

Reviewer #2 (Comments to the Authors (Required)):

The authors have reasonably addressed all of my concerns.

It might be helpful to mention in the caption that Figure 4C shows fits for different values of the parameter I13, i.e. feedback strengths. Took me a moment to understand.

Response:

We appreciate the reviewer's insightful comment.

In accordance with the reviewer's suggestion, we have specified that I13 is the rate constant for feedback strength in Fig 4A and C.

To further help readers to easily understand that I13 is the rate constant for feedback strength, we added the following sentence in the Fig 4A and C legend.

"I13 is defined as a rate constant for feedback strengths." (page 65, 1668-1669).

"(C) shows fits for different values of the parameter I13 (feedback strength)." (page 65, 1674-1675).

August 11, 2025

RE: Life Science Alliance Manuscript #LSA-2025-03206-TRR

Dr. Takanori Nakamura
Ehime University
Proteo-Science Center, Department of Pathology
454 Shitsukawa
Toon, Toon 791-0295
Japan

Dear Dr. Nakamura,

Thank you for submitting your Research Article entitled "Amino acid-dependent TSC2 dephosphorylation by lysosome-PP2A regulates mTORC1 signaling transduction". It is a pleasure to let you know that your manuscript is now accepted for publication in Life Science Alliance. Congratulations on this interesting work.

DISTRIBUTION OF MATERIALS:

Again, congratulations on a very nice paper. I hope you found the review process to be constructive and are pleased with how the manuscript was handled editorially. We look forward to future exciting submissions from your lab.

Sincerely,
